# Offline Two-Player Zero-Sum Markov Games with KL Regularization

**Claire Chen** [1]  **Yuheng Zhang** [2]  **Xinyu Liu** [3]  **Zixuan Xie** [3]  **Shuze Daniel Liu** [4][5]  **Nan Jiang** [2]

## Abstract

We study the problem of learning Nash equilibria in offline two-player zero-sum Markov games. While existing approaches often rely on explicit pessimism to address distribution shift, we show that KL regularization alone suffices to stabilize learning and guarantee convergence. We first introduce Regularized Offline Sequential Equilibrium (ROSE), a theoretical framework that achieves a fast $\widetilde{\mathcal{O}}(1/n)$ convergence rate under *unilateral concentrability*, improving over the standard $\widetilde{\mathcal{O}}(1/\sqrt{n})$ rates in unregularized settings. We then propose Sequential Offline Self-play Mirror Descent (SOS-MD), a practical model-free algorithm based on least-squares value estimation and iterative self-play updates. We prove that the last iterate of SOS-MD attains the same $\widetilde{\mathcal{O}}(1/n)$ statistical rate up to a vanishing optimization error of order $\widetilde{\mathcal{O}}(1/\sqrt{T})$ in the number of self-play iterations $T$.

## 1. Introduction

Offline Reinforcement Learning (RL) has emerged as a critical paradigm for developing decision-making policies in safety-critical or high-cost domains where online exploration is prohibitive, such as healthcare, autonomous driving, and industrial control (Levine et al., 2020; Fujimoto et al., 2019). While standard offline RL focuses on single-agent optimization, many real-world challenges inherently involve strategic interactions or adversarial dynamics that require a game-theoretic perspective (Liu et al., 2026). Two-player zero-sum Markov games provide the foundational framework for these settings, modeling scenarios where a learner must compete against an opponent to ensure robust performance (Littman, 1994; Zhang et al., 2021). This formulation

is particularly essential for risk-sensitive decision-making, where the adversary represents environmental disturbances or model uncertainties that the agent must withstand (Pinto et al., 2017; Cui & Du, 2022a). In this offline game-theoretic setting, the objective is to identify a Nash equilibrium (NE) solely from previously obtained offline data. Defined as a stable saddle point where no player can improve their outcome by changing their strategy alone, an NE ensures the learned policy remains robust even against the adversary (Başar & Olsder, 1998; Osborne & Rubinstein, 1994).

However, the offline nature of these games introduces the severe challenge of distribution shift, where the learned policy deviates from the behavior distribution represented in the dataset (Xie et al., 2021; Fujimoto et al., 2019). In such cases, learners often suffer from extrapolation error, assigning overoptimistic value estimates to state-action pairs poorly supported by the data (Kumar et al., 2020). To address the challenges of distribution shift, standard approaches typically rely on the principle of explicit pessimism (Liu et al., 2020; Jin et al., 2021b; Rashidinejad et al., 2021). Methods such as Conservative Q-Learning (CQL) (Kumar et al., 2020) and Lower Confidence Bound (LCB) approaches (Jin et al., 2021b; Cui & Du, 2022a) incorporate manually constructed *pessimistic bonuses* to suppress the values of out-of-distribution actions. While effective, these methods often require extensive manual tuning of penalties and complex implementation details (Cui & Du, 2022a; Zhong et al., 2022).

Recently, KL regularization with respect to a reference policy has emerged as a fundamental objective for incorporating prior knowledge and enforcing behavioral constraints (Xiong et al., 2023; Munos et al., 2024). It anchors updates to a reference distribution, such as a pre-trained model or human demonstrations. A prominent example is Reinforcement Learning from Human Feedback (RLHF) (Ouyang et al., 2022), where a KL penalty is explicitly added to the reward function to prevent the aligned model from deviating excessively from the pre-trained language model. This regularization ensures that learned policies remain within the support of the reference distribution (Ye et al., 2024), preventing overoptimistic value estimates in regions without prior guidance (Nayak et al., 2025). However, despite the widespread empirical success of KL regularization, its theoretical impact on sample efficiency remains poorly un-

[1]California Institute of Technology. [2]University of Illinois Urbana-Champaign. [3]University of Virginia. [4]Massachusetts Institute of Technology. [5]Purdue University. Correspondence to: Claire Chen <clairechen@caltech.edu>, Yuheng Zhang <yuhengz2@illinois.edu>.

*Proceedings of the 43rd International Conference on Machine Learning*, Seoul, South Korea. PMLR 306, 2026. Copyright 2026 by the author(s).

*Table 1.* Comparison with previous offline Markov game results. Ye et al. (2024) and our paper study KL-regularized games, while other works consider the unregularized setting. We provide the first $\widetilde{\mathcal{O}}(1/n)$ statistical rate analysis in the multi-step setting, improving over the standard $\widetilde{\mathcal{O}}(1/\sqrt{n})$ rate in prior literature.

| Method | KL-reg. | Multi-step | Statistical Rate |
|---|---|---|---|
| PNVI (Cui & Du, 2022a) | ✗ | ✓ | $\widetilde{\mathcal{O}}(1/\sqrt{n})$ |
| PMVI (Zhong et al., 2022) | ✗ | ✓ | $\widetilde{\mathcal{O}}(1/\sqrt{n})$ |
| BCEL (Zhang et al., 2023) | ✗ | ✓ | $\widetilde{\mathcal{O}}(1/\sqrt{n})$ |
| Ye et al. (2024) | ✓ | ✗ | $\widetilde{\mathcal{O}}(1/\sqrt{n})$ |
| **Ours (SOS-MD)** | ✓ | ✓ | $\widetilde{\mathcal{O}}(1/n)$ |

derstood. Existing analyses typically derive convergence rates identical to those without KL regularization (Ye et al., 2024), thereby missing the opportunity to theoretically justify the accelerated convergence observed in practice.

In this work, we demonstrate that a faster statistical rate of $\widetilde{\mathcal{O}}(1/n)$ is achievable in KL-regularized offline two-player zero-sum Markov games. Our analysis reveals that anchoring to a fixed reference policy is sufficient to stabilize offline learning in this setting, eliminating the need for explicit pessimistic bonuses. We first propose Regularized Offline Sequential Equilibrium (ROSE, Algorithm 1), a theoretical framework that directly computes the regularized Nash equilibrium via backward induction. ROSE achieves this fast rate relying only on *unilateral concentrability*, the minimal coverage condition to learn the NE in the worst case (Cui & Du, 2022a), matching the learnability requirements of offline Markov games. To address the computational intractability of solving exact equilibria in large-scale settings, we further propose Sequential Offline Self-play Mirror Descent (SOS-MD, Algorithm 2), a practical model-free method that approximates the equilibrium via iterative mirror descent. Our primary contributions are as follows:

1. **Pessimism-Free Theoretical Framework**: We introduce ROSE, which establishes that KL regularization eliminates the need for explicit pessimism in offline Markov games. This approach bypasses the technical overhead of constructing and tuning confidence bonuses, a requirement central to prior pessimistic approaches (Jin et al., 2021b; Zhong et al., 2022).

2. **Fast Statistical Rate**: We establish that ROSE achieves a fast $\widetilde{\mathcal{O}}(1/n)$ statistical rate for the duality gap. As summarized in Table 1, this significantly improves upon the standard $\widetilde{\mathcal{O}}(1/\sqrt{n})$ rates found in prior offline game-theoretic literature (Cui & Du, 2022a; Zhong et al., 2022; Zhang et al., 2023). While KL regularization has been studied in single-step or online settings (Ye et al., 2024), our work is the first to unlock the fast-rate phenomenon in the *multi-step* offline game setting. Our analysis reveals that the strong convex-

ity of the KL-regularized objective ensures the regret scales quadratically with the estimation error, extending the fast rates from single-agent systems (Hu et al., 2021) and regularized contextual games (Zhang et al., 2026) to Markov games.

3. **Computationally Efficient Approximation**: We prove that the last iterate of our practical instantiation, SOS-MD, preserves the same $\widetilde{\mathcal{O}}(1/n)$ statistical rate up to a vanishing optimization error of order $\widetilde{\mathcal{O}}(1/\sqrt{T})$. Unlike standard game-theoretic approaches that rely on iterate averaging to guarantee convergence (Freund & Schapire, 1999), we show that the regularity induced by KL regularization enables the *last iterate* of mirror descent to converge directly. By establishing a total error bound that incorporates both optimization and statistical errors, we demonstrate that SOS-MD bridges the gap between theoretical solvability and computational efficiency.

**Conflicts of Interest Disclosure** The authors declare no financial or other substantive conflicts of interest that could reasonably be perceived to influence this submission.

## 2. Related Work

**Offline Reinforcement Learning.** The primary challenge in offline reinforcement learning (RL) arises from the distribution shift between the data collection policy (behavior policy) and the target policy (Liu et al., 2024; Liu & Zhang, 2024; Chen et al., 2025; Liu et al., 2025b). To address the insufficient coverage of the dataset, the principle of *pessimism* has been established as a standard technique for provably efficient learning (Liu et al., 2020; Rashidinejad et al., 2021; Jin et al., 2021b; Xie et al., 2021; Uehara & Sun, 2021; Zhan et al., 2022). The key idea is to penalize state-action pairs that are rarely visited in the dataset to avoid overestimation bias. This pessimistic framework has been shown to be minimax optimal under single-policy concentrability (Li et al., 2024). In contrast to this prevailing paradigm, we propose a new algorithm for KL-regularized Markov games

that operates without explicit pessimism, and our analysis establishes that it also achieves provably efficient learning.

**Markov Games.** Markov games, also known as stochastic games, generalize Markov decision processes (MDPs) to multi-agent settings, where multiple players interact strategically over time under shared environment dynamics (Littman, 1994). These games have been extensively studied in the online setting, covering both two-player zero-sum games (Wei et al., 2017; Bai et al., 2020; 2022; Liu et al., 2021) and multi-player general-sum games (Zhang et al., 2021; Jin et al., 2021a; Song et al., 2021; Mao & Başar, 2023). More recently, there has been growing interest in *offline* Markov games, where agents must learn equilibrium strategies from a fixed dataset without further interaction. Existing works typically address this challenge by extending the principle of pessimism to game-theoretic objectives (Cui & Du, 2022a; Zhong et al., 2022; Cui & Du, 2022b; Zhang et al., 2023). A key distinction of offline Markov games from single-agent RL is the coverage requirement: while single-agent tasks often require only single-policy coverage, learning Nash equilibria in offline Markov games necessitates covering all unilateral deviations, formalized as the *unilateral concentrability* condition (Cui & Du, 2022a; Chen & Zhang, 2026a;b). Our analysis adopts this assumption, which is established as the weakest sufficient coverage condition in the worst case.

**KL-regularized Objectives.** Motivated by the widespread success of Reinforcement Learning from Human Feedback (RLHF) in aligning large language models (Ouyang et al., 2022), there has been a surge of research interest in studying KL-regularized objectives in RL, which penalize deviations from a reference policy. The KL-regularized objectives have been theoretically analyzed in both the single-agent setting (Xiong et al., 2023; Xie et al., 2024; Zhao et al., 2025) and the multi-agent setting (Munos et al., 2024; Ye et al., 2024; Zhang et al., 2025b; Nayak et al., 2025). However, much of this existing theory focuses on bandit settings (single-step) or online interactions. In contrast, our work investigates the multi-step dynamics inherent in sequential decision-making, formulating the problem as an offline KL-regularized Markov game.

## 3. Preliminaries

We consider the offline learning problem in episodic two-player zero-sum Markov games with KL regularization. In this section, we define the problem setup, the regularized objective, and the solution concepts used throughout the paper. For convenience, a summary of the notation used throughout this paper is provided in Table 2 of Appendix A.

### 3.1. Two-Player Zero-Sum Markov Games

We define a finite-horizon Markov game by the tuple $\mathcal{M} = (\mathcal{S}, \mathcal{A}_1, \mathcal{A}_2, P, r, H)$, where $\mathcal{S}$ is the possibly infinite state space, and $\mathcal{A}_1, \mathcal{A}_2$ are the finite action spaces for Player 1 and Player 2, respectively. The game proceeds over $H$ steps. At each step $h \in [H]$, given the current state $s_h$, both players simultaneously choose actions $a_{h,1} \in \mathcal{A}_1$ and $a_{h,2} \in \mathcal{A}_2$. The system transitions to the next state $s_{h+1} \sim P_h(\cdot \mid s_h, a_{h,1}, a_{h,2})$, and Player 1 receives a deterministic reward $r_h^\star(s_h, a_{h,1}, a_{h,2}) \in [0, 1]$. As the game is zero-sum, Player 2 receives reward $-r_h^\star(s_h, a_{h,1}, a_{h,2})$.

We denote a policy for Player $k \in \{1, 2\}$ as $\pi_k = \{\pi_{h,k}\}_{h=1}^H$, where $\pi_{h,k} : \mathcal{S} \rightarrow \Delta(\mathcal{A}_k)$ maps states to distributions over actions. A joint policy is denoted by $\pi = (\pi_1, \pi_2)$. For brevity, we denote the joint action by $a = (a_1, a_2)$ and the product distribution of the policies by $\pi(a|s) = \pi_1(a_1|s)\pi_2(a_2|s)$. We use $\mathbb{E}_{a \sim \pi}[\cdot]$ to represent the expectation with respect to this joint distribution. For a joint policy $\pi$, we define the visitation distribution at step $h$ as $d_h^\pi(s, a_1, a_2) := \mathbb{P}_\pi(s_h = s, a_{h,1} = a_1, a_{h,2} = a_2)$, where the probability is taken over the trajectory induced by $\pi$ and the transition kernel $P$, starting from $s_1 \sim \rho$. We use $d_h^\pi(s) := \sum_{a_1, a_2} d_h^\pi(s, a_1, a_2)$ for the state marginal, so that $\mathbb{E}_{s \sim d_h^\pi}[\cdot]$ and $\mathbb{E}_{(s, a_1, a_2) \sim d_h^\pi}[\cdot]$ both refer consistently to the same underlying distribution.

### 3.2. KL-Regularized Value Functions

To incorporate prior knowledge and restrict the learned policy to valid regions, we employ KL regularization with respect to fixed reference policies $\pi^{\mathrm{ref}} = (\pi_1^{\mathrm{ref}}, \pi_2^{\mathrm{ref}})$. For a regularization coefficient $\eta > 0$, the goal of the agent is to maximize the expected cumulative reward subject to a KL penalty at each step.

Accordingly, the regularized value functions $V_h^\pi$ and $Q_h^\pi$ satisfy the following Bellman recursions. The Q-function is the expected regularized return starting from a state-action pair:

$$Q_h^\pi(s, a_1, a_2) := r_h^\star(s, a_1, a_2) + \mathbb{E}_{s' \sim P_h(\cdot|s, a_1, a_2)}[V_{h+1}^\pi(s')],$$

with $V_{H+1}^\pi(s) = 0$. The value function $V_h^\pi(s)$ incorporates the immediate regularization penalty:

$$V_h^\pi(s) = \mathbb{E}_{a \sim \pi_h(\cdot|s)}[Q_h^\pi(s, a)] - \eta^{-1}\mathrm{KL}\big(\pi_{h,1}(\cdot|s) \,\|\, \pi_{h,1}^{\mathrm{ref}}(\cdot|s)\big)$$

$$+ \eta^{-1}\mathrm{KL}\big(\pi_{h,2}(\cdot|s) \,\|\, \pi_{h,2}^{\mathrm{ref}}(\cdot|s)\big). \tag{1}$$

Under this formulation, minimizing $V_h^\pi$ with respect to $\pi_2$ is equivalent to Player 2 maximizing their own cumulative reward penalized by the KL divergence from $\pi_2^{\mathrm{ref}}$.

## 3.3. Regularized Nash Equilibrium

In the regularized game, a regularized *Nash equilibrium* (NE) is a saddle point of the value function. We define the best-response values for Player 1 and Player 2 respectively:

$$V_h^{\dagger,\pi_2}(s) := \max_{\pi_1'} V_h^{\pi_1',\pi_2}(s), \quad \text{and}$$

$$V_h^{\pi_1,\dagger}(s) := \min_{\pi_2'} V_h^{\pi_1,\pi_2'}(s).$$

A joint policy $\pi^* = (\pi_1^*, \pi_2^*)$ is a regularized Nash equilibrium if, for all $s \in \mathcal{S}$ and $h \in [H]$:

$$V_h^{\pi_1^*,\dagger}(s) = V_h^{\pi^*}(s) = V_h^{\dagger,\pi_2^*}(s).$$

To evaluate the quality of a learned policy $\hat{\pi}$, we use the *Duality Gap*, which measures how exploitable the policy is by optimal adversaries:

$$\text{Gap}(\hat{\pi}) := \mathbb{E}_{s_1 \sim \rho}\left[V_1^{\dagger,\hat{\pi}_2}(s_1) - V_1^{\hat{\pi}_1,\dagger}(s_1)\right],$$

where $\rho$ is the initial state distribution. $\text{Gap}(\hat{\pi}) = 0$ if and only if $\hat{\pi}$ is a Nash equilibrium.

## 3.4. Offline Learning with Function Approximation

We operate in the offline setting, where the learner has access to a static dataset $\mathcal{D}$ collected by a behavior policy. We assume the dataset consists of $n$ independent trajectories, denoted as:

$$\mathcal{D} = \left\{(s_{i,h}, a_{i,h,1}, a_{i,h,2}, r_{i,h}, s_{i,h}')\right\}_{i=1,h=1}^{n,H},$$

where $s_{i,h}' = s_{i,h+1}$ is the state at the next time step. The rewards in the dataset are noisy realizations of the true deterministic reward (Liu et al., 2025a):

$$r_{i,h} = r_h^\star(s_{i,h}, a_{i,h,1}, a_{i,h,2}) + \xi_{i,h},$$

where $\xi_{i,h}$ is independent zero-mean 1-sub-Gaussian noise.

We assume access to a function class $\mathcal{Q} \subset (\mathcal{S} \times \mathcal{A}_1 \times \mathcal{A}_2 \to \mathbb{R})$ to estimate the Q-values. We adopt the standard completeness assumption (e.g., see also Zhang et al. (2023) and Xie et al. (2021)) that for any $V_{h+1}$ induced by the algorithm, the Bellman update $\mathcal{T}_h V_{h+1}$ lies within $\mathcal{Q}$, which we further assume is uniformly bounded. Concentration is applied uniformly over $\mathcal{Q}$ and the algorithm-induced value class.

## 4. Method: Pessimism-Free Learning via KL Regularization

In this section, we propose Regularized Offline Sequential Equilibrium (ROSE, Algorithm 1), a model-free framework that utilizes KL regularization to achieve sample-efficient

---

**Algorithm 1** Regularized Offline Sequential Equilibrium (ROSE)

---

**Require:** Regularization $\eta > 0$, reference policies $\pi_1^{\text{ref}}, \pi_2^{\text{ref}}$, offline dataset $\mathcal{D}$, function class $\mathcal{Q}$.

**Initialization:** Set $\hat{V}_{H+1}(s) = 0$ for all $s$.

**Backward Induction:** For $h = H, \ldots, 1$ do:

1. **Regression Targets:**

$$y_{i,h} = r_{i,h} + \hat{V}_{h+1}(s_{i,h}'), \quad \forall i \in \mathcal{D}_h.$$

2. **Q-Function Estimation:** Compute $\hat{Q}_h$ by solving the least-squares regression:

$$\hat{Q}_h \in \arg\min_{f \in \mathcal{Q}} \sum_{i \in \mathcal{D}_h} \left(f(s_{i,h}, a_{i,h,1}, a_{i,h,2}) - y_{i,h}\right)^2.$$

3. **Equilibrium Computation:** Compute the pair $(\hat{\pi}_{h,1}, \hat{\pi}_{h,2}) \in \Delta(\mathcal{A}_1) \times \Delta(\mathcal{A}_2)$:

$$(\hat{\pi}_{h,1}, \hat{\pi}_{h,2}) = \arg\max_{\pi_1} \min_{\pi_2} \big(\mathbb{E}_{a \sim \pi_1 \times \pi_2}[\hat{Q}_h(s,a)]$$
$$- \eta^{-1}\text{KL}(\pi_1 \| \pi_{h,1}^{\text{ref}}) + \eta^{-1}\text{KL}(\pi_2 \| \pi_{h,2}^{\text{ref}})\big).$$

4. **Value Update:** Update the value function $\hat{V}_h(s)$ as the value of the unique regularized equilibrium:

$$\hat{V}_h(s) = \mathbb{E}_{a \sim \hat{\pi}_{h,1} \times \hat{\pi}_{h,2}}[\hat{Q}_h(s,a)]$$
$$- \eta^{-1}\text{KL}(\hat{\pi}_{h,1} \| \pi_{h,1}^{\text{ref}}) + \eta^{-1}\text{KL}(\hat{\pi}_{h,2} \| \pi_{h,2}^{\text{ref}}).$$

**Output:** Learned policy $\hat{\pi} = \{\hat{\pi}_h\}_{h=1}^H$.

---

learning without explicit pessimistic bonuses. It integrates Least-Squares Value Iteration with an exact Regularized Nash Equilibrium solver to establish the theoretical foundations of our approach. We then discuss the intuition behind why KL regularization effectively handles distribution shift in offline games.

### 4.1. Algorithm Description

Our approach, outlined in Algorithm 1, adopts a fitted Q-iteration approach. The core idea is to iteratively estimate the regularized Q-function using the offline dataset $\mathcal{D}$ and then solve for the regularized Nash equilibrium (NE) of the estimated game at each step.

The algorithm proceeds via backward induction from $h = H$ to 1. At each step $h$, we first construct regression targets $y_{i,h}$ using the reward and the estimated value from the next step, $\hat{V}_{h+1}$. We then estimate the Q-function $\hat{Q}_h$ by solving a least-squares regression problem over the function class $\mathcal{Q}$:
$\hat{Q}_h \in \arg\min_{f \in \mathcal{Q}} \sum_{i=1}^n (f(s_{i,h}, a_{i,h,1}, a_{i,h,2}) - y_{i,h})^2.$

Given the estimated $\hat{Q}_h$, we compute the joint policy $(\hat{\pi}_{h,1}, \hat{\pi}_{h,2})$ that forms a Regularized Nash Equilibrium for the current stage game. Crucially, Algorithm 1 uses the *raw* estimated Q-values directly, without requiring explicit pessimistic penalties or modifications to the regression targets. The regularization is handled entirely by the equilibrium solver step, which maximizes the KL-regularized objective defined in Equation (1).

### 4.2. Implicit Pessimism via KL Regularization

A key distinction of our work is the absence of explicit uncertainty quantification. While standard offline RL methods (Kumar et al., 2020; Jin et al., 2021b) and their game-theoretic extensions (Cui & Du, 2022a; Zhong et al., 2022) rely on manually constructed penalty terms (e.g., LCBs) to suppress out-of-distribution actions, Algorithm 1 achieves robustness intrinsically through the geometry of the KL-regularized objective.

To see this, consider the best-response problem for Player 1 against a fixed opponent policy $\pi_2$. The closed-form solution is given by the Gibbs distribution:

$$\hat{\pi}_1(a_1|s) \propto \pi_1^{\text{ref}}(a_1|s) \cdot \exp\left(\eta \mathbb{E}_{a_2 \sim \pi_2}[\hat{Q}_h(s, a_1, a_2)]\right). \tag{2}$$

This formulation reveals that the reference policy $\pi^{\text{ref}}$ functions as a probabilistic anchor. Crucially, the update operates via multiplicative re-weighting: any action $a_1$ unsupported by the reference (i.e., $\pi_1^{\text{ref}}(a_1|s) \approx 0$) will inherently receive negligible probability in the learned policy $\hat{\pi}_1$, regardless of the estimated Q-value. This mechanism creates a barrier, effectively constraining the agent to the "safe" support of the reference distribution without requiring the explicit construction of confidence intervals.

It is important to note that Equation (2) characterizes the optimality condition for a fixed opponent. In the full game setting, both players optimize their strategies simultaneously, leading to a coupled equilibrium problem. However, computing this exact equilibrium is computationally prohibitive, particularly in large-scale settings. This necessitates the iterative self-play algorithm (SOS-MD) detailed in Section 5.

## 5. Sequential Offline Self-play Mirror Descent

While ROSE (Algorithm 1) establishes the theoretical foundation of the KL-regularized framework, computing the exact regularized Nash equilibrium at each step is often computationally intractable in practice. To bridge this gap, we introduce Sequential Offline Self-play Mirror Descent (SOS-MD), a computationally efficient instantiation that approximates the equilibrium via iterative updates.

Instead of relying on an exact solver, SOS-MD employs a dual Mirror Descent approach to iteratively refine the policy. By exploiting the geometry of the KL-divergence, our method produces simple, closed-form updates based on exponentiated gradients. Notably, for sufficiently small policy updates, our KL-regularized mirror descent rule reduces to the Natural Policy Gradient (NPG) (Kakade, 2001) update. This connection aligns our offline game-theoretic framework with successful online policy optimization algorithms such as TRPO (Schulman et al., 2015).

### 5.1. Algorithm Details

As outlined in Algorithm 2, SOS-MD operates via backward induction. At each stage $h$, the algorithm alternates between estimating the game structure and solving the induced game.

**Stage-wise Q-Function Regression.** We employ the same least-squares regression procedure as Algorithm 1 to estimate $\hat{Q}_h$, minimizing the squared error over $\mathcal{Q}$ with targets derived from the next-step value estimate $\hat{V}_{h+1}$.

**Self-Play Optimization via Mirror Descent.** To resolve the coupled optimization problem where players optimize simultaneously, SOS-MD employs an iterative self-play procedure. We initialize both players' policies to their respective reference policies $\pi^{\text{ref}}$. For $T$ iterations, players update their strategies based on the *marginal payoffs* derived from the current estimated $\hat{Q}_h$. The update rule (3)(4) performs a KL-regularized mirror descent step. Specifically, the update interpolates between the current policy $\pi^{(t)}$ and the reference $\pi^{\text{ref}}$, weighted by the exponential of the estimated advantages. This ensures that the learned policy improves its payoff while remaining close to the reference distribution.

**Value Function Update.** We compute $\hat{V}_h(s)$ using the last iterates $\pi_h^{(T)}$. By leveraging the strong convexity induced by regularization, we ensure the fast convergence of final iterates, eliminating the need for standard time-averaging.

## 6. Theoretical Analysis

In this section, we establish the theoretical guarantees for our approach. We first analyze the statistical sample complexity of the Algorithm 1, demonstrating the "implicit pessimism" property. We then analyze the convergence of SOS-MD (Algorithm 2) and derive the final total error bound, which accounts for both statistical and optimization errors.

### 6.1. Statistical Efficiency

We begin by bounding the sub-optimality of the policy learned by Algorithm 1. This analysis decouples the statistical error from the optimization error, focusing exclusively

---

**Algorithm 2** Sequential Offline Self-play Mirror Descent (SOS-MD)

---

**Require:** Regularization $\eta > 0$, Reference policies $\pi^{\text{ref}} = \{\pi_{h,1}^{\text{ref}}, \pi_{h,2}^{\text{ref}}\}_{h=1}^{H}$, Offline dataset $\mathcal{D}$, Stepsize schedule $\gamma_t$.

1: Initialize terminal value function $\hat{V}_{H+1}(s) = 0$ for all $s \in \mathcal{S}$.
2: **for** $h = H, \dots, 1$ **do**
3:     **Stage-wise Q-Function Regression:**
4:     Construct empirical regression targets using the next-step value estimate:

$$y_{i,h} = r_{i,h} + \hat{V}_{h+1}(s'_{i,h}) \quad \forall i \in \mathcal{D}_h$$

5:     Estimate $\hat{Q}_h$ via stage-wise least-squares minimization over function class $\mathcal{Q}$:

$$\hat{Q}_h \leftarrow \underset{f \in \mathcal{Q}}{\operatorname{argmin}} \sum_{i=1}^{|\mathcal{D}_h|} \left(f(s_{i,h}, a_{i,h,1}, a_{i,h,2}) - y_{i,h}\right)^2$$

6:     **Self-Play Optimization (Dual Mirror Descent):**
7:     Initialize policy iterates: $\pi_{h,1}^{(0)} = \pi_{h,1}^{\text{ref}}$ and $\pi_{h,2}^{(0)} = \pi_{h,2}^{\text{ref}}$.
8:     **for** $t = 0, \dots, T-1$ **do**
9:         **Marginal Payoff Evaluation:**
10:        Compute Player 1's expected payoff against current Player 2 iterate:

$$\hat{q}_{h,1}^{(t)}(s, a_1) = \mathbb{E}_{a_2 \sim \pi_{h,2}^{(t)}(\cdot|s)}[\hat{Q}_h(s, a_1, a_2)]$$

11:       Compute Player 2's expected payoff against current Player 1 iterate:

$$\hat{q}_{h,2}^{(t)}(s, a_2) = \mathbb{E}_{a_1 \sim \pi_{h,1}^{(t)}(\cdot|s)}[\hat{Q}_h(s, a_1, a_2)]$$

12:     **Policy Update:**
13:     Update Player 1 (Ascent - Maximizing $\hat{Q}_h$) using stepsize $\gamma_t$:

$$\pi_{h,1}^{(t+1)}(a_1|s) \propto \pi_{h,1}^{(t)}(a_1|s)^{1-\gamma_t\eta^{-1}} \cdot \exp\left(\gamma_t \cdot \hat{q}_{h,1}^{(t)}(s, a_1)\right) \cdot \pi_{h,1}^{\text{ref}}(a_1|s)^{\gamma_t\eta^{-1}} \tag{3}$$

14:     Update Player 2 (Descent - Minimizing $\hat{Q}_h$) using stepsize $\gamma_t$:

$$\pi_{h,2}^{(t+1)}(a_2|s) \propto \pi_{h,2}^{(t)}(a_2|s)^{1-\gamma_t\eta^{-1}} \cdot \exp\left(-\gamma_t \cdot \hat{q}_{h,2}^{(t)}(s, a_2)\right) \cdot \pi_{h,2}^{\text{ref}}(a_2|s)^{\gamma_t\eta^{-1}} \tag{4}$$

15:     **end for**
16:     **Update Value Function:**
17:     Compute stage $h$ value using the last iterate $\pi_h^{(T)}$:

$$\hat{V}_h(s) \leftarrow \mathbb{E}_{\substack{a_1 \sim \pi_{h,1}^{(T)} \\ a_2 \sim \pi_{h,2}^{(T)}}}\left[\hat{Q}_h(s, a_1, a_2)\right] - \eta^{-1}\text{KL}(\pi_{h,1}^{(T)}\|\pi_{h,1}^{\text{ref}}) + \eta^{-1}\text{KL}(\pi_{h,2}^{(T)}\|\pi_{h,2}^{\text{ref}})$$

18: **end for**
19: **Return:** Learned policy sequence $\pi^{(T)} = (\{\pi_{h,1}^{(T)}\}_{h=1}^{H}, \{\pi_{h,2}^{(T)}\}_{h=1}^{H})$.

---

on the sub-optimality arising from finite sampling and function approximation.

To characterize the coverage of the offline dataset, we first introduce the $D^2$-divergence as a function-class-aware pointwise extrapolation factor, following Ye et al. (2024); Zhao

et al. (2025).

**Definition 6.1** ($D^2$-divergence). Let $\mu = \{\mu_h\}_{h=1}^{H}$ denote the underlying distribution of the offline dataset $\mathcal{D}$, where $\mu_h \in \Delta(\mathcal{S} \times \mathcal{A}_1 \times \mathcal{A}_2)$ is the marginal state-action distribution at step $h$ induced by the behavior policy. The

$D^2$-divergence at $(s, a_1, a_2)$ relative to $\mu_h$ is defined as

$$D_{\mathcal{Q}}^2\big((s, a_1, a_2); \mu_h\big)$$

$$:= \sup_{\substack{Q_1, Q_2 \in \mathcal{Q} \\ \mathbb{E}_{\mu_h}[(Q_1 - Q_2)^2] > 0}} \frac{\big(Q_1(s, a_1, a_2) - Q_2(s, a_1, a_2)\big)^2}{\mathbb{E}_{(s', a_1', a_2') \sim \mu_h}[(Q_1 - Q_2)^2]},$$

with the convention $D_{\mathcal{Q}}^2((s, a_1, a_2); \mu_h) := 0$ when the supremum is over the empty set. The $D^2$-divergence quantifies how much pointwise discrepancy between two functions in $\mathcal{Q}$ can be amplified at $(s, a_1, a_2)$ relative to the expected discrepancy under $\mu_h$.

Equipped with this notion, we adopt the *Unilateral Concentrability* condition proposed by Cui & Du (2022a), which is recognized as the minimal coverage requirement for offline learning in Markov games in the worst case.

**Assumption 6.2** (Unilateral Concentrability). Let $\Pi = \Pi_1 \times \Pi_2$ be the class of joint policies. We define the set of *unilateral deviation policies* $\Pi_{\text{uni}} \subset \Pi$ as the union of policy pairs where at least one player adheres to the Nash equilibrium $\pi^*$:

$$\Pi_{\text{uni}} := \big(\{\pi_1^*\} \times \Pi_2\big) \cup \big(\Pi_1 \times \{\pi_2^*\}\big).$$

There exists a constant $C_{\text{uni}} \geq 1$ such that, for all $h \in [H]$ and every player $i \in \{1, 2\}$,

$$\sup_{\pi_i \in \Pi_i} \mathbb{E}_{(s, a_1, a_2) \sim d_h^{\pi_i \times \pi_{-i}^*}} \big[D_{\mathcal{Q}}^2\big((s, a_1, a_2); \mu_h\big)\big] \leq C_{\text{uni}},$$

where $d_h^{\pi'}$ denotes the visitation distribution at step $h$ induced by $\pi' \in \Pi_{\text{uni}}$.

To quantify the scale of the regularized value functions, we define $\alpha := \inf\{\pi_{\text{ref}, h, i}(a|s) \mid \pi_{\text{ref}, h, i}(a|s) > 0\}$ as the minimum positive probability mass of the reference policy. This definition applies to arbitrary (including deterministic) reference policies, as it analyzes the policy within the support of $\pi_{\text{ref}}$. In our algorithms, the KL regularization term forces the learned policy $\hat{\pi}$ to assign zero mass to any action where $\pi_{\text{ref}}$ vanishes. In practice, standard remedies such as reference smoothing or regularizer clipping (Geist et al., 2019; Nayak et al., 2025) can also be applied to ensure well-defined KL divergence. For notational convenience, we denote $\lambda := 1 + \eta^{-1} \log(1/\alpha)$, $C_{\text{trans}} := e \cdot \alpha^{-4H^2}$ (the one-sided trajectory density-ratio transfer constant; see Lemma B.4), and $C_\alpha := \alpha^{-8H^2}$. With these conditions established, we now present our main statistical result regarding the finite-sample complexity of Algorithm 1.

**Theorem 6.3.** *Let $\hat{\pi}$ be the policy output by Algorithm 1. Suppose Assumption 6.2 holds and the regularization parameter satisfies*

$$\eta \leq \frac{1}{4H^2}. \tag{5}$$

*Then with probability at least $1 - \delta$, the duality gap of $\hat{\pi}$ satisfies:*

$$\text{Gap}(\hat{\pi}) \leq \widetilde{\mathcal{O}}\left(\frac{\eta H^6 C_{\text{uni}} \log(|\mathcal{Q}|/\delta)}{n} \cdot C_\alpha\right).$$

Notably, this result establishes a fast $\widetilde{\mathcal{O}}(1/n)$ rate in the sample size, improving upon the standard $\widetilde{\mathcal{O}}(1/\sqrt{n})$ rate in offline Markov games (Cui & Du (2022a), Zhong et al. (2022), Zhang et al. (2023), see Table 1). The bound scales linearly with the unilateral concentrability $C_{\text{uni}}$ and logarithmically with $|\mathcal{Q}|$, confirming that unilateral coverage of the function-class-aware $D^2$-divergence form suffices for efficient learning with function approximation. For any fixed reference policy, $C_\alpha$ is independent of $n$ and preserves the fast $\widetilde{\mathcal{O}}(1/n)$ rate, which is our key improvement over prior offline Markov-game results.

*Proof Sketch.* The proof of Theorem 6.3 relies on a novel decomposition of the regularized duality gap that bypasses pessimism. We analyze the exploitability of Player 1 (symmetric for Player 2) by splitting the error relative to the true Nash equilibrium $\pi^*$:

$$\underbrace{J(\pi^\dagger, \hat{\pi}_2) - J(\pi_1^*, \pi_2^*)}_{\text{Total Error}}$$

$$= \underbrace{(J(\pi^\dagger, \hat{\pi}_2) - J(\pi_1^*, \hat{\pi}_2))}_{\text{Gap 1: Optimization}} + \underbrace{(J(\pi_1^*, \hat{\pi}_2) - J(\pi_1^*, \pi_2^*))}_{\text{Gap 2: Evaluation}},$$

where $\pi^\dagger$ is the best response to $\hat{\pi}_2$. The two gaps are bounded *separately* (Lemmas 6.6 and 6.5 below) and each ends at the same cumulative squared $Q$-estimation error term.

**Lemma 6.4.** *For any state $s \in \mathcal{S}$ and step $h \in [H]$, the $L_1$-distance between the learned policy $\hat{\pi}_h$ and the regularized Nash equilibrium $\pi_h^*$ is controlled by the estimation error of the Q-function:*

$$\|\hat{\pi}_h(\cdot|s) - \pi_h^*(\cdot|s)\|_1 \leq 2\eta\|\hat{Q}_h(s, \cdot) - Q_h^*(s, \cdot)\|_\infty.$$

Its proof is in Appendix B.4 and follows from the strong monotonicity of the KL-regularized gradient map. The appendix also establishes the squared form $\|\hat{\pi}_h(\cdot|s) - \pi_h^*(\cdot|s)\|_1^2 \leq 4\eta^2 \cdot \xi_h^2(s)$ (see (45)), where $\xi_h^2(s)$ is the *single-player Nash-averaged squared Q-error*

$$\xi_h^2(s) := \max\Big(\max_{a_1} \mathbb{E}_{a_2 \sim \pi_{h,2}^*(\cdot|s)}\big[(\hat{Q}_h - Q_h^*)^2(s, a_1, a_2)\big],$$

$$\max_{a_2} \mathbb{E}_{a_1 \sim \pi_{h,1}^*(\cdot|s)}\big[(\hat{Q}_h - Q_h^*)^2(s, a_1, a_2)\big]\Big). \tag{6}$$

Each branch of the $\max$ is a Player-$i$ deterministic deviation paired with the opponent at Nash, so $\xi_h^2$ lies in $\Pi_{\text{uni}}$-form

and is directly compatible with Assumption 6.2. Using this stability, we bound the optimization gap (Gap 1) and the evaluation gap (Gap 2) independently in the following lemmas, each in terms of the cumulative squared $Q$-estimation error.

**Lemma 6.5.** *Under the side condition* (5)*, the optimization gap is bounded by the cumulative single-player Nash-averaged squared $Q$-estimation error:*

$$\underbrace{J(\pi^\dagger, \hat\pi_2) - J(\pi_1^*, \hat\pi_2)}_{\text{Gap 1}} \leq \mathcal{O}\big(C_{\text{trans}}^2\, \eta^3 H^4 \lambda^2\big)$$
$$\cdot \sum_{h=1}^{H} \sup_{\pi' \in \Pi_{\text{uni}}} \mathbb{E}_{s \sim d_h^{\pi'}}\big[\xi_h^2(s)\big].$$

**Lemma 6.6.** *Under the side condition* (5)*, the evaluation gap is bounded linearly in $\eta$ by the cumulative single-player Nash-averaged squared $Q$-estimation error $\xi_h^2$ (defined in* (6)*):*

$$\underbrace{J(\pi_1^*, \hat\pi_2) - J(\pi_1^*, \pi_2^*)}_{\text{Gap 2}}$$
$$\leq \mathcal{O}\big(\eta\,(C_{\text{trans}} + \lambda^2)\big) \cdot \sum_{h=1}^{H} \mathbb{E}_{s \sim d_h^{\pi_1^*, \hat\pi_2}}\big[\xi_h^2(s)\big].$$

The proofs of Lemmas 6.5 and 6.6 are deferred to Appendices B.2 and B.5, respectively. Both bounds end at the same cumulative squared $Q$-estimation error, which is the key feature of the bandit-style analysis: it enables Gap 1 and Gap 2 to be combined and controlled jointly via Bellman unrolling without circular dependencies. The *squared* error scaling here is the mathematical source of the fast $\widetilde{\mathcal{O}}(1/n)$ rate, as opposed to the standard $\widetilde{\mathcal{O}}(1/\sqrt{n})$ rates associated with unregularized gaps.

Combining Lemmas 6.5 and 6.6 with a recursive Bellman error unrolling completes the proof of Theorem 6.3 (detailed in Appendix B).

## 6.2. Optimization Convergence

Next, we analyze the practical instantiation, SOS-MD (Algorithm 2), where the equilibrium is approximated via $T$ steps of Mirror Descent. We quantify the optimization error of the obtained last-iterate policy $\pi^{(T)}$ relative to the idealized regularized equilibrium $\hat\pi$ analyzed in Algorithm 1.

We control the optimization error via mirror-descent KL convergence (Lemma 6.7) combined with Pinsker's inequality. Two supporting regularity facts—that all SOS-MD iterates remain in a bounded log-density-ratio class relative to $\pi^{\text{ref}}$ (Lemma C.1), and that the regularized duality gap is Lipschitz in the $L_1$ policy distance over this class (Lemma C.4)—are deferred to Appendices C.3 and C.6.

**Lemma 6.7.** *Let $\hat\pi_h$ be the unique Nash equilibrium of the regularized game defined by $\hat{Q}_h$. With the time-varying*

stepsize $\gamma_t = \frac{2\eta}{t+2}$ *in Algorithm 2, the optimization error satisfies, for every $h \in [H]$,*

$$\sup_{s \in \mathcal{S}} \text{KL}\big(\hat\pi_h(\cdot|s) \,\big\|\, \pi_h^{(T)}(\cdot|s)\big)$$
$$\leq \frac{36\,\eta^2 H^2 (1 + \eta^{-1} \log(1/\alpha))^2}{T+1}.$$

The proof, in Appendix C.5, is a state-wise online mirror descent argument with the Hoeffding-type oscillation bound (Lemma C.2) applied within the log-linear-bounded class.

**Theorem 6.8.** *Let $\pi_h^{(T)}$ be the policy generated by Algorithm 2 at step $h$ with stepsize $\gamma_t = \frac{2\eta}{t+2}$. The state-uniform $L_1$-distance to the exact regularized equilibrium $\hat\pi_h$ satisfies*

$$\sup_{s \in \mathcal{S}} \|\hat\pi_h(\cdot|s) - \pi_h^{(T)}(\cdot|s)\|_1 \leq \widetilde{\mathcal{O}}\bigg(\frac{\eta H \lambda}{\sqrt{T}}\bigg),$$

*where $\lambda = 1 + \eta^{-1} \log(1/\alpha)$ is the regularized value-magnitude factor from Lemma B.2.*

*Proof Sketch.* Combine the KL bound of Lemma 6.7 with Pinsker's inequality $\|\pi - \pi'\|_1 \leq \sqrt{2\,\text{KL}(\pi\|\pi')}$ and Jensen's inequality. Detailed in Appendix C.

## 6.3. Total Sample Complexity

By combining the statistical bound for the idealized equilibrium (Theorem 6.3) with the Lipschitz transfer to the SOS-MD last iterate (Lemmas C.4, 6.7, and Theorem 6.8), we derive the final error bound for Algorithm 2.

**Corollary 6.9.** *Let $\pi^{(T)}$ be the policy learned by Algorithm 2 with $T$ iterations per stage using $n$ offline samples and stepsize $\gamma_t = \frac{2\eta}{t+2}$. Under the condition* (5)*, with probability at least $1 - \delta$, the duality gap satisfies*

$$\text{Gap}(\pi^{(T)})$$
$$\leq \underbrace{\widetilde{\mathcal{O}}\bigg(\frac{\eta H^4 \lambda^2}{\sqrt{T}}\bigg)}_{\textit{Optimization Error}} + \underbrace{\widetilde{\mathcal{O}}\bigg(\frac{\eta H^6\, C_{\text{uni}} \log(|\mathcal{Q}|/\delta)}{n} \cdot C_\alpha\bigg)}_{\textit{Statistical Error}}.$$

Its proof is in Appendix C.8. As with the statistical bound, the polylog $\lambda^2$ factor can be absorbed into a multiplicative $\alpha$-dependent constant $C_\alpha' = (1/\alpha)^{\mathcal{O}(1)}$ if desired; we leave it explicit here to mirror the $\lambda$-dependence in Theorem 6.8. This corollary establishes that the last iterate of SOS-MD inherits the fast $\widetilde{\mathcal{O}}(1/n)$ statistical guarantee of the idealized framework up to a vanishing $\widetilde{\mathcal{O}}(1/\sqrt{T})$ optimization error. As $T$ grows, the optimization error vanishes and the total bound approaches the pessimism-free $\widetilde{\mathcal{O}}(1/n)$ statistical rate, demonstrating that SOS-MD provides a computationally tractable approximation to the exact regularized equilibrium without sacrificing the fast statistical convergence. Notably, this $\widetilde{\mathcal{O}}(1/n)$ statistical convergence is significantly

faster than the standard $\widetilde{\mathcal{O}}(1/\sqrt{n})$ rate typically observed in offline reinforcement learning (Xie et al., 2021; Rashidine-jad et al., 2021) and two-player zero-sum offline games (Cui & Du, 2022a).

## 7. Discussion

In this section, we discuss the practical implications of our theoretical framework, particularly for Large Language Model (LLM) alignment, and highlight how our results improve upon the existing literature.

**Implications for LLM Alignment.** Recently, formulating LLM alignment as a two-player zero-sum game has attracted significant research attention (Munos et al., 2024; Wu et al., 2024; Zhang et al., 2025a;b). The primary motivation lies in the limitations of standard Reinforcement Learning from Human Feedback (RLHF) (Ouyang et al., 2022; Xiong et al., 2023; Rafailov et al., 2023), which typically relies on the Bradley-Terry (BT) modeling of human preferences. The BT model inherently assumes the *transitivity* of preferences (i.e., if $A \succ B$ and $B \succ C$, then $A \succ C$). However, this assumption contradicts empirical evidence in human decision-making, where preferences can be intransitive (May, 1954).

To bypass the restrictive BT assumption and capture general preferences, recent works (Azar et al., 2024; Munos et al., 2024) propose a KL-regularized game-theoretic framework. In this setting, the objective is to identify a policy that maximizes the win rate against an adversarial opponent while staying close to a reference policy. However, the majority of this literature focuses on the *online* setting, where the agent can actively query preference signals. Furthermore, existing theoretical analyses are predominantly asymptotic, failing to provide finite-sample complexity bounds.

**Comparison with Offline Nash Learning.** While Ye et al. (2024) provide sample complexity results for the offline setting, their approach largely mirrors the algorithmic design and analysis of unregularized offline Markov games (Cui & Du, 2022a; Zhang et al., 2023). Consequently, they do not fully leverage the geometry of KL regularization, resulting in a standard $\widetilde{\mathcal{O}}(1/\sqrt{n})$ statistical rate. Our work advances beyond this result in three key aspects: (1) Multi-step Dynamics: We study the multi-step setting, generalizing the single-step (bandit) formulation in Ye et al. (2024). This formulation captures more complex LLM applications, such as multi-turn user interactions and tool-use scenarios involving intermediate observations. (2) No Explicit Pessimism: Unlike prior approaches, our algorithm requires no explicit pessimistic bonuses added to the value function, making it align better with practical post-training algorithms. (3) Fast Rates: We develop a novel analysis distinct from the stan-dard pessimistic framework used in offline Markov games, unlocking the strong convexity of the objective to achieve a fast rate of $\widetilde{\mathcal{O}}(1/n)$.

## 8. Conclusion

In this paper, we study offline learning of Nash equilibria in two-player zero-sum Markov games under KL regularization. We first introduce Regularized Offline Sequential Equilibrium (ROSE), an idealized framework for offline learning with KL regularization. We show that KL regularization alone stabilizes learning and yields a fast $\widetilde{\mathcal{O}}(1/n)$ convergence rate under the minimal *unilateral concentrability* assumption. We then propose Sequential Offline Self-play Mirror Descent (SOS-MD), a practical model-free algorithm that replaces the exact equilibrium solver with iterative mirror-descent self-play. Our analysis shows that the last iterate of SOS-MD inherits the same $\widetilde{\mathcal{O}}(1/n)$ statistical guarantee, with optimization error vanishing at rate $\widetilde{\mathcal{O}}(1/\sqrt{T})$ as the number of self-play iterations $T$ grows. Future work includes extending our analysis to partially observable games and developing KL-regularized self-play methods for aligning large language models via Nash Learning from Human Feedback.

## Impact Statement

This paper presents work whose goal is to advance the field of machine learning. There are many potential societal consequences of our work, none of which we feel must be specifically highlighted here.

## Acknowledgment

Yuheng Zhang is supported by a fellowship from the Amazon-Illinois Center on AI for Interactive Conversational Experiences (AICE). Nan Jiang acknowledges funding support from NSF CNS-2112471, NSF CAREER IIS-2141781, and Sloan Fellowship.

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

# A. Notation

*Table 2.* Summary of Notation

| Symbol | Description |
| --- | --- |
| *Markov Games & Reinforcement Learning* | |
| $\mathcal{M}$ | Episodic two-player zero-sum Markov game tuple $(\mathcal{S}, \mathcal{A}_1, \mathcal{A}_2, P, r, H)$ |
| $\mathcal{S}$ | State space |
| $\mathcal{A}_1, \mathcal{A}_2$ | Finite action spaces for Player 1 and Player 2 |
| $H$ | Episode horizon length |
| $P_h(\cdot \mid s, a_1, a_2)$ | Transition probability kernel at step $h$ |
| $r_h^\star(s, a_1, a_2)$ | Deterministic reward function at step $h$ (Player 1 maximizes, Player 2 minimizes) |
| $\rho$ | Initial state distribution |
| $\pi = (\pi_1, \pi_2)$ | Joint policy, where $\pi_k = \{\pi_{h,k}\}_{h=1}^H$ |
| $d_h^\pi$ | State visitation distribution at step $h$ induced by joint policy $\pi$ |
| $V_h^\pi, Q_h^\pi$ | Regularized Value function and Q-function under policy $\pi$ |
| $\pi^\dagger, \pi^\ddagger$ | Best response policy for Player 1 (maximizer) and Player 2 (minimizer) |
| *Regularization & Offline Learning* | |
| $\eta$ | KL regularization coefficient |
| $\pi^{\mathrm{ref}}$ | Fixed reference policy pair $(\pi_1^{\mathrm{ref}}, \pi_2^{\mathrm{ref}})$ |
| $\alpha$ | Minimum positive probability of the reference policy within its support |
| $\mathcal{D}$ | Offline dataset consisting of $n$ trajectories |
| $n$ | Number of samples/trajectories in the offline dataset |
| $\mathcal{Q}$ | Function class for Q-value estimation |
| $\Pi_{\mathrm{uni}}$ | Set of unilateral deviation policies (where at least one player plays Nash) |
| $C_{\mathrm{uni}}$ | Unilateral concentrability coefficient (Assumption 6.2) |
| $\mathrm{Gap}(\pi)$ | Duality gap of joint policy $\pi$ (sum of exploitability) |
| *Algorithms (ROSE & SOS-MD)* | |
| $\hat{Q}_h, \hat{V}_h$ | Estimated Q-function and Value function at step $h$ |
| $\hat{\pi}$ | Policy returned by the exact equilibrium solver (ROSE) |
| $\pi^{(T)}$ | Last iterate policy returned by SOS-MD after $T$ iterations |
| $\gamma$ | Learning rate for Mirror Descent |
| $T$ | Number of iterations for SOS-MD |
| $\mathcal{T}_h$ | Bellman optimality operator at step $h$ |

# B. Proof of Statistical Efficiency (Theorem 6.3)

In this section, we provide the detailed proofs for the statistical efficiency of Algorithm 1. Throughout the appendix proofs (statistical and optimization), we work under the following uniform boundedness condition on the function class $\mathcal{Q}$, used in the least-squares concentration analysis and in the density-ratio and log-linear stability arguments.

**Assumption B.1** (Bounded function class). *The function class $\mathcal{Q}$ is uniformly bounded:*

$$\mathcal{Q} \subseteq \{f : \|f\|_\infty \le C_{\mathrm{val}}\}, \qquad C_{\mathrm{val}} := H\big(1 + \eta^{-1}\log(1/\alpha)\big) = H\lambda,$$

matching the regularized value-function magnitude established in Lemma B.2. This is consistent with completeness, since whenever $V_{h+1}$ obeys $\|V_{h+1}\|_\infty \le C_{\mathrm{val}}$ the Bellman target satisfies $\|\mathcal{T}_h V_{h+1}\|_\infty \le 1 + C_{\mathrm{val}}$ and in particular lies in the $C_{\mathrm{val}}$-ball for $h \le H-1$. If needed, this can also be enforced operationally by clipping $\hat{Q}_h$ to $[-C_{\mathrm{val}}, C_{\mathrm{val}}]$ after the least-squares step. Such bounded function classes are standard in offline RL (Xie et al., 2021) and KL-regularized learning (Xiong et al., 2023; Ye et al., 2024; Zhao et al., 2025).

## B.1. Proof of Lemma B.2

**Lemma B.2** (Uniform Value Bound). *For any opponent policy sequence $\nu = \{\nu_h\}_{h=1}^H$, the magnitude of the regularized best-response value function is uniformly bounded. Specifically, $\forall h \in [H]$:*

$$\|V_h^{\dagger,\nu}\|_\infty \le (H-h+1)\big(1 + \eta^{-1}\log(1/\alpha)\big). \tag{7}$$

We will also prove that, in particular, for all $h \in [H]$,

$$1 + \|V_{h+1}^{\dagger,\nu}\|_\infty \le (H-h+1)\big(1 + \eta^{-1}\log(1/\alpha)\big). \tag{8}$$

*Proof.* We prove (7) by backward induction on $h$. We adopt the standard convention that the terminal value satisfies $V_{H+1}^{\dagger,\nu} \equiv 0$, so (7) holds for $h = H+1$.

Fix $h \in [H]$ and assume (7) holds for $h+1$. Fix any state $s \in \mathcal{S}$. Recall that the value function in a regularized Markov game includes regularization terms for both players. Since Player 2's policy $\nu$ is fixed, their regularization term acts as a state-dependent constant shift. We define:

$$C_{h,s}^\nu := \eta^{-1}\mathrm{KL}(\nu_h(\cdot \mid s)\|\pi_{\mathrm{ref},h,2}(\cdot \mid s)).$$

The value of the best response is the maximization of Player 1's regularized objective plus this constant:

$$V_h^{\dagger,\nu}(s) = \max_{\mu \in \Delta(\mathcal{A}_1)} \left\{\left\langle \mu, Q_{h,\nu}^\dagger(s,\cdot)\right\rangle - \eta^{-1}\mathrm{KL}(\mu \,\|\, \pi_{\mathrm{ref},h,1}(\cdot \mid s))\right\} + C_{h,s}^\nu, \tag{9}$$

where $Q_{h,\nu}^\dagger(s,a_1) := \mathbb{E}_{a_2 \sim \nu_h(\cdot|s)}[Q_h^{\dagger,\nu}(s,a_1,a_2)]$ and

$$Q_h^{\dagger,\nu}(s,a_1,a_2) = r_h^*(s,a_1,a_2) + \mathbb{E}_{s' \sim P_h(\cdot|s,a_1,a_2)}\left[V_{h+1}^{\dagger,\nu}(s')\right].$$

Using $|r_h^*| \le 1$ and Jensen's inequality, for all $(s,a_1,a_2)$,

$$\left|Q_h^{\dagger,\nu}(s,a_1,a_2)\right| \le 1 + \|V_{h+1}^{\dagger,\nu}\|_\infty \qquad \Rightarrow \qquad \|Q_{h,\nu}^\dagger(s,\cdot)\|_\infty \le 1 + \|V_{h+1}^{\dagger,\nu}\|_\infty. \tag{10}$$

Next we upper bound the maximization term in (9) (denoted as $V_{h,s}^{\max}$) using the exact log-sum-exp form. Specifically, using the variational identity $\max_\mu\{\langle \mu, q\rangle - \eta^{-1}\mathrm{KL}(\mu\|\pi)\} = \eta^{-1}\log\sum_a \pi(a)\exp(\eta q(a))$ with $q = Q_{h,\nu}^\dagger(s,\cdot)$ and $\pi = \pi_{\mathrm{ref},h,1}(\cdot \mid s)$, we have:

$$V_{h,s}^{\max} = \eta^{-1}\log \sum_{a \in \mathcal{A}_1} \pi_{\mathrm{ref},h,1}(a \mid s)\exp\left(\eta Q_{h,\nu}^\dagger(s,a)\right).$$

Using the property that $\eta^{-1} \log \sum \pi_i e^{\eta q_i} \in [\min q_i, \max q_i]$, we bound the magnitude:

$$|V_{h,s}^{\max}| \leq \|Q_{h,\nu}^{\dagger}(s, \cdot)\|_{\infty}.$$

Substituting this back into (9):

$$|V_h^{\dagger,\nu}(s)| \leq |V_{h,s}^{\max}| + |C_{h,s}^{\nu}| \leq \|Q_{h,\nu}^{\dagger}(s, \cdot)\|_{\infty} + C_{h,s}^{\nu}. \tag{11}$$

(Note that KL divergence is non-negative, so $|C_{h,s}^{\nu}| = C_{h,s}^{\nu}$).

Under the definition that $\pi_{\text{ref}} \geq \alpha$, for any distribution $p$ we have $\text{KL}(p\|\pi_{\text{ref}}) = \sum p(a) \log(p(a)/\pi_{\text{ref}}(a)) \leq \sum p(a) \log(1/\alpha) = \log(1/\alpha)$. Therefore, $C_{h,s}^{\nu} \leq \eta^{-1} \log(1/\alpha)$ for all $(h, s)$.

Taking the supremum over $s$ in (11) and using (10), we obtain the recursion:

$$\|V_h^{\dagger,\nu}\|_{\infty} \leq \left(1 + \|V_{h+1}^{\dagger,\nu}\|_{\infty}\right) + \eta^{-1} \log(1/\alpha). \tag{12}$$

Applying the induction hypothesis $\|V_{h+1}^{\dagger,\nu}\|_{\infty} \leq (H-h)\left(1 + \eta^{-1} \log(1/\alpha)\right)$ to (12) gives

$$\|V_h^{\dagger,\nu}\|_{\infty} \leq 1 + \eta^{-1} \log(1/\alpha) + (H-h)\left(1 + \eta^{-1} \log(1/\alpha)\right)$$
$$= (H-h+1)\left(1 + \eta^{-1} \log(1/\alpha)\right),$$

which establishes (7). The inequality (8) follows immediately. □

## B.2. Proof of Lemma 6.5 (Optimization Gap Bound)

**Duality Gap Decomposition.** Let $\hat{\pi} = (\hat{\pi}_1, \hat{\pi}_2)$ denote the learned policy pair. Throughout this proof, $J(r, \pi_1, \pi_2)$ denotes the KL-*regularized* expected return defined in Section 3; that is, $J(r, \pi_1, \pi_2) = \mathbb{E}_{s_1 \sim \rho}[V_1^{\pi_1, \pi_2}(s_1)]$ where $V_1^{\pi_1, \pi_2}$ is the regularized value function with reward $r$ and KL penalties for both players. We define the *regularized* best response policies relative to $\hat{\pi}$ under the true game reward $r^*$. Specifically, let $\hat{\pi}_2$ be fixed. The regularized best response of Player 1 is the unique maximizer of the KL-regularized objective:

$$\pi^{\dagger} \in \arg\max_{\pi_1} J(r^*, \pi_1, \hat{\pi}_2),$$

where the argmax is taken over the KL-regularized objective (so the maximizer is automatically a Gibbs-form softmax policy with logits $\eta Q_{h,1}^{\dagger, \hat{\pi}_2} + \log \pi_{h,1}^{\text{ref}}$). Similarly, let $\pi^{\ddagger}$ be the regularized best response of Player 2 against $\hat{\pi}_1$.

Recall that the learned policy $\hat{\pi}$ is the solution to the regularized equilibrium problem with estimated rewards $\hat{r}$:

$$\hat{\pi}_2 \in \arg\min_{\pi_2} \max_{\pi_1} J(\hat{r}, \pi_1, \pi_2), \quad \hat{\pi}_1 \in \arg\max_{\pi_1} \min_{\pi_2} J(\hat{r}, \pi_1, \pi_2).$$

The duality gap of the learned policy pair $\hat{\pi}$ is defined as the sum of exploitabilities:

$$\text{DualGap}(\hat{\pi}) := \underbrace{\left(\max_{\pi_1'} J(r^*, \pi_1', \hat{\pi}_2) - J(r^*, \hat{\pi}_1, \hat{\pi}_2)\right)}_{\text{Player 1 Exploitability}} + \underbrace{\left(J(r^*, \hat{\pi}_1, \hat{\pi}_2) - \min_{\pi_2'} J(r^*, \hat{\pi}_1, \pi_2')\right)}_{\text{Player 2 Exploitability}}.$$

Equivalently, using the best response notation and the value of the true Nash equilibrium $(\pi_1^*, \pi_2^*)$:

$$\text{DualGap}(\hat{\pi}) = \underbrace{\left(J(r^*, \pi^{\dagger}, \hat{\pi}_2) - J(r^*, \pi_1^*, \pi_2^*)\right)}_{\text{Term A}} + \underbrace{\left(J(r^*, \pi_1^*, \pi_2^*) - J(r^*, \hat{\pi}_1, \pi^{\ddagger})\right)}_{\text{Term B}}.$$

Our goal is to bound Term A (Term B follows by symmetry). We decompose Term A as follows:

$$J(r^*, \pi^{\dagger}, \hat{\pi}_2) - J(r^*, \pi_1^*, \pi_2^*) = \underbrace{J(r^*, \pi^{\dagger}, \hat{\pi}_2) - J(r^*, \pi_1^*, \hat{\pi}_2)}_{\textbf{Gap 1: Optimization / Stability}} + \underbrace{J(r^*, \pi_1^*, \hat{\pi}_2) - J(r^*, \pi_1^*, \pi_2^*)}_{\textbf{Gap 2: Evaluation / Fixed-Opponent}}. \tag{13}$$

In the following, we bound **Gap 1** directly in terms of the cumulative squared $Q$-estimation error, by combining the KL-divergence identity for the regularized best-response gap (Step 1 below, which also serves the proof of Lemma 6.6), a logits/Q-difference decomposition (Steps 2–3), and the local-VI stability bound (Lemma 6.4, applied in Step 4.4).

**Step 1: KL Identity for the Evaluation Gap (Gap 2).** We analyze the second term in the decomposition (13), namely

$$\text{Gap 2} := J(r^*, \pi_1^*, \hat{\pi}_2) - J(r^*, \pi_1^*, \pi_2^*) = \mathbb{E}_{s_1 \sim \rho}\big[ V_1^{\pi_1^*, \hat{\pi}_2}(s_1) - V_1^{\pi_1^*, \pi_2^*}(s_1)\big].$$

Fix $(h, s)$ and fix Player 1 to $\pi_1^*$. For any Player 2 policy $\nu = \{\nu_t\}_{t=1}^H$, define the (regularized) stagewise objective of Player 2 at $(h, s)$ as a function of the stage-$h$ mixed action $\nu_h(\cdot \mid s)$:

$$L_{h,s}^\nu\big(\nu_h(\cdot \mid s)\big) := \sum_{a_2 \in \mathcal{A}_2} \nu_h(a_2 \mid s) \underbrace{\mathbb{E}_{a_1 \sim \pi_{h,1}^*(\cdot \mid s)}\big[ Q_h^{\pi_1^*, \nu}(s, a_1, a_2)\big]}_{=: \, h_h^\nu(s, a_2)} + \eta^{-1} \sum_{a_2 \in \mathcal{A}_2} \nu_h(a_2 \mid s) \log \frac{\nu_h(a_2 \mid s)}{\pi_{h,2}^{\text{ref}}(a_2 \mid s)}.$$

Here $Q_h^{\pi_1^*, \nu}$ is defined using the *true continuation* under $(\pi_1^*, \nu)$:

$$Q_h^{\pi_1^*, \nu}(s, a_1, a_2) := r_h^*(s, a_1, a_2) + \mathbb{E}_{s' \sim P_h(\cdot \mid s, a_1, a_2)}\big[ V_{h+1}^{\pi_1^*, \nu}(s')\big], \qquad V_{H+1}^{\pi_1^*, \nu} \equiv 0.$$

Since Player 2 minimizes the value, $L_{h,s}^\nu$ is a strictly convex objective in $\nu_h(\cdot \mid s)$.

**Local optimality of $\pi_2^*$.** Because $(\pi_1^*, \pi_2^*)$ is a (regularized) Nash equilibrium, for every $(h, s)$ we have

$$\pi_{h,2}^*(\cdot \mid s) \in \arg \min_{\mu \in \Delta(\mathcal{A}_2)} L_{h,s}^{\pi_2^*}(\mu).$$

Using a Lagrange multiplier $\lambda$ for the simplex constraint, the first-order optimality condition at $\mu = \pi_{h,2}^*(\cdot \mid s)$ yields, for every $a_2$,

$$0 = \frac{\partial}{\partial \mu(a_2)} \left( L_{h,s}^{\pi_2^*}(\mu) + \lambda \Big( \sum_{a'} \mu(a') - 1 \Big) \right) \Bigg|_{\mu = \pi_{h,2}^*(\cdot \mid s)}$$
$$= h_h^{\pi_2^*}(s, a_2) + \eta^{-1} \Big( 1 + \log \frac{\pi_{h,2}^*(a_2 \mid s)}{\pi_{h,2}^{\text{ref}}(a_2 \mid s)} \Big) + \lambda. \tag{14}$$

Rearranging (14) gives

$$h_h^{\pi_2^*}(s, a_2) = -\eta^{-1} \log \frac{\pi_{h,2}^*(a_2 \mid s)}{\pi_{h,2}^{\text{ref}}(a_2 \mid s)} - C_{h,s}, \tag{15}$$

where $C_{h,s} := \lambda + \eta^{-1}$ is a $(h, s)$-dependent constant.

**Evaluating the local gap.** Now fix $(h, s)$ and compare $\hat{\pi}_{h,2}(\cdot \mid s)$ to $\pi_{h,2}^*(\cdot \mid s)$ *under the same continuation*. Using (15), we obtain the exact identity

$$L_{h,s}^{\pi_2^*}\big(\hat{\pi}_{h,2}(\cdot \mid s)\big) - L_{h,s}^{\pi_2^*}\big(\pi_{h,2}^*(\cdot \mid s)\big) = \eta^{-1} \text{KL}\Big( \hat{\pi}_{h,2}(\cdot \mid s) \, \big\| \, \pi_{h,2}^*(\cdot \mid s)\Big). \tag{16}$$

**Telescoping to Gap 2.** Define $\Delta_h(s) := V_h^{\pi_1^*, \hat{\pi}_2}(s) - V_h^{\pi_1^*, \pi_2^*}(s)$. Using the Bellman equations for $V_h^{\pi_1^*, \hat{\pi}_2}$ and $V_h^{\pi_1^*, \pi_2^*}$ and the definition of $L_{h,s}^{\pi_2^*}$ (which uses the continuation $V_{h+1}^{\pi_1^*, \pi_2^*}$), one obtains the recursion

$$\Delta_h(s) = \Big( L_{h,s}^{\pi_2^*}\big(\hat{\pi}_{h,2}(\cdot \mid s)\big) - L_{h,s}^{\pi_2^*}\big(\pi_{h,2}^*(\cdot \mid s)\big) \Big) + \mathbb{E}_{s' \sim P_h(\cdot \mid s, \pi_1^*, \hat{\pi}_2)}\big[ \Delta_{h+1}(s')\big], \tag{17}$$

with terminal condition $\Delta_{H+1} \equiv 0$. Unrolling (17) yields

$$\text{Gap 2} = \mathbb{E}_{s_1 \sim \rho}[\Delta_1(s_1)] = \sum_{h=1}^H \mathbb{E}_{s \sim d_h^{\pi_1^*, \hat{\pi}_2}} \Big[ L_{h,s}^{\pi_2^*}\big(\hat{\pi}_{h,2}(\cdot \mid s)\big) - L_{h,s}^{\pi_2^*}\big(\pi_{h,2}^*(\cdot \mid s)\big) \Big]. \tag{18}$$

Combining (18) with (16) gives

$$\text{Gap 2} = \frac{1}{\eta} \sum_{h=1}^H \mathbb{E}_{s \sim d_h^{\pi_1^*, \hat{\pi}_2}} \big[ \text{KL}\big(\hat{\pi}_{h,2}(\cdot \mid s) \, \big\| \, \pi_{h,2}^*(\cdot \mid s)\big)\big]. \tag{19}$$

Equation (19) is the exact KL identity for Gap 2 that is invoked by Lemma 6.6 (proved in Appendix B.5). The remainder of this subsection focuses on bounding Gap 1.

**Step 2: Lipschitz Continuity of the Best Response.** We establish the stability of the best response operator in the sequential setting. Throughout this step, all value and $Q$-functions are defined with respect to the true reward function $r^*$.

Fix a time–state pair $(h, s)$ and a Player 2 policy $\nu = \{\nu_{h'}\}_{h'=1}^H$. Define the best-response value of Player 1 against $\nu$ by

$$V_h^{\dagger,\nu}(s) := \max_{\pi_1} V_h^{\pi_1,\nu}(s),$$

and the associated best-response $Q$-function

$$Q_h^{\dagger,\nu}(s, a_1, a_2) := r_h^*(s, a_1, a_2) + \mathbb{E}_{s' \sim P_h(\cdot|s,a_1,a_2)}\big[V_{h+1}^{\dagger,\nu}(s')\big].$$

We also write the opponent-averaged $Q$-vector

$$Q_{h,\nu}^{\dagger}(s, a_1) := \mathbb{E}_{a_2 \sim \nu_h(\cdot|s)}\big[ Q_h^{\dagger,\nu}(s, a_1, a_2) \big].$$

Under KL regularization with reference policy $\pi_{\text{ref},h}(\cdot \mid s)$, the stagewise best response of Player 1 to $\nu$ admits the Gibbs form

$$\pi_{h,1}^{\text{br}}(\cdot \mid s; \nu) = \text{Softmax}\Big(\eta\, Q_{h,\nu}^{\dagger}(s, \cdot) + \log \pi_{\text{ref},h}(\cdot \mid s)\Big). \tag{20}$$

Define the corresponding logits for the best response to $\hat{\pi}$ and $\pi^*$:

$$z_h^{\dagger} := \eta\, Q_{h,\hat{\pi}_2}^{\dagger}(s, \cdot) + \log \pi_{\text{ref},h}(\cdot \mid s), \qquad z_h^* := \eta\, Q_{h,\pi_2^*}^{\dagger}(s, \cdot) + \log \pi_{\text{ref},h}(\cdot \mid s).$$

Since the Softmax mapping is 1-Lipschitz with respect to the $\ell_\infty$-norm, we have

$$\big\|\pi_{h,1}^{\dagger}(\cdot \mid s) - \pi_{h,1}^*(\cdot \mid s)\big\|_1 \leq \big\|z_h^{\dagger} - z_h^*\big\|_\infty$$
$$= \eta\big\|Q_{h,\hat{\pi}_2}^{\dagger}(s, \cdot) - Q_{h,\pi_2^*}^{\dagger}(s, \cdot)\big\|_\infty,$$

where we used the fact that the reference term $\log \pi_{\text{ref},h}(\cdot \mid s)$ cancels out in the logit difference.

Next, we bound the difference between the opponent-averaged $Q$-vectors. By definition, $|r_h^*(s, a_1, a_2)| \leq 1$ for all $(h, s, a_1, a_2)$. Fix $(h, s)$ and $a_1$. We decompose the difference by adding and subtracting a mixed term:

$$Q_{h,\hat{\pi}_2}^{\dagger}(s, a_1) - Q_{h,\pi_2^*}^{\dagger}(s, a_1)$$
$$= \underbrace{\left(\mathbb{E}_{a_2 \sim \hat{\pi}_{h,2}}\big[Q_h^{\dagger,\hat{\pi}}(s, a_1, a_2)\big] - \mathbb{E}_{a_2 \sim \pi_{h,2}^*}\big[Q_h^{\dagger,\hat{\pi}}(s, a_1, a_2)\big]\right)}_{T_1}$$
$$+ \underbrace{\left(\mathbb{E}_{a_2 \sim \pi_{h,2}^*}\big[Q_h^{\dagger,\hat{\pi}}(s, a_1, a_2)\big] - \mathbb{E}_{a_2 \sim \pi_{h,2}^*}\big[Q_h^{\dagger,\pi^*}(s, a_1, a_2)\big]\right)}_{T_2}.$$

**Bounding the first term.** Writing the expectation as a finite sum and applying Hölder's inequality yields

$$|T_1| = \left|\sum_{a_2 \in \mathcal{A}_2} \big(\hat{\pi}_{h,2}(a_2 \mid s) - \pi_{h,2}^*(a_2 \mid s)\big) Q_h^{\dagger,\hat{\pi}}(s, a_1, a_2)\right|$$
$$\leq \|\hat{\pi}_{h,2}(\cdot \mid s) - \pi_{h,2}^*(\cdot \mid s)\|_1 \|Q_h^{\dagger,\hat{\pi}}(s, a_1, \cdot)\|_\infty.$$

Next, using the Bellman form

$$Q_h^{\dagger,\hat{\pi}}(s, a_1, a_2) = r_h^*(s, a_1, a_2) + \mathbb{E}_{s' \sim P_h(\cdot|s,a_1,a_2)}\big[V_{h+1}^{\dagger,\hat{\pi}}(s')\big],$$

we have

$$\|Q_h^{\dagger,\hat{\pi}}(s, a_1, \cdot)\|_\infty \leq \|r_h^*(s, a_1, \cdot)\|_\infty + \sup_{a_2 \in \mathcal{A}_2} \left|\mathbb{E}_{s' \sim P_h(\cdot|s,a_1,a_2)}\big[V_{h+1}^{\dagger,\hat{\pi}}(s')\big]\right|$$
$$\leq 1 + \|V_{h+1}^{\dagger,\hat{\pi}}\|_\infty,$$

where we used $|r_h^*| \leq 1$ and $\left|\mathbb{E}[V_{h+1}^{\dagger,\hat{\pi}}(s')]\right| \leq \mathbb{E}\big[|V_{h+1}^{\dagger,\hat{\pi}}(s')|\big] \leq \|V_{h+1}^{\dagger,\hat{\pi}}\|_\infty$. Therefore,

$$|T_1| \leq \Big(1 + \|V_{h+1}^{\dagger,\hat{\pi}}\|_\infty\Big) \|\hat{\pi}_{h,2}(\cdot \mid s) - \pi_{h,2}^*(\cdot \mid s)\|_1. \tag{21}$$

**Bounding the second term.** For the second term, substituting the Bellman form of $Q_h^{\dagger,\nu}$ gives

$$
\begin{aligned}
T_2 =& \mathbb{E}_{a_2 \sim \pi_{h,2}^*(\cdot|s)}\Big[r_h^*(s,a_1,a_2) + \mathbb{E}_{s' \sim P_h(\cdot|s,a_1,a_2)}[V_{h+1}^{\dagger,\hat{\pi}}(s')] \\
& - r_h^*(s,a_1,a_2) - \mathbb{E}_{s' \sim P_h(\cdot|s,a_1,a_2)}[V_{h+1}^{\dagger,\pi^*}(s')]\Big] \\
=& \mathbb{E}_{a_2 \sim \pi_{h,2}^*(\cdot|s)}\Big[\mathbb{E}_{s' \sim P_h(\cdot|s,a_1,a_2)}[V_{h+1}^{\dagger,\hat{\pi}}(s') - V_{h+1}^{\dagger,\pi^*}(s')]\Big].
\end{aligned}
$$

Now denote the next-step value difference function

$$
\Delta_{h+1}(s') := V_{h+1}^{\dagger,\hat{\pi}}(s') - V_{h+1}^{\dagger,\pi^*}(s').
$$

Then the above becomes

$$
T_2 = \mathbb{E}_{a_2 \sim \pi_{h,2}^*(\cdot|s)}\mathbb{E}_{s' \sim P_h(\cdot|s,a_1,a_2)}[\Delta_{h+1}(s')].
$$

We proceed by analyzing the squared magnitude $|T_2|^2$, which allows us to utilize Jensen's inequality without invoking the global infinity norm. Using the inequality $(a+b)^2 \leq 2a^2 + 2b^2$ on the decomposition $Q - Q^* = T_1 + T_2$:

$$
\left\|Q_{h,\hat{\pi}_2}^{\dagger}(s,\cdot) - Q_{h,\pi_2^*}^{\dagger}(s,\cdot)\right\|_\infty^2 \leq \max_{a_1}\left(2|T_1(s,a_1)|^2 + 2|T_2(s,a_1)|^2\right).
$$

Applying the bound for $T_1$ from (21) yields

$$
|T_1(s,a_1)|^2 \leq \left(1 + \|V_{h+1}^{\dagger,\hat{\pi}}\|_\infty\right)^2 \|\hat{\pi}_{h,2}(\cdot \mid s) - \pi_{h,2}^*(\cdot \mid s)\|_1^2.
$$

For $T_2$, we apply Jensen's inequality $(\mathbb{E}[X])^2 \leq \mathbb{E}[X^2]$:

$$
\begin{aligned}
|T_2(s,a_1)|^2 &= \left|\mathbb{E}_{a_2 \sim \pi_{h,2}^*(\cdot|s)}\mathbb{E}_{s' \sim P_h(\cdot|s,a_1,a_2)}[\Delta_{h+1}(s')]\right|^2 \\
&\leq \mathbb{E}_{a_2 \sim \pi_{h,2}^*(\cdot|s)}\mathbb{E}_{s' \sim P_h(\cdot|s,a_1,a_2)}[\Delta_{h+1}(s')^2].
\end{aligned}
$$

Combining these estimates, we obtain the pointwise (in $s$) squared $Q$-difference bound:

$$
\begin{aligned}
\left\|Q_{h,\hat{\pi}_2}^{\dagger}(s,\cdot) - Q_{h,\pi_2^*}^{\dagger}(s,\cdot)\right\|_\infty^2 \leq\ & 2\left(1 + \|V_{h+1}^{\dagger,\hat{\pi}}\|_\infty\right)^2 \|\hat{\pi}_{h,2}(\cdot \mid s) - \pi_{h,2}^*(\cdot \mid s)\|_1^2 \\
& + 2\max_{a_1}\mathbb{E}_{a_2 \sim \pi_{h,2}^*(\cdot|s)}\mathbb{E}_{s' \sim P_h(\cdot|s,a_1,a_2)}[\Delta_{h+1}(s')^2].
\end{aligned}
\tag{22}
$$

**Step 3: Upper Bounding Gap 1.** We focus on **Gap 1**, which measures the sub-optimality of the Nash equilibrium policy $\pi^*$ against a fixed opponent policy $\hat{\pi}$, relative to the true best response $\pi^\dagger$.

$$
\begin{aligned}
\text{Gap 1} \ :=\ & J(r^*, \pi^\dagger, \hat{\pi}_2) \ -\ J(r^*, \pi_1^*, \hat{\pi}_2) \\
=\ & \mathbb{E}_{s_1 \sim \rho}\left[\ V_1^{\dagger,\hat{\pi}}(s_1) \ -\ V_1^{\pi^*,\hat{\pi}}(s_1)\ \right].
\end{aligned}
$$

For each time–state pair $(h,s)$, define the regularized stage-wise objective of Player 1 against $\hat{\pi}_2$ as

$$
f_{h,s}(\mu) \ :=\ \langle \mu,\ Q_{h,\hat{\pi}_2}^{\dagger}(s,\cdot)\ \rangle\ -\ \eta^{-1}\,\text{KL}(\ \mu\,\|\,\pi_{\text{ref},h,1}(\cdot \mid s)\ ),
$$

where $Q_{h,\hat{\pi}_2}^{\dagger}(s,\cdot)$ is the opponent-averaged best-response $Q$-vector defined above.

The stage-wise best response of Player 1 is given by

$$
\pi_{h,1}^\dagger(\cdot \mid s)\ \in\ \arg\max_{\mu \in \Delta(\mathcal{A}_1)}\ f_{h,s}(\mu).
$$

**Exact Characterization via KL Divergence.** By the first-order optimality conditions of this regularized maximization problem, the sub-optimality gap at each $(h, s)$ admits the exact characterization

$$f_{h,s}(\pi_{h,1}^\dagger) - f_{h,s}(\pi_{h,1}^*) = \eta^{-1} \, \mathrm{KL}\left( \pi_{h,1}^*(\cdot \mid s) \, \middle\| \, \pi_{h,1}^\dagger(\cdot \mid s) \right).$$

Consequently, the overall gap can be expressed as

$$\text{Gap 1} = \eta^{-1} \sum_{h=1}^{H} \mathbb{E}_{s \sim d_h^{\pi^*, \hat\pi}} \left[ \mathrm{KL}\left( \pi_{h,1}^*(\cdot \mid s) \, \middle\| \, \pi_{h,1}^\dagger(\cdot \mid s) \right) \right], \tag{23}$$

where $d_h^{\pi^*, \hat\pi}$ denotes the state visitation distribution induced by $(\pi^*, \hat\pi)$.

**Derivation of** (23). We define the value difference at step $h$ as:

$$\Delta_h(s) := V_h^{\dagger, \hat\pi}(s) - V_h^{\pi^*, \hat\pi}(s).$$

Recall the definition of the regularized value function $V^{\pi_1, \pi_2}$, which includes regularization terms for both players. For a fixed opponent $\hat\pi_2$, the regularization term for Player 2 is independent of Player 1's action. We denote this state-dependent term as:

$$C_{h,s} := \eta^{-1} \mathrm{KL}(\hat\pi_{h,2}(\cdot|s) \, \| \, \pi_{\mathrm{ref},h,2}(\cdot|s)).$$

We also define the regularization penalty for Player 1 as:

$$\mathrm{Reg}_{1,h,s}(\mu) := \eta^{-1} \mathrm{KL}(\mu \, \| \, \pi_{\mathrm{ref},h,1}(\cdot \mid s)).$$

Using these definitions, the stage-wise objective $f_{h,s}(\mu)$ can be written as:

$$f_{h,s}(\mu) = \langle \mu, Q_{h,\hat\pi_2}^\dagger(s, \cdot) \rangle - \mathrm{Reg}_{1,h,s}(\mu).$$

The value of the best response $V_h^{\dagger, \hat\pi}(s)$ is the maximum of the total regularized objective (Player 1's part plus the fixed opponent part):

$$\begin{aligned} V_h^{\dagger, \hat\pi}(s) &= \max_\mu \left( \langle \mu, Q_{h,\hat\pi_2}^\dagger \rangle - \mathrm{Reg}_{1,h,s}(\mu) + C_{h,s} \right) \\ &= \max_\mu f_{h,s}(\mu) + C_{h,s} \\ &= f_{h,s}(\pi_{h,1}^\dagger) + C_{h,s}. \end{aligned}$$

Similarly, the value of the Nash policy $\pi^*$ satisfies the Bellman equation, which we can decompose using the same terms:

$$\begin{aligned} V_h^{\pi^*, \hat\pi}(s) &= \mathbb{E}_{a_1 \sim \pi_{h,1}^*} \left[ r_h^*(s, a_1, \hat\pi_2) + \mathbb{E}_{s'}[V_{h+1}^{\pi^*, \hat\pi}(s')] \right] - \mathrm{Reg}_{1,h,s}(\pi_{h,1}^*) + C_{h,s} \\ &= \underbrace{\mathbb{E}_{a_1 \sim \pi_{h,1}^*} \left[ r_h^*(s, a_1, \hat\pi_2) + \mathbb{E}_{s'}[V_{h+1}^{\dagger, \hat\pi}(s')] \right] - \mathrm{Reg}_{1,h,s}(\pi_{h,1}^*)}_{=f_{h,s}(\pi_{h,1}^*)} + C_{h,s} \\ &\quad - \mathbb{E}_{s' \sim P_h(\cdot|s,\pi^*,\hat\pi)} \left[ V_{h+1}^{\dagger, \hat\pi}(s') - V_{h+1}^{\pi^*, \hat\pi}(s') \right]. \end{aligned}$$

When we subtract $V_h^{\pi^*, \hat\pi}(s)$ from $V_h^{\dagger, \hat\pi}(s)$, the constant term $C_{h,s}$ cancels out:

$$\begin{aligned} \Delta_h(s) &= \left( f_{h,s}(\pi_{h,1}^\dagger) + C_{h,s} \right) - \left( f_{h,s}(\pi_{h,1}^*) + C_{h,s} \right) + \mathbb{E}_{s' \sim P_h}[\Delta_{h+1}(s')] \\ &= f_{h,s}(\pi_{h,1}^\dagger) - f_{h,s}(\pi_{h,1}^*) + \mathbb{E}_{s' \sim P_h(\cdot|s,\pi^*,\hat\pi)}[\Delta_{h+1}(s')]. \end{aligned}$$

Using the exact characterization $f(\pi^\dagger) - f(\pi^*) = \eta^{-1} \mathrm{KL}(\pi^* \| \pi^\dagger)$, we obtain the recurrence:

$$\Delta_h(s) = \eta^{-1} \mathrm{KL}\left( \pi_{h,1}^*(\cdot|s) \, \middle\| \, \pi_{h,1}^\dagger(\cdot|s) \right) + \mathbb{E}_{s' \sim P_h(\cdot|s,\pi^*,\hat\pi)}[\Delta_{h+1}(s')].$$

**Smoothness of the Potential Function.** To bound the KL divergence above, we leverage the convex geometry of the Softmax mapping. Fix a time–state pair $(h, s)$. Define the log-partition function

$$\Phi(\theta) := \log \sum_{a \in \mathcal{A}_1} \exp(\theta_a).$$

Then, by standard results, we know that $\Phi$ is 1-**smooth** with respect to the $\ell_\infty$-norm, i.e., for all $\theta, \theta' \in \mathbb{R}^{|\mathcal{A}_1|}$,

$$\Phi(\theta') \le \Phi(\theta) + \langle \nabla\Phi(\theta), \theta' - \theta \rangle + \frac{1}{2}\|\theta' - \theta\|_\infty^2.$$

**Quadratic Upper Bound on Bregman Divergence.** Fix a time–state pair $(h, s)$. Recall the log-partition function

$$\Phi(\theta) = \log \sum_{a \in \mathcal{A}_1} \exp(\theta_a),$$

whose gradient map $\nabla\Phi(\theta)$ corresponds to the Softmax distribution $P_\theta = \mathrm{Softmax}(\theta)$. The Bregman divergence generated by $\Phi$ is defined as

$$D_\Phi(\theta', \theta) := \Phi(\theta') - \Phi(\theta) - \langle \nabla\Phi(\theta), \theta' - \theta \rangle.$$

A standard result in convex analysis implies that this Bregman divergence coincides exactly with the KL divergence between the induced distributions:

$$D_\Phi(\theta', \theta) = \mathrm{KL}(\ P_\theta \parallel P_{\theta'}\ ).$$

Since $\Phi$ is 1-smooth with respect to the $\ell_\infty$-norm, the associated Bregman divergence admits the quadratic upper bound

$$D_\Phi(\theta', \theta) \le \frac{1}{2}\ \|\theta' - \theta\|_\infty^2.$$

Consequently, for any $\theta, \theta' \in \mathbb{R}^{|\mathcal{A}_1|}$,

$$\mathrm{KL}(\ P_\theta \parallel P_{\theta'}\ ) \le \frac{1}{2}\ \|\theta' - \theta\|_\infty^2. \tag{24}$$

**Application to Policy Logits.** Fix a time–state pair $(h, s)$. Define the natural parameters (logits)

$$\theta_h^\dagger(s, \cdot) := \eta\, Q_{h, \hat{\pi}_2}^\dagger(s, \cdot) + \log \pi_{\mathrm{ref}, h, 1}(\cdot \mid s),$$
$$\theta_h^*(s, \cdot) := \eta\, Q_{h, \pi_2^*}^\dagger(s, \cdot) + \log \pi_{\mathrm{ref}, h, 1}(\cdot \mid s),$$

which induce the policies

$$\pi_{h,1}^\dagger(\cdot \mid s) = \mathrm{Softmax}(\theta_h^\dagger(s, \cdot)), \qquad \pi_{h,1}^*(\cdot \mid s) = \mathrm{Softmax}(\theta_h^*(s, \cdot)).$$

The difference between the logits is therefore

$$\theta_h^\dagger(s, \cdot) - \theta_h^*(s, \cdot) = \eta\big(\ Q_{h, \hat{\pi}_2}^\dagger(s, \cdot) - Q_{h, \pi_2^*}^\dagger(s, \cdot)\big).$$

Applying the quadratic KL bound from (24) yields

$$\mathrm{KL}\Big(\ \pi_{h,1}^*(\cdot \mid s)\ \Big\|\ \pi_{h,1}^\dagger(\cdot \mid s)\ \Big) \le \frac{1}{2}\Big\|\ \theta_h^\dagger(s, \cdot) - \theta_h^*(s, \cdot)\ \Big\|_\infty^2$$
$$= \frac{\eta^2}{2}\Big\|\ Q_{h, \hat{\pi}_2}^\dagger(s, \cdot) - Q_{h, \pi_2^*}^\dagger(s, \cdot)\ \Big\|_\infty^2.$$

Substituting this bound into the representation of Gap 1, we obtain

$$
\begin{aligned}
\text{Gap 1} \;=\;& \eta^{-1} \sum_{h=1}^{H} \mathbb{E}_{s \sim d_h^{\pi^*,\hat{\pi}}} \Big[ \; \text{KL}\Big( \; \pi_{h,1}^*(\cdot \mid s) \; \Big\| \; \pi_{h,1}^\dagger(\cdot \mid s) \; \Big) \Big] \\
\;\leq\;& \frac{\eta}{2} \sum_{h=1}^{H} \mathbb{E}_{s \sim d_h^{\pi^*,\hat{\pi}}} \Big[ \; \Big\| \; Q_{h,\hat{\pi}_2}^\dagger(s,\cdot) \; - \; Q_{h,\pi_2^*}^\dagger(s,\cdot) \; \Big\|_\infty^2 \Big].
\end{aligned}
\tag{25}
$$

By (22), for every $(h,s)$,

$$
\begin{aligned}
\big\| Q_{h,\hat{\pi}_2}^\dagger(s,\cdot) - Q_{h,\pi_2^*}^\dagger(s,\cdot) \big\|_\infty^2 \leq\;& 2\Big(1 + \|V_{h+1}^{\dagger,\hat{\pi}}\|_\infty\Big)^2 \|\hat{\pi}_{h,2}(\cdot \mid s) - \pi_{h,2}^*(\cdot \mid s)\|_1^2 \\
& + 2 \max_{a_1} \mathbb{E}_{a_2 \sim \pi_{h,2}^*(\cdot|s)} \mathbb{E}_{s' \sim P_h(\cdot|s,a_1,a_2)}\big[\Delta_{h+1}(s')^2\big].
\end{aligned}
$$

Substituting into (25) gives

$$
\begin{aligned}
\text{Gap 1} \leq\;& \eta \sum_{h=1}^{H} \Big(1 + \|V_{h+1}^{\dagger,\hat{\pi}}\|_\infty\Big)^2 \|\hat{\pi}_{h,2}(\cdot \mid s) - \pi_{h,2}^*(\cdot \mid s)\|_1^2 \\
& + \eta \sum_{h=1}^{H} \max_{a_1} \mathbb{E}_{a_2 \sim \pi_{h,2}^*(\cdot|s)} \mathbb{E}_{s' \sim P_h(\cdot|s,a_1,a_2)}\big[\Delta_{h+1}(s')^2\big].
\end{aligned}
\tag{26}
$$

**Lemma B.3** (Stability for value difference). *Fix a Player 1 policy $\pi_1^*$ and let $\hat{\pi}_2, \pi_2^*$ be two Player 2 policies. Let $d_h^{\pi_1^*,\hat{\pi}_2}$ denote the state visitation distribution at step $h$ induced by $(\pi_1^*, \hat{\pi}_2)$. Define the value difference and policy difference as*

$$
\Delta_h(s) := V_h^{\pi_1^*,\hat{\pi}_2}(s) - V_h^{\pi_1^*,\pi_2^*}(s), \qquad \Delta\pi_{h,2}(\cdot \mid s) := \hat{\pi}_{h,2}(\cdot \mid s) - \pi_{h,2}^*(\cdot \mid s).
$$

*Define the coefficient*

$$
L_{t+1} := \eta^{-1} + 1 + \|V_{t+1}^{\pi_1^*,\pi_2^*}\|_\infty + \eta^{-1}\log(1/\alpha).
$$

*Then for every $h \in [H]$,*

$$
\mathbb{E}_{s \sim d_h^{\pi_1^*,\hat{\pi}_2}}\big[\Delta_h(s)^2\big] \;\leq\; 6H \sum_{t=h}^{H} L_{t+1}^2 \, \mathbb{E}_{s \sim d_t^{\pi_1^*,\hat{\pi}_2}}\big[\|\Delta\pi_{t,2}(\cdot \mid s)\|_1^2\big].
\tag{27}
$$

*Proof.* Fix $t \in [H]$ and any state $s$. Recall that the value $V_t^{\pi_1^*,\nu}(s)$ is the expected regularized return. Since $\pi_1^*$ is fixed, the regularization term for Player 1 is identical in both $V_t^{\pi_1^*,\hat{\pi}_2}(s)$ and $V_t^{\pi_1^*,\pi_2^*}(s)$ and cancels out in the difference. We decompose $\Delta_t(s)$ into a policy error term and a value propagation term. Let $Q_t^*(s,a) := Q_t^{\pi_1^*,\pi_2^*}(s,a)$.

$$
\begin{aligned}
\Delta_t(s) &= \mathbb{E}_{a \sim \pi_1^* \times \hat{\pi}_2}\big[Q_t^{\pi_1^*,\hat{\pi}_2}(s,a)\big] - \mathbb{E}_{a \sim \pi_1^* \times \pi_2^*}\big[Q_t^*(s,a)\big] + \eta^{-1}\big(\text{KL}(\hat{\pi}_{t,2}\|\pi^{\text{ref}}) - \text{KL}(\pi_{t,2}^*\|\pi^{\text{ref}})\big) \\
&= \underbrace{\mathbb{E}_{a \sim \pi_1^* \times \hat{\pi}_2}\big[Q_t^{\pi_1^*,\hat{\pi}_2}(s,a) - Q_t^*(s,a)\big]}_{(\text{I})} + \underbrace{\mathbb{E}_{a \sim \pi_1^* \times \hat{\pi}_2}\big[Q_t^*(s,a)\big] - \mathbb{E}_{a \sim \pi_1^* \times \pi_2^*}\big[Q_t^*(s,a)\big] + \Delta\text{Reg}_2(s)}_{(\text{II})},
\end{aligned}
$$

where $\text{Reg}_2(\pi) := \eta^{-1}\text{KL}(\pi(\cdot|s) \,\|\, \pi_{t,2}^{\text{ref}}(\cdot|s))$. Term (I) captures the value difference propagation:

$$
(\text{I}) = \mathbb{E}_{a \sim \pi_1^* \times \hat{\pi}_2} \mathbb{E}_{s' \sim P_t(\cdot|s,a)}[\Delta_{t+1}(s')].
$$

Term (II) captures the policy error at step $t$. Note that

$$
\begin{aligned}
\mathbb{E}_{a \sim \pi_1^* \times \hat{\pi}_2}[Q_t^*] - \mathbb{E}_{a \sim \pi_1^* \times \pi_2^*}[Q_t^*] &= \mathbb{E}_{a_1 \sim \pi_{t,1}^*}\Big[\sum_{a_2}(\hat{\pi}_{t,2}(a_2|s) - \pi_{t,2}^*(a_2|s))Q_t^*(s,a_1,a_2)\Big] \\
&= \mathbb{E}_{a_1 \sim \pi_{t,1}^*}\Big[\langle \Delta\pi_{t,2}(\cdot|s), \, Q_t^*(s,a_1,\cdot)\rangle\Big].
\end{aligned}
$$

To bound the regularization difference, we apply the Mean Value Theorem. There exists a policy $\xi$ on the line segment between $\hat{\pi}_{t,2}(\cdot|s)$ and $\pi^*_{t,2}(\cdot|s)$ such that

$$\text{Reg}_2(\hat{\pi}_{t,2}) - \text{Reg}_2(\pi^*_{t,2}) = \langle \nabla\text{Reg}_2(\xi), \hat{\pi}_{t,2} - \pi^*_{t,2} \rangle.$$

We now derive an explicit bound $B_{\text{reg}}$ on $\|\nabla\text{Reg}_2(\xi)\|_\infty$ over all such convex combinations $\xi$. By construction, the regularized best-response policies $\hat{\pi}_{t,2}$ and $\pi^*_{t,2}$ take the Gibbs (softmax) form

$$\pi(a) \propto \pi^{\text{ref}}_{t,2}(a|s) \exp(\eta Q(s,a))$$

for some bounded payoff vector $Q$. Since the reference policy satisfies $\pi^{\text{ref}}_{t,2}(a|s) \geq \alpha$ for all $(s,a)$ in the support of $\pi^{\text{ref}}$, and $\|Q\|_\infty \leq H(1 + \eta^{-1}\log(1/\alpha))$ (by Lemma B.2 for the $\pi^*_{t,2}$ case where $Q$ marginalizes $Q^*_t$, and by Assumption B.1 for the $\hat{\pi}_{t,2}$ case where $Q$ marginalizes $\hat{Q}_t$), the unnormalized log-density satisfies

$$\log\frac{\pi(a)}{\pi^{\text{ref}}_{t,2}(a|s)} = \eta Q(s,a) - \log Z, \qquad \left|\log\frac{\pi(a)}{\pi^{\text{ref}}_{t,2}(a|s)}\right| \leq 2\eta H(1 + \eta^{-1}\log(1/\alpha)).$$

Therefore $\pi(a) \geq \alpha \cdot \exp\left(-2\eta H(1 + \eta^{-1}\log(1/\alpha))\right)$ is strictly positive on the support of $\pi^{\text{ref}}$. Any convex combination $\xi = (1-\theta)\hat{\pi}_{t,2} + \theta\pi^*_{t,2}$ (for $\theta \in [0,1]$) inherits this strict positivity, so $\xi$ lies in the interior of the simplex where $\nabla_\pi \eta^{-1}\text{KL}(\pi\|\pi^{\text{ref}}_{t,2}) = \eta^{-1}(1 + \log(\pi/\pi^{\text{ref}}_{t,2}))$ is well-defined and bounded:

$$\|\nabla\text{Reg}_2(\xi)\|_\infty \leq \eta^{-1}\left(1 + \|\log(\xi/\pi^{\text{ref}}_{t,2})\|_\infty\right) \leq \eta^{-1}\left(1 + 2\eta H(1 + \eta^{-1}\log(1/\alpha))\right) = \eta^{-1} + 2H\left(1 + \eta^{-1}\log(1/\alpha)\right).$$

Hence $B_{\text{reg}} := \sup_\xi \|\nabla\text{Reg}_2(\xi)\|_\infty \leq \eta^{-1} + 2H\lambda$, where $\lambda := 1 + \eta^{-1}\log(1/\alpha)$. Note: in the regime where the reference policy is close to uniform (small $\log(1/\alpha)$), the bare $\eta^{-1}$ term need not be absorbed by $H\lambda$; we therefore keep it explicit throughout the rest of this proof and absorb it together with the other constants into the headline $(1/\alpha)^{\mathcal{O}(H^2)}$ multiplier of Theorem 6.3 via the chain of Step 4. Applying Hölder's inequality:

$$\left|\text{Reg}_2(\hat{\pi}_{t,2}) - \text{Reg}_2(\pi^*_{t,2})\right| \leq \|\nabla\text{Reg}_2(\xi)\|_\infty\|\hat{\pi}_{t,2} - \pi^*_{t,2}\|_1 \leq B_{\text{reg}}\|\Delta\pi_{t,2}(\cdot|s)\|_1.$$

Combining this with the $Q$-term bound yields the final Lipschitz coefficient for Term (II):

$$\begin{aligned}
|(\text{II})| &\leq \|Q^*_t(s,\cdot,\cdot)\|_\infty\|\Delta\pi_{t,2}(\cdot|s)\|_1 + B_{\text{reg}}\|\Delta\pi_{t,2}(\cdot|s)\|_1 \\
&= \left(\|Q^*_t(s,\cdot,\cdot)\|_\infty + B_{\text{reg}}\right)\|\Delta\pi_{t,2}(\cdot|s)\|_1 \\
&\leq 3L_{t+1}\|\Delta\pi_{t,2}(\cdot|s)\|_1,
\end{aligned}$$

where the bound $\|Q^*_t\|_\infty + B_{\text{reg}} \leq 3L_{t+1}$ uses Lemma B.2 (gives $\|Q^*_t\|_\infty \leq 1 + \|V^*_{t+1}\|_\infty \leq L_{t+1}$) and $B_{\text{reg}} \leq \eta^{-1} + 2H\lambda \leq 2L_{t+1}$. The $\eta^{-1}$ summand in $L_{t+1}$ (introduced precisely to absorb the bare KL-gradient leading term in $B_{\text{reg}}$) ensures this bound remains valid even when the reference policy is close to uniform (small $\log(1/\alpha)$); under (5) we have $\eta^{-1} \geq 4H^2 \geq H\lambda$ in this regime, so the $\eta^{-1}$ summand is dominated by $L_{t+1}$.

Thus, we have the pointwise recursive bound:

$$|\Delta_t(s)| \leq L_{t+1}\|\Delta\pi_{t,2}(\cdot\mid s)\|_1 + \left|\mathbb{E}_{a\sim\pi^*_1\times\hat{\pi}_2}\mathbb{E}_{s'\sim P_t(\cdot|s,a)}[\Delta_{t+1}(s')]\right|.$$

Square both sides and apply the weighted AM-GM inequality $(x+y)^2 \leq (1+H)x^2 + (1+\frac{1}{H})y^2$:

$$\Delta_t(s)^2 \leq (1+H)L^2_{t+1}\|\Delta\pi_{t,2}(\cdot\mid s)\|^2_1 + \left(1 + \frac{1}{H}\right)\left(\mathbb{E}_{a\sim\pi^*_1\times\hat{\pi}_2}\mathbb{E}_{s'}[\Delta_{t+1}(s')]\right)^2.$$

Apply Jensen's inequality to the second term: $(\mathbb{E}Z)^2 \leq \mathbb{E}[Z^2]$.

$$\Delta_t(s)^2 \leq 2HL^2_{t+1}\|\Delta\pi_{t,2}(\cdot\mid s)\|^2_1 + \left(1 + \frac{1}{H}\right)\mathbb{E}_{a\sim\pi^*_1\times\hat{\pi}_2}\mathbb{E}_{s'\sim P_t}[\Delta_{t+1}(s')^2].$$

(Using $1 + H \leq 2H$). Now take the expectation with respect to $s \sim d^{\pi^*_1,\hat{\pi}_2}_t$. By definition, if $s \sim d^{\pi^*_1,\hat{\pi}_2}_t$ and we take actions $a \sim \pi^*_1 \times \hat{\pi}_2$ and transition $s' \sim P_t$, the next state $s'$ is distributed according to $d^{\pi^*_1,\hat{\pi}_2}_{t+1}$. Thus:

$$\mathbb{E}_{s\sim d^{\pi^*_1,\hat{\pi}_2}_t}[\Delta_t(s)^2] \leq 2HL^2_{t+1}\mathbb{E}_{s\sim d^{\pi^*_1,\hat{\pi}_2}_t}[\|\Delta\pi_{t,2}\|^2_1] + \left(1 + \frac{1}{H}\right)\mathbb{E}_{s'\sim d^{\pi^*_1,\hat{\pi}_2}_{t+1}}[\Delta_{t+1}(s')^2].$$

Unrolling this recurrence from $t = h$ to $H$ (with $\Delta_{H+1} \equiv 0$):

$$\mathbb{E}_{d_h}[\Delta_h^2] \le 2H \sum_{t=h}^{H} \left(1 + \frac{1}{H}\right)^{t-h} L_{t+1}^2 \mathbb{E}_{d_t}[\|\Delta\pi_{t,2}\|_1^2].$$

Since $(1 + \frac{1}{H})^H \le e < 3$, the pre-factor is bounded by $6H$, yielding (27). $\qquad\square$

**Step 4: Combining the two terms in the Gap 1 bound.** Recall from (26) that

$$\text{Gap } 1 \le \eta \sum_{h=1}^{H} \left(1 + \|V_{h+1}^{\dagger,\hat{\pi}}\|_\infty\right)^2 \mathbb{E}_{s \sim d_h^{\pi_1^*,\hat{\pi}_2}} \left[\left\|\hat{\pi}_{h,2}(\cdot|s) - \pi_{h,2}^*(\cdot|s)\right\|_1^2\right]$$

$$+ \eta \sum_{h=1}^{H} \max_{a_1} \mathbb{E}_{a_2 \sim \pi_{h,2}^*(\cdot|s)} \mathbb{E}_{s' \sim P_h(\cdot|s,a_1,a_2)}[\Delta_{h+1}(s')^2].$$

**Step 4.1: Controlling the first term.** By Lemma B.2, for every $h \in [H]$,

$$1 + \|V_{h+1}^{\dagger,\hat{\pi}}\|_\infty \le H\left(1 + \eta^{-1}\log(1/\alpha)\right).$$

Plugging into the first term of (26) yields

$$\eta \sum_{h=1}^{H} \left(1 + \|V_{h+1}^{\dagger,\hat{\pi}}\|_\infty\right)^2 \mathbb{E}_{s \sim d_h^{\pi_1^*,\hat{\pi}_2}} \left[\left\|\hat{\pi}_{h,2}(\cdot|s) - \pi_{h,2}^*(\cdot|s)\right\|_1^2\right]$$

$$\le \eta H^2 \left(1 + \eta^{-1}\log(1/\alpha)\right)^2 \sum_{h=1}^{H} \mathbb{E}_{s \sim d_h^{\pi_1^*,\hat{\pi}_2}} \left[\left\|\hat{\pi}_{h,2}(\cdot|s) - \pi_{h,2}^*(\cdot|s)\right\|_1^2\right]. \tag{28}$$

**Step 4.2: Controlling the second term.** Remember we defined

$$L_{t+1} := \eta^{-1} + 1 + \|V_{t+1}^{\pi_1^*,\pi_2^*}\|_\infty + \eta^{-1}\log(1/\alpha).$$

**$\Delta$-decomposition (bridging (22) and Lemma B.3).** The $\Delta_{h+1}$ in (26) is the best-response value difference $V_{h+1}^{\dagger,\hat{\pi}}(s') - V_{h+1}^{\dagger,\pi^*}(s')$, while Lemma B.3 bounds the value difference $V_{h+1}^{\pi_1,\hat{\pi}_2}(s') - V_{h+1}^{\pi_1,\pi_2^*}(s')$ for any fixed Player 1 policy $\pi_1$. Let $\pi_1^\dagger(\hat{\pi}_2)$ denote the regularized best response of Player 1 to $\hat{\pi}_2$. Since $V_{h+1}^{\dagger,\hat{\pi}} = V_{h+1}^{\pi_1^\dagger(\hat{\pi}_2),\hat{\pi}_2}$ and $V_{h+1}^{\dagger,\pi^*} = V_{h+1}^{\pi_1^*,\pi_2^*} \ge V_{h+1}^{\pi_1^\dagger(\hat{\pi}_2),\pi_2^*}$ by Nash optimality of $\pi_1^*$ against $\pi_2^*$,

$$\Delta_{h+1}(s') \le R(s') := V_{h+1}^{\pi_1^\dagger(\hat{\pi}_2),\hat{\pi}_2}(s') - V_{h+1}^{\pi_1^\dagger(\hat{\pi}_2),\pi_2^*}(s').$$

Moreover, by Player 1's best-response optimality, $V_{h+1}^{\pi_1^\dagger(\hat{\pi}_2),\hat{\pi}_2}(s') \ge V_{h+1}^{\pi_1^*,\hat{\pi}_2}(s') \ge V_{h+1}^{\pi_1^*,\pi_2^*}(s')$ (the last inequality uses Nash optimality of $\pi_2^*$ against $\pi_1^*$), so $\Delta_{h+1}(s') \ge 0$. Combined with $\Delta_{h+1}(s') \le R(s')$, this gives $0 \le \Delta_{h+1}(s') \le R(s')$, hence $R(s') \ge 0$ and $\Delta_{h+1}(s')^2 \le R(s')^2$. The right-hand side $R(s')^2$ is exactly the value-difference quantity to which Lemma B.3 applies with $\pi_1 = \pi_1^\dagger(\hat{\pi}_2)$.

The inner integrand of the second term of (26) is $\max_{a_1} \mathbb{E}_{a_2 \sim \pi_{h,2}^*} \mathbb{E}_{s' \sim P_h(\cdot|s,a_1,a_2)}[\Delta_{h+1}(s')^2]$, which is *pointwise* in $s'$. We invoke the pointwise recursion that Lemma B.3 establishes (the per-state inequality used in its proof).

**Step 4.2.a: Pointwise V-stability and rerouting to a $\Pi_{\text{uni}}$ measure.** Applying Lemma B.3's pointwise recursion at $s_{h+1} = s'$ with $\pi_1 = \pi_1^\dagger(\hat{\pi}_2)$ and unrolling from $h+1$ to $H$ yields

$$\Delta_{h+1}(s')^2 \le 2H \sum_{t=h+1}^{H} \left(1 + \frac{1}{H}\right)^{t-h-1} L_{t+1}^2 \mathbb{E}_{s_t \sim d_t^{(\pi_1^\dagger(\hat{\pi}_2),\hat{\pi}_2)\,|\,s_{h+1}=s'}} \left[\|\hat{\pi}_{t,2}(\cdot|s_t) - \pi_{t,2}^*(\cdot|s_t)\|_1^2\right], \tag{29}$$

where $d_t^{(\pi_1^\dagger(\hat{\pi}_2),\hat{\pi}_2)\,|\,s_{h+1}=s'}$ denotes the visitation distribution at step $t$ under the joint policy $(\pi_1^\dagger(\hat{\pi}_2),\hat{\pi}_2)$ conditioned on starting at $s_{h+1}=s'$. Applying Lemma B.4 (i) to this *conditional* measure (the per-step Player 1 softmax-ratio argument of Step D.1–D.4 of Appendix B.3 is pointwise and therefore extends to any conditional starting state) bridges $d_t^{(\pi_1^\dagger(\hat{\pi}_2),\hat{\pi}_2)\,|\,s'}$ to $d_t^{(\pi_1^*,\hat{\pi}_2)\,|\,s'}$ at constant cost $C_{\mathrm{trans}}$:

$$\Delta_{h+1}(s')^2 \;\le\; 2\,C_{\mathrm{trans}}\,H\sum_{t=h+1}^{H}\big(1+\tfrac{1}{H}\big)^{t-h-1}L_{t+1}^2\,\mathbb{E}_{s_t\sim d_t^{(\pi_1^*,\hat{\pi}_2)\,|\,s_{h+1}=s'}}\big[\|\hat{\pi}_{t,2}(\cdot|s_t)-\pi_{t,2}^*(\cdot|s_t)\|_1^2\big]. \tag{30}$$

**Step 4.2.b: Outer averaging produces a hybrid measure that we reroute to $\Pi_{\mathrm{uni}}$ via Lemma B.4 (ii).** Plugging (30) into the second term of (26):

$$\eta\sum_{h=1}^{H}\mathbb{E}_{s_h\sim d_h^{\pi_1^*,\hat{\pi}_2}}\Big[\max_{a_1}\mathbb{E}_{a_2\sim\pi_{h,2}^*(\cdot|s_h)}\mathbb{E}_{s'\sim P_h(\cdot|s_h,a_1,a_2)}\big[\Delta_{h+1}(s')^2\big]\Big]$$

$$\le\; 2eH\eta\sum_{h=1}^{H}\sum_{t=h+1}^{H}\big(1+\tfrac{1}{H}\big)^{t-h-1}L_{t+1}^2\,\mathbb{E}_{s_t\sim\nu'_{t,h}}\big[\|\hat{\pi}_{t,2}(\cdot|s_t)-\pi_{t,2}^*(\cdot|s_t)\|_1^2\big], \tag{31}$$

where $\nu'_{t,h}$ is the joint trajectory marginal at step $t$ obtained by composing $s_h\sim d_h^{\pi_1^*,\hat{\pi}_2}$ with the inner $\big(\max_{a_1},\mathbb{E}_{a_2\sim\pi_2^*},\mathbb{E}_{s'\sim P_h}\big)$ and then rolling out under $(\pi_1^*,\hat{\pi}_2)$ from $s_{h+1}=s'$ to step $t$. Explicitly, $\nu'_{t,h}$ is the visitation at step $t$ under the time-varying joint policy

$$\big(\pi_1^*,\hat{\pi}_2\big)\text{ for steps }1,\dots,h-1,\qquad \big(\nu_h^{\arg\max},\pi_{h,2}^*\big)\text{ at step }h,\qquad \big(\pi_1^*,\hat{\pi}_2\big)\text{ for steps }h+1,\dots,t-1,$$

with $\nu_h^{\arg\max}:\mathcal{S}\to\mathcal{A}_1$ the state-dependent argmax of the inner integrand at step $h$. Player 2 plays $\hat{\pi}_2$ at all steps except step $h$ (where it plays $\pi_{h,2}^*$), so $\nu'_{t,h}$ is *not* of the form $d_t^{\pi_1^*,\hat{\pi}_2}$ and is *not* in $\Pi_{\mathrm{uni}}$ directly.

To bring $\nu'_{t,h}$ into a $\Pi_{\mathrm{uni}}$-compatible form, define the multi-step Player 1 policy

$$\pi_1'(h) \;:=\; \big(\underbrace{\pi_{1,1}^*,\dots,\pi_{h-1,1}^*}_{\text{steps }1,\dots,h-1},\,\nu_h^{\arg\max},\,\underbrace{\pi_{h+1,1}^*,\dots,\pi_{t-1,1}^*}_{\text{steps }h+1,\dots,t-1}\big) \;\in\; \Pi_1,$$

where $\nu_h^{\arg\max}$ at step $h$ is a deterministic stage policy in $\Delta(\mathcal{A}_1)$ (a Dirac mass). Since $\Pi_1$ admits time-varying Markov policies with arbitrary stage components (including deterministic ones), $\pi_1'(h)\in\Pi_1$. The joint $(\pi_1'(h),\pi_2^*)\in\Pi_1\times\{\pi_2^*\}\subset\Pi_{\mathrm{uni}}$.

*Per-trajectory ratio computation.* The hybrid $\nu'_{t,h}$ and the target $d_t^{(\pi_1'(h),\pi_2^*)}$ share the same initial distribution $\rho$, the same transition kernels $\{P_{h'}\}_{h'<t}$, the *same* step-$h$ Player 1 action distribution ($\nu_h^{\arg\max}$), and the *same* step-$h$ Player 2 action distribution ($\pi_{h,2}^*$); they share Player 1's policy at steps $h'\neq h$ ($\pi_{h',1}^*$); they differ *only* in Player 2's policy at steps $h'\neq h$ ($\hat{\pi}_{h',2}$ vs. $\pi_{h',2}^*$). Hence the per-trajectory density ratio collapses to a Player-2 product over $\{1,\dots,t-1\}\setminus\{h\}$:

$$\frac{\nu'_{t,h}(\tau)}{d_t^{(\pi_1'(h),\pi_2^*)}(\tau)} \;=\; \prod_{h'\in\{1,\dots,t-1\}\setminus\{h\}}\frac{\hat{\pi}_{h',2}(a_{h',2}\mid s_{h'})}{\pi_{h',2}^*(a_{h',2}\mid s_{h'})}. \tag{32}$$

Each per-step factor is a softmax-vs-softmax Player 2 ratio bounded by $\exp\big(4\eta H(1+\eta^{-1}\log(1/\alpha))\big)$ (the same per-step bound used in Step D.5 of Appendix B.3, applied pointwise at every $(s_{h'},a_{h',2})$). The product has at most $t-2\le H-2$ factors. By the same Step D.4 expansion (with $\eta\le 1/(4H^2)$ controlling the $\eta$-driven part), the telescoped product satisfies

$$\prod_{h'\in\{1,\dots,t-1\}\setminus\{h\}}\frac{\hat{\pi}_{h',2}(a_{h',2}\mid s_{h'})}{\pi_{h',2}^*(a_{h',2}\mid s_{h'})} \;\le\; \exp\big(4\eta H^2(1+\eta^{-1}\log(1/\alpha))\big) \;\le\; C_{\mathrm{trans}}=e\cdot\alpha^{-4H^2}. \tag{33}$$

Crucially, the step-$h$ Player 1 ratio is identically 1 (the deterministic $\nu_h^{\arg\max}$ appears on both sides via the same state-dependent definition), so there is no $1/\alpha$ cost from the argmax-vs-softmax comparison at step $h$. By importance reweighting, for any nonnegative $f$,

$$\mathbb{E}_{s_t\sim\nu'_{t,h}}[f(s_t)] \;\le\; C_{\mathrm{trans}}\cdot\mathbb{E}_{s_t\sim d_t^{(\pi_1'(h),\pi_2^*)}}[f(s_t)]. \tag{34}$$

Applying (34) with $f(s_t) = \|\hat{\pi}_{t,2}(\cdot|s_t) - \pi_{t,2}^*(\cdot|s_t)\|_1^2$ to (31), and bounding $(1 + 1/H)^{t-h-1} \le e$, $2e \le 6$:

$$\eta \sum_{h=1}^{H} \mathbb{E}_{s_h \sim d_h^{\pi_1^*, \hat{\pi}_2}} \Big[ \max_{a_1} \mathbb{E}_{a_2 \sim \pi_{h,2}^*} \mathbb{E}_{s'} \big[ \Delta_{h+1}(s')^2 \big] \Big]$$

$$\le 6 C_{\text{trans}}^2 H \eta \sum_{h=1}^{H} \sum_{t=h+1}^{H} L_{t+1}^2 \mathbb{E}_{s_t \sim d_t^{(\pi_1'(h), \pi_2^*)}} \big[ \|\hat{\pi}_{t,2}(\cdot|s_t) - \pi_{t,2}^*(\cdot|s_t)\|_1^2 \big], \tag{35}$$

where the $C_{\text{trans}}^2$ prefactor collects the Player-1 conditional transfer (from (30)) and the Player-2 hybrid bridge (from (34)) and is absorbed into the headline $(1/\alpha)^{\mathcal{O}(H^2)}$ multiplier of Theorem 6.3. **Swap and summation.** Each inner term $L_{t+1}^2 \mathbb{E}_{s_t \sim d_t^{(\pi_1'(h), \pi_2^*)}} \big[ \|\hat{\pi}_{t,2}(\cdot|s_t) - \pi_{t,2}^*(\cdot|s_t)\|_1^2 \big]$ depends on the outer index $h$ only through the deterministic step-$h$ stage policy $\nu_h^{\text{arg max}}$ embedded in $\pi_1'(h)$ (a fixed function of $s_h$ that is determined by the algorithm and the underlying data, not by the outer integration variable). For each fixed $t$, bound the inner sum over $h$ by replacing each $\Pi_{\text{uni}}$-measure with the sup over $\Pi_{\text{uni}}$:

$$\sum_{h=1}^{t-1} \mathbb{E}_{s_t \sim d_t^{(\pi_1'(h), \pi_2^*)}} \big[ \|\hat{\pi}_{t,2} - \pi_{t,2}^*\|_1^2 \big] \le (t-1) \cdot \sup_{\pi' \in \Pi_{\text{uni}}} \mathbb{E}_{s_t \sim d_t^{\pi'}} \big[ \|\hat{\pi}_{t,2} - \pi_{t,2}^*\|_1^2 \big], \tag{36}$$

and use $(t-1) \le H$. Substituting (36) into (35) and swapping $\sum_h \sum_t = \sum_t \sum_h$:

$$\eta \sum_{h=1}^{H} \mathbb{E}_{s_h \sim d_h^{\pi_1^*, \hat{\pi}_2}} \Big[ \max_{a_1} \mathbb{E}_{a_2} \mathbb{E}_{s'} \big[ \Delta_{h+1}(s')^2 \big] \Big] \le 6 C_{\text{trans}}^2 \eta H^2 \sum_{t=1}^{H} L_{t+1}^2 \sup_{\pi' \in \Pi_{\text{uni}}} \mathbb{E}_{s_t \sim d_t^{\pi'}} \big[ \|\hat{\pi}_{t,2} - \pi_{t,2}^*\|_1^2 \big]. \tag{37}$$

The $\sup_{\pi' \in \Pi_{\text{uni}}}$ on the right side is exactly the unilateral state-marginal at step $t$ that the rest of the chain (squared stability (45) and the Bellman recursion below) operates on.

**Step 4.3: Final bound on Gap 1.** Putting (28) and (37) into (26), and noting that $(\pi_1^*, \hat{\pi}_2) \in \Pi_{\text{uni}}$ so $\mathbb{E}_{s \sim d_h^{\pi_1^*, \hat{\pi}_2}}[\cdot] \le \sup_{\pi' \in \Pi_{\text{uni}}} \mathbb{E}_{s \sim d_h^{\pi'}}[\cdot]$, we conclude

$$\text{Gap 1} \le \eta H^2 \big(1 + \eta^{-1} \log(1/\alpha)\big)^2 \sum_{h=1}^{H} \mathbb{E}_{s \sim d_h^{\pi_1^*, \hat{\pi}_2}} \big[ \|\hat{\pi}_{h,2}(\cdot|s) - \pi_{h,2}^*(\cdot|s)\|_1^2 \big]$$

$$+ 6 C_{\text{trans}}^2 \eta H^2 \sum_{t=1}^{H} L_{t+1}^2 \sup_{\pi' \in \Pi_{\text{uni}}} \mathbb{E}_{s_t \sim d_t^{\pi'}} \big[ \|\hat{\pi}_{t,2}(\cdot|s_t) - \pi_{t,2}^*(\cdot|s_t)\|_1^2 \big],$$

where the first term comes from (28) and the second term from (37). Recall $L_{t+1} := \eta^{-1} + 1 + \|V_{t+1}^{\pi_1^*, \pi_2^*}\|_\infty + \eta^{-1} \log(1/\alpha)$. By Lemma B.2, $\|V_{t+1}^{\pi_1^*, \pi_2^*}\|_\infty \le (H - t)(1 + \eta^{-1} \log(1/\alpha))$, so $L_{t+1} \le \eta^{-1} + H(1 + \eta^{-1} \log(1/\alpha))$. The bare $\eta^{-1}$ summand reflects the KL-gradient leading term in $B_{\text{reg}}$ (Lemma B.3's proof) and is absorbed—together with the $C_{\text{trans}}^2$ density-transfer cost below—into the headline $(1/\alpha)^{\mathcal{O}(H^2)}$ multiplier of Theorem 6.3. Bounding the first term's distribution $d_h^{\pi_1^*, \hat{\pi}_2}$ by $\sup_{\pi' \in \Pi_{\text{uni}}} d_h^{\pi'}$ (since $(\pi_1^*, \hat{\pi}_2) \in \Pi_{\text{uni}}$) and substituting,

$$\text{Gap 1} \le \eta H^2 \big(1 + \eta^{-1} \log(1/\alpha)\big)^2 \sum_{h=1}^{H} \sup_{\pi' \in \Pi_{\text{uni}}} \mathbb{E}_{s \sim d_h^{\pi'}} \big[ \|\hat{\pi}_{h,2}(\cdot|s) - \pi_{h,2}^*(\cdot|s)\|_1^2 \big]$$

$$+ 6 C_{\text{trans}}^2 \eta H^4 \big(1 + \eta^{-1} \log(1/\alpha)\big)^2 \sum_{t=1}^{H} \sup_{\pi' \in \Pi_{\text{uni}}} \mathbb{E}_{s_t \sim d_t^{\pi'}} \big[ \|\hat{\pi}_{t,2}(\cdot|s_t) - \pi_{t,2}^*(\cdot|s_t)\|_1^2 \big]$$

$$\le 7 C_{\text{trans}}^2 \eta H^4 \big(1 + \eta^{-1} \log(1/\alpha)\big)^2 \sum_{h=1}^{H} \sup_{\pi' \in \Pi_{\text{uni}}} \mathbb{E}_{s \sim d_h^{\pi'}} \big[ \|\hat{\pi}_{h,2}(\cdot|s) - \pi_{h,2}^*(\cdot|s)\|_1^2 \big]. \tag{38}$$

**Step 4.4: Conversion to single-player Nash-averaged squared $Q$-error via squared $L_1$ stability.** By the squared stability bound (45) (Appendix B.4), for every $(h, s)$,

$$\|\hat{\pi}_h(\cdot|s) - \pi_h^*(\cdot|s)\|_1^2 \le 4\eta^2 \cdot \xi_h^2(s),$$

with $\xi_h^2(s)$ the single-player Nash-averaged squared $Q$-error from (46). Substituting into (38),

$$
\begin{aligned}
\text{Gap 1} \;\le\; & 7\,C_{\text{trans}}^2\,\eta H^4\big(1+\eta^{-1}\log(1/\alpha)\big)^2 \cdot 4\eta^2 \sum_{h=1}^{H} \sup_{\pi'\in\Pi_{\text{uni}}} \mathbb{E}_{s\sim d_h^{\pi'}}\big[\xi_h^2(s)\big] \\
=\; & 28\,C_{\text{trans}}^2\,\eta^3 H^4\big(1+\eta^{-1}\log(1/\alpha)\big)^2 \sum_{h=1}^{H} \sup_{\pi'\in\Pi_{\text{uni}}} \mathbb{E}_{s\sim d_h^{\pi'}}\big[\xi_h^2(s)\big].
\end{aligned}
\tag{39}
$$

This proves Lemma 6.5 with the $\mathcal{O}\big(C_{\text{trans}}^2\,\eta^3 H^4(1+\eta^{-1}\log(1/\alpha))^2\big)$ leading constant; the $C_{\text{trans}}^2$ factor is absorbed into the headline $(1/\alpha)^{\mathcal{O}(H^2)}$ multiplier of Theorem 6.3.

The chain proceeds rigorously as follows: (i) the BR-style $\Delta_{h+1}^2 = (V_{h+1}^{\dagger,\hat{\pi}} - V_{h+1}^{\dagger,\pi^*})^2$ is bounded by the V-stability quantity via the $\Delta$-decomposition (Step 4.2 above); (ii) Lemma B.3's pointwise recursion (eq. (29)) applied at $s_{h+1} = s'$ with $\pi_1 = \pi_1^\dagger(\hat{\pi}_2)$ gives the per-state bound; (iii) Lemma B.4 (i) transfers the conditional BR-prefix from $h+1$ to $t$ to a Nash-prefix conditional measure (eq. (30)); (iv) the outer hybrid measure $\nu'_{t,h}$ produced by composing $s_h \sim d_h^{\pi_1^*,\hat{\pi}_2}$ with the inner $(\max_{a_1}, \mathbb{E}_{a_2\sim\pi_2^*}, \mathbb{E}_{s'\sim P_h})$ is bridged to $d_t^{(\pi_1'(h),\pi_2^*)} \in \Pi_{\text{uni}}$ via the truncated Player-2 telescoping of Lemma B.4 (ii) (eqs. (32)–(33)), at constant cost $C_{\text{trans}}$ under (5); (v) the step-$h$ Player 1 argmax cancels exactly in the per-trajectory ratio (both $\nu'_{t,h}$ and $d_t^{(\pi_1'(h),\pi_2^*)}$ use the same deterministic stage policy $\nu_h^{\text{arg max}}$ at step $h$, so no argmax-vs-softmax ratio appears anywhere in the bridge); (vi) absorbing the constant factor $C_{\text{trans}}$ and bounding each per-$h$ summand by $\sup_{\pi'\in\Pi_{\text{uni}}}$ yields the sup-form in (39).

## B.3. Proof of Lemma B.4 (Density-Ratio Stability under Small $\eta$)

**Lemma B.4** (Density-Ratio Stability under Small $\eta$). *Under the condition (5) stated with Theorem 6.3 ($\eta \le 1/(4H^2)$), let $\pi_1^\dagger(\hat{\pi}_2)$ denote the regularized best response of Player 1 against $\hat{\pi}_2$, and set $C_{\text{trans}} := e \cdot \alpha^{-4H^2}$. Then the following pointwise density-ratio bounds hold for every $h \in [H]$ and every state-action triple $(s, a_1, a_2)$:*

*(i) (Player 1 transfer)* $\dfrac{d_h^{\pi_1^\dagger(\hat{\pi}_2), \hat{\pi}_2}(s, a_1, a_2)}{d_h^{\pi_1^*, \hat{\pi}_2}(s, a_1, a_2)} \;\le\; C_{\text{trans}}.$

*(ii) (Player 2 transfer)* $\dfrac{d_h^{\pi_1^*, \hat{\pi}_2}(s, a_1, a_2)}{d_h^{\pi_1^*, \pi_2^*}(s, a_1, a_2)} \;\le\; C_{\text{trans}}.$

*Consequently, for any nonnegative function $f$ and any $h \in [H]$,*

$$
\mathbb{E}_{(s,a)\sim d_h^{\pi_1^\dagger(\hat{\pi}_2), \hat{\pi}_2}}[f] \;\le\; C_{\text{trans}} \cdot \mathbb{E}_{(s,a)\sim d_h^{\pi_1^*, \hat{\pi}_2}}[f] \;\le\; C_{\text{trans}}^2 \cdot \mathbb{E}_{(s,a)\sim d_h^{\pi_1^*, \pi_2^*}}[f].
\tag{40}
$$

The proof leverages the softmax representation of regularized best responses together with the worst-case bound $\|Q_h^{\dagger,\hat{\pi}_2} - Q_h^*\|_\infty \le 2H(1+\eta^{-1}\log(1/\alpha))$ from Lemma B.2; the condition $\eta \le 1/(4H^2)$ controls the $\eta$-driven part of the per-step softmax-ratio amplification, while the residual $\alpha$-dependent part telescopes to the multiplicative factor $\alpha^{-4H^2}$ over the horizon. Applying transfers (i) and (ii) in succession brings trajectory prefixes of the form $d_h^{\pi_1^\dagger(\hat{\pi}_2),\hat{\pi}_2}$ to the Nash-Nash form $d_h^{\pi_1^*,\pi_2^*}$, which is absorbed by Assumption 6.2 via (53) at the multiplicative cost $C_{\text{trans}}^2$. The same softmax-ratio argument also establishes (Steps D.7–D.8 below): (i', ii') the reverse directions of (i) and (ii); (iii) the Player-1 transfer $d_h^{\hat{\pi}_1,\pi_2^*}/d_h^{\pi_1^*,\pi_2^*} \le C_{\text{trans}}$ with shared Nash Player 2; (iv) the joint two-player transfer between any two $\Pi_{\text{uni}}$-policy pairs (at cost $C_{\text{trans}}^2$); and (v) the analogous conditional-measure versions for any starting state $s'$.

We bound the pointwise density ratio between the BR-prefix trajectory distribution $d_h^{\pi_1^\dagger(\hat{\pi}_2),\hat{\pi}_2}$ and the Nash-prefix trajectory distribution $d_h^{\pi_1^*,\hat{\pi}_2}$. Both distributions share the same Player 2 policy $\hat{\pi}_2$ and the same transition kernel $\{P_{h'}\}_{h'<h}$; they differ only in Player 1's policy.

**Step D.1: Per-step softmax-ratio bound.** At each step $h' \in [H]$ and each state $s$, both $\pi_1^\dagger(\hat{\pi}_2)$ and $\pi_1^*$ are KL-regularized best responses (the former to $\hat{\pi}_2$, the latter to $\pi_2^*$, which is itself the regularized Nash). By the Gibbs form (20),

$$\pi_{h',1}^\dagger(\hat{\pi}_2)(\cdot \mid s) = \text{Softmax}\big(\eta\, Q_{h',\hat{\pi}_2}^{\dagger,\hat{\pi}_2}(s,\cdot) + \log \pi_{\text{ref},h',1}(\cdot \mid s)\big), \quad \pi_{h',1}^*(\cdot \mid s) = \text{Softmax}\big(\eta\, Q_{h',\pi_2^*}^{\dagger,\pi_2^*}(s,\cdot) + \log \pi_{\text{ref},h',1}(\cdot \mid s)\big).$$

The reference-policy logit cancels in any ratio, so for every $a_1$,

$$\frac{\pi_{h',1}^\dagger(\hat{\pi}_2)(a_1 \mid s)}{\pi_{h',1}^*(a_1 \mid s)} = \frac{\exp\big(\eta\, Q_{h',\hat{\pi}_2}^{\dagger,\hat{\pi}_2}(s,a_1)\big)}{\exp\big(\eta\, Q_{h',\pi_2^*}^{\dagger,\pi_2^*}(s,a_1)\big)} \cdot \frac{Z_{h',s}^*}{Z_{h',s}^\dagger},$$

where $Z_{h',s}^\dagger$ and $Z_{h',s}^*$ are the corresponding partition functions. By a standard softmax-ratio bound, for any logits $z, z' \in \mathbb{R}^{|\mathcal{A}_1|}$,

$$\frac{\text{Softmax}(z)(a)}{\text{Softmax}(z')(a)} \leq \exp\big(\|z - z'\|_\infty + \log \tfrac{Z'}{Z}\big) \leq \exp\big(2\,\|z - z'\|_\infty\big), \tag{41}$$

where the second inequality uses $\big|\log(Z'/Z)\big| \leq \|z - z'\|_\infty$ (LSE 1-Lipschitz in $\ell_\infty$).

**Step D.2: Worst-case logit-difference bound.** The logit difference at step $h'$ is

$$\eta\,\big(Q_{h',\hat{\pi}_2}^{\dagger,\hat{\pi}_2}(s,a_1) - Q_{h',\pi_2^*}^{\dagger,\pi_2^*}(s,a_1)\big).$$

By Lemma B.2, both $Q_{h',\hat{\pi}_2}^{\dagger,\hat{\pi}_2}$ and $Q_{h',\pi_2^*}^{\dagger,\pi_2^*}$ are bounded by $H(1 + \eta^{-1}\log(1/\alpha))$ in absolute value (the value bound applies uniformly over Player 2 policies). Hence the worst-case (TV) logit-difference bound is

$$\big\|\eta\,\big(Q_{h',\hat{\pi}_2}^{\dagger,\hat{\pi}_2}(s,\cdot) - Q_{h',\pi_2^*}^{\dagger,\pi_2^*}(s,\cdot)\big)\big\|_\infty \leq 2\eta H\big(1 + \eta^{-1}\log(1/\alpha)\big).$$

Combining with (41),

$$\frac{\pi_{h',1}^\dagger(\hat{\pi}_2)(a_1 \mid s)}{\pi_{h',1}^*(a_1 \mid s)} \leq \exp\big(4\eta H\big(1 + \eta^{-1}\log(1/\alpha)\big)\big). \tag{42}$$

**Step D.3: Telescoping over the horizon.** The trajectory density at step $h$ factorizes as

$$d_h^{\pi_1,\hat{\pi}_2}(s,a_1,a_2) = \rho(s_1) \prod_{h'<h} P_{h'}(s_{h'+1} \mid s_{h'}, a_{h',1}, a_{h',2}) \prod_{h'\leq h} \pi_{h',1}(a_{h',1} \mid s_{h'}) \prod_{h'\leq h} \hat{\pi}_{h',2}(a_{h',2} \mid s_{h'}),$$

where the trajectory $(s_1, a_{1,1}, a_{1,2}, \ldots, s_h, a_{h,1}, a_{h,2}) = (s, a_1, a_2)$ at step $h$ is implicit. The ratio $d_h^{\pi_1^\dagger(\hat{\pi}_2),\hat{\pi}_2}/d_h^{\pi_1^*,\hat{\pi}_2}$ collapses to the product of per-step Player 1 policy ratios:

$$\frac{d_h^{\pi_1^\dagger(\hat{\pi}_2),\hat{\pi}_2}(s,a_1,a_2)}{d_h^{\pi_1^*,\hat{\pi}_2}(s,a_1,a_2)} = \prod_{h'=1}^{h} \frac{\pi_{h',1}^\dagger(\hat{\pi}_2)(a_{h',1} \mid s_{h'})}{\pi_{h',1}^*(a_{h',1} \mid s_{h'})} \leq \exp\big(4\eta H \cdot h \cdot \big(1 + \eta^{-1}\log(1/\alpha)\big)\big),$$

where the last inequality applies (42) to each of the $h$ factors. Since $h \leq H$,

$$\frac{d_h^{\pi_1^\dagger(\hat{\pi}_2),\hat{\pi}_2}}{d_h^{\pi_1^*,\hat{\pi}_2}} \leq \exp\big(4\eta H^2\big(1 + \eta^{-1}\log(1/\alpha)\big)\big).$$

**Step D.4: Closing the Player 1 transfer under the condition $\eta \leq 1/(4H^2)$.** Expanding the telescoped exponent in (42),

$$4\eta H^2\big(1 + \eta^{-1}\log(1/\alpha)\big) = 4\eta H^2 + 4H^2\log(1/\alpha).$$

The condition (5) bounds the first summand by 1, while the second is an unavoidable $\alpha$-dependent contribution. Hence

$$\exp\big(4\eta H^2\big(1 + \eta^{-1}\log(1/\alpha)\big)\big) \leq \exp(1) \cdot \alpha^{-4H^2} = e \cdot \alpha^{-4H^2} = C_{\text{trans}}.$$

This proves part (i) of Lemma B.4.

**Step D.5: Symmetric argument for the Player 2 transfer.** We now prove part (ii): under the same condition $\eta \leq 1/(4H^2)$,

$$\frac{d_h^{\pi_1^*, \hat{\pi}_2}(s, a_1, a_2)}{d_h^{\pi_1^*, \pi_2^*}(s, a_1, a_2)} \leq C_{\text{trans}} \qquad \text{pointwise.}$$

Both distributions share the same Player 1 policy $\pi_1^*$ and the same transition kernel; they differ only in Player 2's policy ($\hat{\pi}_2$ vs. $\pi_2^*$). The argument is fully symmetric to Steps D.1–D.4, so we only sketch the key step. Player 2's regularized stage policies admit Gibbs forms

$$\hat{\pi}_{h',2}(\cdot \mid s) = \text{Softmax}\big(-\eta \hat{Q}_{h',\hat{\pi}_1}(s, \cdot) + \log \pi_{\text{ref},h',2}(\cdot \mid s)\big), \quad \pi_{h',2}^*(\cdot \mid s) = \text{Softmax}\big(-\eta Q_{h',\pi_1^*}^*(s, \cdot) + \log \pi_{\text{ref},h',2}(\cdot \mid s)\big).$$

Both opponent-marginalized $Q$-vectors satisfy the same uniform bound $\|\hat{Q}_{h',\hat{\pi}_1}\|_\infty, \|Q_{h',\pi_1^*}^*\|_\infty \leq H(1 + \eta^{-1} \log(1/\alpha)) =: C_{\text{val}}$: Lemma B.2 gives this bound for $Q^*$ uniformly over opponent policies, and the empirical $\hat{Q}_h$ inherits the same bound by Assumption B.1. The corresponding logit difference is therefore bounded in $\ell_\infty$ by $2\eta H(1 + \eta^{-1} \log(1/\alpha))$ pointwise, and (41) gives a per-step Player 2 ratio of at most $\exp(4\eta H(1 + \eta^{-1} \log(1/\alpha)))$. Telescoping over $h \leq H$ steps and applying the same expansion as in Step D.4 yields the bound $C_{\text{trans}} = e \cdot \alpha^{-4H^2}$ for part (ii).

**Step D.6: Composed transfer** (40). For any nonnegative $f$, importance reweighting using part (i) gives

$$\mathbb{E}_{(s,a) \sim d_h^{\pi_1^\dagger(\hat{\pi}_2), \hat{\pi}_2}}\big[f(s, a)\big] = \mathbb{E}_{(s,a) \sim d_h^{\pi_1^*, \hat{\pi}_2}}\Big[f(s, a) \cdot \frac{d_h^{\pi_1^\dagger(\hat{\pi}_2), \hat{\pi}_2}(s, a)}{d_h^{\pi_1^*, \hat{\pi}_2}(s, a)}\Big] \leq C_{\text{trans}} \cdot \mathbb{E}_{(s,a) \sim d_h^{\pi_1^*, \hat{\pi}_2}}\big[f(s, a)\big],$$

and a second importance reweighting using part (ii) gives

$$\mathbb{E}_{(s,a) \sim d_h^{\pi_1^*, \hat{\pi}_2}}\big[f(s, a)\big] \leq C_{\text{trans}} \cdot \mathbb{E}_{(s,a) \sim d_h^{\pi_1^*, \pi_2^*}}\big[f(s, a)\big].$$

Composing yields the bound in (40).

**Step D.7: Reverse directions and cross-branch transfers.** The per-step softmax-ratio bound (41) and the worst-case logit-difference bound (Step D.2) depend only on the uniform $\ell_\infty$-bound $\|Q\|_\infty \leq H(1 + \eta^{-1} \log(1/\alpha))$ on the regularized $Q$-vectors that appear in the Gibbs form of any regularized stage policy. This bound holds simultaneously for $Q^{\dagger,\hat{\pi}_2}, Q^{\dagger,\pi_2^*}, \hat{Q}_{h,\hat{\pi}_1}, Q_{h,\pi_1^*}^\dagger$ (and analogously for Player 2-marginalized versions): Lemma B.2 establishes the $H(1 + \eta^{-1} \log(1/\alpha))$ bound for all regularized BR $Q$-functions of the true game ($Q_h^{\dagger,\nu}$ for any $\nu$) uniformly over opponent policies, and the empirical $\hat{Q}_h$ inherits the same bound by Assumption B.1. Consequently, the per-step softmax-ratio bound $\exp(4\eta H(1 + \eta^{-1} \log(1/\alpha)))$ holds in both directions for any pair of regularized stage policies of the same player, and similarly for any pair sharing one player's policy. Telescoping over $h \leq H$ steps and applying the Step D.4 expansion yields the following pointwise ratio bounds, each $\leq C_{\text{trans}}$:

(i′) (Reverse of (i)) $d_h^{\pi_1^*, \hat{\pi}_2}(s, a_1, a_2) / d_h^{\pi_1^\dagger(\hat{\pi}_2), \hat{\pi}_2}(s, a_1, a_2) \leq C_{\text{trans}}$.

(ii′) (Reverse of (ii)) $d_h^{\pi_1^*, \pi_2^*}(s, a_1, a_2) / d_h^{\pi_1^*, \hat{\pi}_2}(s, a_1, a_2) \leq C_{\text{trans}}$.

(iii) (Player 1 transfer with $\pi_2^*$ shared) $d_h^{\hat{\pi}_1, \pi_2^*}(s, a_1, a_2) / d_h^{\pi_1^*, \pi_2^*}(s, a_1, a_2) \leq C_{\text{trans}}$ and its reverse.

(iv) (*Cross-branch trajectory transferred to a* $\Pi_{\text{uni}}$ *proxy*) A trajectory generated by a *cross-branch* policy schedule — one in which Player 1 plays Nash on some segments and Player 2 plays Nash on the complementary segments — is *not* natively a single multi-step policy in $\Pi_{\text{uni}}$. However, picking a pure $\Pi_{\text{uni}}$ proxy policy (e.g., Player 2 at Nash throughout) and comparing the cross-branch trajectory marginal to the proxy's marginal step-by-step, the two differ in at most *both* players' stage policies. Applying the per-step softmax-ratio bound (42) to each player's stage cumulatively bounds the resulting ratio by the product of the two single-player cumulative transfers, each $\leq C_{\text{trans}}$ by (i)–(iii) above:

$$\frac{d_h^{\text{cross-branch}}(s, a_1, a_2)}{d_h^{\text{proxy} \in \Pi_{\text{uni}}}(s, a_1, a_2)} \leq C_{\text{trans}}^2 = e^2 \cdot \alpha^{-8H^2}.$$

This allows us to shift expectations from the cross-branch trajectory measure into the $\Pi_{\text{uni}}$-compatible concentrability class of Assumption 6.2 at a multiplicative cost $C_{\text{trans}}^2$ absorbed into the prefactor of (62).

If only one player's policy differs between the source trajectory and the proxy (the standard cases (i)–(iii) above, where both source and proxy already lie in $\Pi_{\mathrm{uni}}$), the bound improves to $C_{\mathrm{trans}}$; the bound $C_{\mathrm{trans}}^2$ in (iv) is the price paid for absorbing cross-branch trajectories into a single proxy.

**Step D.8: Pointwise nature extends to conditional measures (part (v)).** The per-step bound (42) and its analogues are pointwise in $(s_{h'}, a_{h',1}, a_{h',2})$ — they hold uniformly at every state-action triple. Consequently, telescoping the per-step ratios across any sub-horizon $\{h' + 1, \ldots, h\}$ produces the same $C_{\mathrm{trans}}$-bound (or $C_{\mathrm{trans}}^2$ for joint two-player transfers, as in (iv)) regardless of the starting-state distribution. Formally:

(v) (Conditional-measure version of (i)–(iv)) For any state $s' \in \mathcal{S}$ and any $h' < t \le H$, the conditional measure $d_t^{(\pi_1, \pi_2) \mid s_{h'+1} = s'}$ satisfies the same ratio bounds (i)–(iv) above. This conditional version is invoked in the pointwise V-stability step of Lemma 6.5 (eqs. (29)–(30)).

$\square$

## B.4. Proof of Lemma 6.4

**Step 5: Local VI Stability.** The proof proceeds by studying a *local variational inequality (VI)* at each time–state pair $(h, s)$. We emphasize that the local VI analysis below is conducted with the continuation value fixed at each $(h, s)$, so that the resulting operator is purely stagewise.

**Local VI operator.** Fix $(h, s) \in [H] \times \mathcal{S}$ and let $\pi_{h,s} = (\pi_{h,s,1}, \pi_{h,s,2}) \in \Delta(\mathcal{A}_1) \times \Delta(\mathcal{A}_2)$ denote a joint action distribution. Given any payoff matrix $G \in \mathbb{R}^{|\mathcal{A}_1| \times |\mathcal{A}_2|}$, define the stagewise KL-regularized objective

$$J_{G,h,s}(\pi_{h,s,1}, \pi_{h,s,2}) := \mathbb{E}_{a \sim \pi_{h,s}}\big[G(a_1, a_2)\big] - \eta^{-1}\mathrm{KL}(\pi_{h,s,1} \,\|\, \pi_{\mathrm{ref},h,1}(\cdot \mid s)) + \eta^{-1}\mathrm{KL}(\pi_{h,s,2} \,\|\, \pi_{\mathrm{ref},h,2}(\cdot \mid s)).$$

We define the associated local VI operator (the gradient map of the regularized zero-sum game) by

$$F_{G,h,s}(\pi_{h,s}) := \begin{pmatrix} -\nabla_{\pi_1} J_{G,h,s}(\pi_{h,s}) \\ \nabla_{\pi_2} J_{G,h,s}(\pi_{h,s}) \end{pmatrix} = M_G \, \pi_{h,s} + \eta^{-1}\nabla\mathrm{Reg}_{h,s}(\pi_{h,s}),$$

where $M_G$ is the skew-symmetric matrix composed of blocks $-G$ and $G^\top$, and $\mathrm{Reg}_{h,s}$ collects the KL regularization terms.

Define the continuation-augmented payoff matrix for the true game and the estimated game respectively:

$$G_{h,s}^*(a_1, a_2) := Q_h^{\pi^*, \pi^*}(s, a_1, a_2),$$
$$\hat{G}_{h,s}(a_1, a_2) := \hat{Q}_h(s, a_1, a_2).$$

By the stagewise first-order optimality conditions, the true equilibrium $\pi_{h,s}^*$ is a root of $F_{G_{h,s}^*}$, and the learned policy $\hat{\pi}_{h,s}$ is a root of $F_{\hat{G}_{h,s}}$:

$$F_{G_{h,s}^*, h, s}(\pi_{h,s}^*) = 0, \qquad F_{\hat{G}_{h,s}, h, s}(\hat{\pi}_{h,s}) = 0.$$

**Strong Monotonicity and Stability.** The KL-regularized operator satisfies a strong monotonicity property with respect to the $\ell_1$-norm. Specifically, for any $G$ and any $\pi, \pi'$:

$$\big\langle F_{G,h,s}(\pi) - F_{G,h,s}(\pi'), \, \pi - \pi' \big\rangle \ge \frac{\eta^{-1}}{2} \|\pi - \pi'\|_1^2.$$

Applying this to $\pi = \hat{\pi}_{h,s}$ and $\pi' = \pi_{h,s}^*$ with the true operator $F_{G^*}$:

$$\frac{\eta^{-1}}{2}\|\hat{\pi}_{h,s} - \pi_{h,s}^*\|_1^2 \le \big\langle F_{G_{h,s}^*, h, s}(\hat{\pi}_{h,s}) - F_{G_{h,s}^*, h, s}(\pi_{h,s}^*), \, \hat{\pi}_{h,s} - \pi_{h,s}^* \big\rangle$$
$$= \big\langle F_{G_{h,s}^*, h, s}(\hat{\pi}_{h,s}) - F_{\hat{G}_{h,s}, h, s}(\hat{\pi}_{h,s}), \, \hat{\pi}_{h,s} - \pi_{h,s}^* \big\rangle.$$

The difference in operators is linear in the difference of payoff matrices. Let $\Delta M_{h,s} = M_{G^*} - M_{\hat{G}}$. Since $\Delta M_{h,s}$ is skew-symmetric, $\langle \Delta M_{h,s} v, v \rangle = 0$. Thus:

$$\text{RHS} = \langle \Delta M_{h,s} \hat{\pi}_{h,s}, \hat{\pi}_{h,s} - \pi^*_{h,s} \rangle = \langle \Delta M_{h,s} \pi^*_{h,s}, \hat{\pi}_{h,s} - \pi^*_{h,s} \rangle.$$

Let $\mathcal{E}_{h,s} := \|\Delta M_{h,s} \pi^*_{h,s}\|_\infty$. We *refine* the standard estimate $\mathcal{E}_{h,s} \leq \|\hat{Q}_h(s) - Q^*_h(s)\|_\infty$ (joint-action sup-norm) by exploiting the explicit block structure of $\Delta M_{h,s} \pi^*_{h,s}$. Recall that $M_G$ has the skew-symmetric block form $\begin{pmatrix} 0 & -G \\ G^\top & 0 \end{pmatrix}$, so $\Delta M_{h,s} \pi^*_{h,s} = \begin{pmatrix} -\Delta G_{h,s}\, \pi^*_{h,s,2} \\ \Delta G^\top_{h,s}\, \pi^*_{h,s,1} \end{pmatrix}$, where $\Delta G_{h,s}(a_1, a_2) = \hat{Q}_h(s, a_1, a_2) - Q^*_h(s, a_1, a_2)$. The Player-1 block (indexed by $a_1$) has component $-\mathbb{E}_{a_2 \sim \pi^*_{h,s,2}}[\Delta G_{h,s}(a_1, a_2)]$, and the Player-2 block (indexed by $a_2$) has component $\mathbb{E}_{a_1 \sim \pi^*_{h,s,1}}[\Delta G_{h,s}(a_1, a_2)]$. Hence

$$\mathcal{E}_{h,s} = \max\Big( \max_{a_1} \big|\mathbb{E}_{a_2 \sim \pi^*_{h,s,2}}[\hat{Q}_h(s, a_1, a_2) - Q^*_h(s, a_1, a_2)]\big|, \ \max_{a_2} \big|\mathbb{E}_{a_1 \sim \pi^*_{h,s,1}}[\hat{Q}_h(s, a_1, a_2) - Q^*_h(s, a_1, a_2)]\big| \Big). \tag{43}$$

Each branch in the $\max$ is a Player-$i$ deterministic deviation paired with the opponent at Nash, a structure that lies in $\Pi_{\text{uni}}$ when the outer state expectation is taken.

Using Hölder's inequality with (43),

$$\frac{\eta^{-1}}{2} \|\hat{\pi}_{h,s} - \pi^*_{h,s}\|_1^2 \leq \mathcal{E}_{h,s} \|\hat{\pi}_{h,s} - \pi^*_{h,s}\|_1,$$

and dividing by the norm yields the pointwise stability bound

$$\|\hat{\pi}_{h,s} - \pi^*_{h,s}\|_1 \leq 2\eta\, \mathcal{E}_{h,s}. \tag{44}$$

Squaring (44) and applying $(\max(x,y))^2 \leq x^2 + y^2$ together with Jensen's inequality on each branch of (43) yields the *squared stability bound*

$$\|\hat{\pi}_{h,s} - \pi^*_{h,s}\|_1^2 \leq 4\eta^2 \cdot \xi_h^2(s), \tag{45}$$

where the *single-player Nash-averaged squared Q-error*

$$\xi_h^2(s) := \max\Big( \max_{a_1} \mathbb{E}_{a_2 \sim \pi^*_{h,2}(\cdot|s)}\big[(\hat{Q}_h - Q^*_h)^2(s, a_1, a_2)\big], \ \max_{a_2} \mathbb{E}_{a_1 \sim \pi^*_{h,1}(\cdot|s)}\big[(\hat{Q}_h - Q^*_h)^2(s, a_1, a_2)\big] \Big) \tag{46}$$

is bounded above by $\|\hat{Q}_h(s) - Q^*_h(s)\|_\infty^2$ but lies in $\Pi_{\text{uni}}$-form when the outer state expectation is taken: each branch of the $\max$ corresponds to the worst Player-$i$ deterministic deviation paired with the opponent's Nash policy, so $\xi_h^2(s)$ is exactly the quantity controlled by Assumption 6.2 via (53).

## B.5. Proof of Lemma 6.6 (Evaluation Gap via Squared Error)

**Bounding Gap 2 via Logits Decomposition.** Recall the exact identity established in Step 1 (Eq. (19)):

$$\text{Gap 2} = \eta^{-1} \sum_{h=1}^H \mathbb{E}_{s \sim d_h^{\pi^*_1, \hat{\pi}_2}} \big[ \text{KL}\big(\hat{\pi}_{h,2}(\cdot|s) \,\big\|\, \pi^*_{h,2}(\cdot|s)\big) \big]. \tag{47}$$

The Player-2 stage policies are the regularized best responses in the empirical and true games, respectively. By the saddle-point first-order condition with payoff $Q$ and Player-1 marginalization,

$$\hat{\pi}_{h,2}(a_2|s) \propto \pi^{\text{ref}}_{h,2}(a_2|s) \exp\big(-\eta\, \hat{Q}_{h,\hat{\pi}_1}(s, a_2)\big), \quad \pi^*_{h,2}(a_2|s) \propto \pi^{\text{ref}}_{h,2}(a_2|s) \exp\big(-\eta\, Q^\dagger_{h,\pi^*_1}(s, a_2)\big),$$

where $\hat{Q}_{h,\hat{\pi}_1}(s, a_2) := \mathbb{E}_{a_1 \sim \hat{\pi}_{h,1}}[\hat{Q}_h(s, a_1, a_2)]$ and $Q^\dagger_{h,\pi^*_1}(s, a_2) := \mathbb{E}_{a_1 \sim \pi^*_{h,1}}[Q^*_h(s, a_1, a_2)]$. The corresponding logits are

$$\hat{z}_h(a_2) := -\eta\, \hat{Q}_{h,\hat{\pi}_1}(s, a_2) + \log \pi^{\text{ref}}_{h,2}(a_2|s), \quad z^*_h(a_2) := -\eta\, Q^\dagger_{h,\pi^*_1}(s, a_2) + \log \pi^{\text{ref}}_{h,2}(a_2|s).$$

By the smoothness bound (24),

$$\text{KL}\big(\hat{\pi}_{h,2}(\cdot|s) \,\|\, \pi^*_{h,2}(\cdot|s)\big) \leq \tfrac{1}{2} \|\hat{z}_h - z^*_h\|_\infty^2.$$

Decomposing the logit difference (reference-policy terms cancel),

$$\hat{z}_h(a_2) - z^*_h(a_2) = -\eta \cdot \mathbb{E}_{a_1 \sim \hat{\pi}_{h,1}}\big[\hat{Q}_h(s, a_1, a_2) - Q^*_h(s, a_1, a_2)\big] \quad \text{(Estimation term)}$$

$$\qquad\qquad - \eta \cdot \mathbb{E}_{a_1 \sim (\hat{\pi}_{h,1} - \pi^*_{h,1})}\big[Q^*_h(s, a_1, a_2)\big] \quad \text{(Policy deviation term)}. \tag{48}$$

**Per-state Player-1 action-transfer step (one-step instance of Lemma B.4).** For every fixed $(h,s)$, both $\hat{\pi}_{h,1}(\cdot|s)$ and $\pi^*_{h,1}(\cdot|s)$ are Gibbs softmax distributions with logits bounded in $\ell_\infty$ by $\eta \cdot 2\|Q\|_\infty \le 2\eta C_{\mathrm{val}} = 2\eta H\lambda$ (Lemma B.2), where $\lambda := 1 + \eta^{-1}\log(1/\alpha)$. By the per-step softmax-ratio bound (41) (i.e., Steps D.1–D.2 of Appendix B.3 applied at a single state-step), the per-state action ratio satisfies

$$\frac{\hat{\pi}_{h,1}(a_1|s)}{\pi^*_{h,1}(a_1|s)} \le \exp(4\eta H\lambda) = \exp(4\eta H) \cdot \alpha^{-4H} \le e \cdot \alpha^{-4H} \le C_{\mathrm{trans}} \qquad \text{under (5)},$$

since $\eta \le 1/(4H^2)$ implies $4\eta H \le 1/H \le 1$. Consequently, for any nonnegative function $g(a_1, a_2)$ on $\mathcal{A}_1 \times \mathcal{A}_2$,

$$\mathbb{E}_{a_1 \sim \hat{\pi}_{h,1}(\cdot|s)}[g(a_1, a_2)] \le C_{\mathrm{trans}} \cdot \mathbb{E}_{a_1 \sim \pi^*_{h,1}(\cdot|s)}[g(a_1, a_2)]. \tag{49}$$

**Bounding the estimation term via Jensen + per-state transfer (direct to $\xi^2$).** Applying Jensen to the squared estimation term, then the per-state transfer (49) with $g(a_1, a_2) = (\hat{Q}_h - Q^*_h)^2(s, a_1, a_2)$,

$$\sup_{a_2} \left(\mathbb{E}_{a_1 \sim \hat{\pi}_{h,1}}[\hat{Q}_h - Q^*_h](s, a_1, a_2)\right)^2 \le \sup_{a_2} \mathbb{E}_{a_1 \sim \hat{\pi}_{h,1}}\left[(\hat{Q}_h - Q^*_h)^2(s, a_1, a_2)\right]$$

$$\le C_{\mathrm{trans}} \cdot \sup_{a_2} \mathbb{E}_{a_1 \sim \pi^*_{h,1}}\left[(\hat{Q}_h - Q^*_h)^2(s, a_1, a_2)\right]$$

$$\le C_{\mathrm{trans}} \cdot \xi^2_h(s), \tag{50}$$

where the last inequality uses the definition of $\xi^2_h(s)$ (46) (specifically the $i = 2$ branch: $\sup_{a_2} \mathbb{E}_{a_1 \sim \pi^*_{h,1}}[(\hat{Q} - Q^*)^2]$).

**Bounding the policy-deviation term via squared stability.** For the policy-deviation term, Hölder's inequality with $\|Q^*_h\|_\infty \le H(1 + \eta^{-1}\log(1/\alpha)) =: L$ (Lemma B.2) gives

$$\sup_{a_2} \left(\mathbb{E}_{a_1 \sim (\hat{\pi}_{h,1} - \pi^*_{h,1})}[Q^*_h(s, a_1, a_2)]\right)^2 \le L^2 \cdot \|\hat{\pi}_{h,1}(\cdot|s) - \pi^*_{h,1}(\cdot|s)\|^2_1.$$

Applying the squared stability bound (45) directly to the policy difference,

$$\sup_{a_2} \left(\mathbb{E}_{a_1 \sim (\hat{\pi}_{h,1} - \pi^*_{h,1})}[Q^*_h(s, a_1, a_2)]\right)^2 \le L^2 \cdot 4\eta^2 \xi^2_h(s) = 4\eta^2 L^2 \xi^2_h(s). \tag{51}$$

**Combining.** By $(a + b)^2 \le 2a^2 + 2b^2$ applied to the logit decomposition (48), then (50) and (51),

$$\|\hat{z}_h - z^*_h\|^2_\infty \le 2\eta^2 \cdot C_{\mathrm{trans}} \cdot \xi^2_h(s) + 2\eta^2 \cdot 4\eta^2 L^2 \cdot \xi^2_h(s) = \left(2C_{\mathrm{trans}}\,\eta^2 + 8\eta^4 L^2\right)\xi^2_h(s).$$

Under (5) we have $\eta^2 H^2 \le 1/16$, so $8\eta^4 L^2 = 8\eta^4 H^2\lambda^2 \le \eta^2\lambda^2/2$. Hence

$$\mathrm{KL}\left(\hat{\pi}_{h,2}(\cdot|s)\,\|\,\pi^*_{h,2}(\cdot|s)\right) \le \tfrac{1}{2}\|\hat{z}_h - z^*_h\|^2_\infty \le \left(C_{\mathrm{trans}} + \tfrac{1}{4}\lambda^2\right)\eta^2\,\xi^2_h(s).$$

Plugging into (47),

$$\mathrm{Gap}\,2 \le \mathcal{O}\left(\eta\,(C_{\mathrm{trans}} + \lambda^2)\right)\sum_{h=1}^{H}\mathbb{E}_{s \sim d_h^{\pi^*_1, \hat{\pi}_2}}\left[\xi^2_h(s)\right]. \tag{52}$$

This proves Lemma 6.6, with the $\mathcal{O}(\eta)$ leading constant.

### B.6. Proof of Theorem 6.3 (Main Statistical Bound)

We start from the bounds on Gap 1 and Gap 2 derived in Lemma 6.5 (Eq. (39)) and Lemma 6.6 (Eq. (52)).

**Unilateral Deviation and Concentrability Definitions.** In the analyses of Gap 1 and Gap 2, we encounter trajectory distributions in which one player follows the Nash equilibrium strategy $\pi^*$ throughout while the other follows a deviation policy. The set of *Unilateral Deviation Policies* (Assumption 6.2) is

$$\Pi_{\mathrm{uni}} := \big(\{\pi_1^*\} \times \Pi_2\big) \cup \big(\Pi_1 \times \{\pi_2^*\}\big).$$

Since all policies in the analysis (NE, best responses, $\hat{\pi}$, SOS-MD iterates) are Gibbs-form and hence support-restricted to $\mathrm{supp}(\pi^{\mathrm{ref}})$, we interpret $\Pi_i$ in Assumption 6.2 and in the max over actions in $\xi_h^2$ (6) as the support-restricted class $\Pi_i^{\mathrm{supp}} := \{\pi_i : \mathrm{supp}(\pi_{h,i}(\cdot|s)) \subseteq \mathrm{supp}(\pi_{h,i}^{\mathrm{ref}}(\cdot|s))\}$. The distributions encountered in our analysis (e.g., $d_h^{\pi_1^*,\hat{\pi}_2}$ and $d_h^{\hat{\pi}_1,\pi_2^*}$) belong to the set of visitation distributions induced by policies in $\Pi_{\mathrm{uni}}$. To handle the distribution shift between the offline dataset $\mathcal{D}$ (generated by $\mu$) and these unilateral deviation distributions, we use the unilateral $D^2$-divergence concentrability coefficient $C_{\mathrm{uni}}$ from Assumption 6.2: for every $h \in [H]$, every $i \in \{1, 2\}$, and every $\pi_i \in \Pi_i$,

$$\mathbb{E}_{(s,a_1,a_2) \sim d_h^{\pi_i \times \pi_{-i}^*}}\big[D_{\mathcal{Q}}^2\big((s, a_1, a_2); \mu_h\big)\big] \leq C_{\mathrm{uni}}.$$

As a direct consequence of Definition 6.1 and Assumption 6.2, for any unilateral policy $\pi' \in \Pi_{\mathrm{uni}}$ of the form $(\pi_i, \pi_{-i}^*)$ and any $Q_1, Q_2 \in \mathcal{Q}$,

$$\mathbb{E}_{(s,a_1,a_2) \sim d_h^{\pi'}}\big[(Q_1 - Q_2)^2\big] \leq \mathbb{E}_{(s,a_1,a_2) \sim d_h^{\pi'}}\big[D_{\mathcal{Q}}^2\big] \cdot \mathbb{E}_{\mu_h}\big[(Q_1 - Q_2)^2\big] \leq C_{\mathrm{uni}} \cdot \mathbb{E}_{\mu_h}\big[(Q_1 - Q_2)^2\big]. \tag{53}$$

This function-class-aware $D^2$ form absorbs both the deterministic "worst-action" deviation by one player at each step (a max-over-actions structure that arises naturally in the squared-error decomposition of Lemma 6.5) and the trajectory-distribution shift between $d_h^{\pi'}$ and $\mu_h$, in a single direct inequality, replacing the standard density-ratio assumption $\|\mathrm{d}\, d_h^{\pi'}/\mathrm{d}\, \mu_h\|_\infty \leq C_{\mathrm{uni}}$ used in prior offline Markov-game work (Cui & Du, 2022a).

**Combining Gap 1 and Gap 2 into a single single-player Nash-averaged $Q$-error sum.** By Lemmas 6.6 and 6.5, both Gap 1 and Gap 2 are bounded by the same cumulative single-player Nash-averaged squared $Q$-estimation error $\xi_h^2(s)$ from (46), but with different $\eta$-scalings: Lemma 6.6 gives Gap 2 $\leq \mathcal{O}(\eta(C_{\mathrm{trans}} + \lambda^2)) \sum_h \mathbb{E}[\xi_h^2]$ (linear in $\eta$), while Lemma 6.5 gives Gap 1 $\leq \mathcal{O}(C_{\mathrm{trans}}^2 \eta^3 H^4 \lambda^2) \sum_h \mathbb{E}[\xi_h^2]$ (cubic in $\eta$, with $\lambda := 1 + \eta^{-1}\log(1/\alpha)$). Combining,

$$\mathrm{Gap}(\hat{\pi}) = \mathrm{Gap}\,1 + \mathrm{Gap}\,2 \leq \mathcal{O}\big(\eta\,(C_{\mathrm{trans}} + \lambda^2) + C_{\mathrm{trans}}^2 \eta^3 H^4 \lambda^2\big) \sum_{h=1}^{H} \mathbb{E}_{s \sim d_h^{\mathrm{uni}}}\big[\xi_h^2(s)\big], \tag{54}$$

where $d_h^{\mathrm{uni}}$ denotes the unilateral state-distribution induced by either $(\pi^*, \hat{\pi})$ or $(\hat{\pi}, \pi^*)$, both of which lie in $\Pi_{\mathrm{uni}}$. The single-player Nash-averaged form of $\xi_h^2$ ensures that $\sum_h \mathbb{E}_{d_h^{\mathrm{uni}}}[\xi_h^2(s)]$ is directly compatible with the unilateral $D^2$-divergence concentrability of Assumption 6.2 (after the Player-2 prefix transfer of Lemma B.4); see the Bellman recursion below for the conversion to a Bellman-regression-error sum controlled by least-squares concentration.

The Bellman recursion below bounds $\sum_h \mathbb{E}_{d_h^{\mathrm{uni}}}[\xi_h^2(s)] \lesssim C_{\mathrm{trans}} H^6 C_{\mathrm{uni}} \log(|\mathcal{Q}|/\delta)/n$ (the extra $C_{\mathrm{trans}}$ coming from the Player-2 prefix transfer in that recursion; the extra $H$ relative to the standard $H^5$ FQI bound comes from the union bound over the algorithm-induced value class $\mathcal{V}_{h+1}^{\mathrm{alg}}$ in Lemma B.7); the saddle-point V-non-expansion of Lemma B.5 ensures that the propagation chain operates on $\Pi_{\mathrm{uni}}$-compatible Nash-averaged squared errors at every step. Combined with the prefactor above, this yields the final bound $\widetilde{\mathcal{O}}\big((\eta H^6 + \eta^3 H^{10}) C_{\mathrm{uni}} \log(|\mathcal{Q}|/\delta)/n \cdot (1/\alpha)^{\mathcal{O}(H^2)}\big)$, where the $(1/\alpha)^{\mathcal{O}(H^2)}$ factor absorbs $C_{\mathrm{trans}}$, $C_{\mathrm{trans}}^2$, and $\lambda^2$.

**Final Bound via Recursive Bellman Error Unrolling.** We start from the combined bound (54), which expresses the total duality gap directly in terms of the cumulative single-player Nash-averaged squared $Q$-error $\sum_h \mathbb{E}_{s \sim d_h^{\mathrm{uni}}}[\xi_h^2(s)]$ from (46). We now control this error via Bellman unrolling and unilateral $D^2$-divergence concentrability.

Let $\Delta Q_h(s) := \|\hat{Q}_h(s, \cdot, \cdot) - Q_h^*(s, \cdot, \cdot)\|_\infty$ denote the maximal joint-action $Q$-difference at state $s$; by definition $\xi_h^2(s) \leq \Delta Q_h(s)^2$ pointwise.

**Step 1: Recursive Unrolling via Bellman Error.** Instead of decomposing into reward errors (which applies to model-based learning), we decompose the Q-value difference using the Bellman optimality operator. Let $\mathcal{T}_h$ be the operator such that

$(\mathcal{T}_h f)(s,a) = r_h^*(s,a) + \mathbb{E}_{s' \sim P_h(\cdot|s,a)}[f(s')]$. Note that $Q_h^* = \mathcal{T}_h V_{h+1}^*$. We define the pointwise *Bellman regression error* against the estimated next-step value $\hat{V}_{h+1}$ as:

$$\mathcal{E}_h(s,a) := |\hat{Q}_h(s,a) - (\mathcal{T}_h \hat{V}_{h+1})(s,a)|.$$

Using the triangle inequality, we can bound the total estimation error:

$$|\hat{Q}_h(s,a) - Q_h^*(s,a)| = |\hat{Q}_h(s,a) - \mathcal{T}_h \hat{V}_{h+1}(s,a) + \mathcal{T}_h \hat{V}_{h+1}(s,a) - \mathcal{T}_h V_{h+1}^*(s,a)|$$

$$\leq \underbrace{|\hat{Q}_h(s,a) - (\mathcal{T}_h \hat{V}_{h+1})(s,a)|}_{\text{Regression Error } \mathcal{E}_h(s,a)} + \underbrace{|(\mathcal{T}_h \hat{V}_{h+1})(s,a) - (\mathcal{T}_h V_{h+1}^*)(s,a)|}_{\text{Propagation Error}}.$$

The propagation term is bounded by the expected value difference:

$$|(\mathcal{T}_h \hat{V}_{h+1})(s,a) - (\mathcal{T}_h V_{h+1}^*)(s,a)| = |\mathbb{E}_{s' \sim P_h(\cdot|s,a)}[\hat{V}_{h+1}(s') - V_{h+1}^*(s')]|$$

$$\leq \mathbb{E}_{s' \sim P_h(\cdot|s,a)}[|\hat{V}_{h+1}(s') - V_{h+1}^*(s')|].$$

Next, we establish a non-expansion bound for the value function that leverages the saddle-point structure of the regularized minimax operator. Define

$$W_{h+1}^2(s) := \max\Big\{ \mathbb{E}_{(a_1,a_2) \sim \pi_1^* \times \hat{\pi}_2}[(\hat{Q}_{h+1} - Q_{h+1}^*)^2(s,a_1,a_2)], \; \mathbb{E}_{(a_1,a_2) \sim \hat{\pi}_1 \times \pi_2^*}[(\hat{Q}_{h+1} - Q_{h+1}^*)^2(s,a_1,a_2)] \Big\}. \quad (55)$$

Each term in the max has *exactly one* player at Nash, with the other player at the corresponding learned policy; both joint distributions therefore lie in $\Pi_{\text{uni}}$.

**Lemma B.5** (Saddle-point V-non-expansion). *For every $s \in \mathcal{S}$ and $h \in [H]$,*

$$\big(\hat{V}_{h+1}(s) - V_{h+1}^*(s)\big)^2 \leq W_{h+1}^2(s). \quad (56)$$

*Proof.* Let $J_s(\pi_1, \pi_2; Q) := \mathbb{E}_{a \sim \pi_1 \times \pi_2}[Q(s,a)] - \eta^{-1}\mathrm{KL}(\pi_1 \| \pi_1^{\mathrm{ref}}) + \eta^{-1}\mathrm{KL}(\pi_2 \| \pi_2^{\mathrm{ref}})$ denote the stage-game objective. The regularized minimax equilibrium $(\hat{\pi}_1, \hat{\pi}_2)$ for $\hat{Q}$ and $(\pi_1^*, \pi_2^*)$ for $Q^*$ satisfy the saddle-point inequalities

$$J_s(\pi_1, \hat{\pi}_2; \hat{Q}) \leq J_s(\hat{\pi}_1, \hat{\pi}_2; \hat{Q}) \leq J_s(\hat{\pi}_1, \pi_2; \hat{Q}), \quad J_s(\pi_1, \pi_2^*; Q^*) \leq J_s(\pi_1^*, \pi_2^*; Q^*) \leq J_s(\pi_1^*, \pi_2; Q^*),$$

for all $(\pi_1, \pi_2)$. Setting $\pi_1 = \pi_1^*$ in the first chain and $\pi_2 = \hat{\pi}_2$ in the second, and noting $\hat{V}_{h+1}(s) = J_s(\hat{\pi}_1, \hat{\pi}_2; \hat{Q})$, $V_{h+1}^*(s) = J_s(\pi_1^*, \pi_2^*; Q^*)$, we obtain

$$\hat{V}_{h+1}(s) - V_{h+1}^*(s) \geq J_s(\pi_1^*, \hat{\pi}_2; \hat{Q}) - J_s(\pi_1^*, \hat{\pi}_2; Q^*) = \mathbb{E}_{(a_1,a_2) \sim \pi_1^* \times \hat{\pi}_2}[\hat{Q}_{h+1}(s,a_1,a_2) - Q_{h+1}^*(s,a_1,a_2)],$$

where the regularization terms cancel because the policy pair $(\pi_1^*, \hat{\pi}_2)$ is identical on both sides. Symmetrically, taking $\pi_1 = \hat{\pi}_1$ and $\pi_2 = \pi_2^*$ yields

$$\hat{V}_{h+1}(s) - V_{h+1}^*(s) \leq J_s(\hat{\pi}_1, \pi_2^*; \hat{Q}) - J_s(\hat{\pi}_1, \pi_2^*; Q^*) = \mathbb{E}_{(a_1,a_2) \sim \hat{\pi}_1 \times \pi_2^*}[\hat{Q}_{h+1}(s,a_1,a_2) - Q_{h+1}^*(s,a_1,a_2)].$$

Combining the two and squaring,

$$\big(\hat{V}_{h+1}(s) - V_{h+1}^*(s)\big)^2 \leq \max\Big(\big(\mathbb{E}_{\pi_1^* \times \hat{\pi}_2}[\hat{Q} - Q^*]\big)^2, \; \big(\mathbb{E}_{\hat{\pi}_1 \times \pi_2^*}[\hat{Q} - Q^*]\big)^2\Big).$$

Jensen's inequality applied to each term gives the bound (56). $\qquad\square$

Substituting Lemma B.5 into the propagation-error bound and applying $(a+b)^2 \leq 2a^2 + 2b^2$ together with Jensen's inequality $(\mathbb{E}_{s'}[|\hat{V} - V^*|])^2 \leq \mathbb{E}_{s'}[(\hat{V} - V^*)^2] \leq \mathbb{E}_{s'}[W_{h+1}^2(s')]$, we obtain the pointwise squared inequality

$$(\hat{Q}_h - Q_h^*)^2(s,a_1,a_2) \leq 2\mathcal{E}_h^2(s,a_1,a_2) + 2\mathbb{E}_{s' \sim P_h(\cdot|s,a_1,a_2)}[W_{h+1}^2(s')], \quad (57)$$

which holds for every $(s, a_1, a_2)$. To obtain a state-only bound that is compatible with the unilateral $D^2$-divergence concentrability of Assumption 6.2, we apply $\max_{a_i} \mathbb{E}_{a_{-i} \sim \pi_{h,-i}^*(\cdot|s)}[\cdot]$ to both sides of (57) (for each $i \in \{1, 2\}$) and take the max over $i$. This yields, for the single-player Nash-averaged squared error $\xi_h^2(s)$ defined in (46),

$$\xi_h^2(s) \leq 2\mathcal{Z}_{h,s}^2 + 2\max_{i \in \{1,2\}} \sup_{\nu_i \in \Pi_i} \mathbb{E}_{(a_1,a_2) \sim \nu_i \times \pi_{h,-i}^*}[\mathbb{E}_{s' \sim P_h(\cdot|s,a_1,a_2)}[W_{h+1}^2(s')]], \quad (58)$$

where $\mathcal{Z}_{h,s}^2 := \max_i \sup_{\nu_i \in \Pi_i} \mathbb{E}_{\nu_i \times \pi_{h,-i}^*}[\mathcal{E}_h^2(s, a_1, a_2)]$ is the single-player Nash-averaged squared Bellman regression error (defined formally in (61) below). Each branch of the $\max$ in (58) is over a single-player deterministic deviation $\nu_i$ paired with the opponent at Nash $\pi_{h,-i}^*$, so the joint distribution $\nu_i \times \pi_{h,-i}^*$ at step $h$ lies in $\Pi_{\text{uni}}$.

Taking the joint-action sup of (57) and defining $\zeta_{h,s}^2 := \sup_a \mathcal{E}_h^2(s, a)$ also yields the joint-action form

$$\Delta Q_h(s)^2 \leq 2\zeta_{h,s}^2 + 2 \sup_a \mathbb{E}_{s' \sim P_h(\cdot|s,a)}[W_{h+1}^2(s')], \tag{59}$$

which satisfies $\xi_h^2 \leq \Delta Q_h^2$ and $\mathcal{Z}_{h,s}^2 \leq \zeta_{h,s}^2$ pointwise.

**Step 2: Recursive unrolling with $W^2$ propagation.** We unroll the propagation term in (59) by iteratively applying (57) inside $W^2$. Specifically, by (55) and $\max(\cdot, \cdot) \leq (\cdot) + (\cdot)$,

$$W_{h+1}^2(s') \leq \mathbb{E}_{(a_1,a_2) \sim \pi_1^* \times \hat{\pi}_2}[(\hat{Q}_{h+1} - Q_{h+1}^*)^2(s', a_1, a_2)] + \mathbb{E}_{(a_1,a_2) \sim \hat{\pi}_1 \times \pi_2^*}[(\hat{Q}_{h+1} - Q_{h+1}^*)^2(s', a_1, a_2)].$$

Each term is an action-marginal expectation under a Player-$i$-Nash distribution at step $h+1$ (i.e., a single-step $\Pi_{\text{uni}}$ structure: Player $i$ at $\pi_i^*$, Player $-i$ at the corresponding learned policy). Substituting (57) pointwise into each Player-$i$-Nash branch of (55) and iterating the recursion from $h+1$ down to $H$ (with $W_{H+1}^2 \equiv 0$), we obtain the unrolled cumulative bound

$$W_{h+1}^2(s') \leq 2 \sum_{t=h+1}^{H} \sum_{i \in \{1,2\}} \mathbb{E}_{\tau_t^{(i)}}[\mathcal{E}_t^2(s_t, a_{t,1}, a_{t,2})], \tag{60}$$

where $\tau_t^{(i)} = (s_{h+1}, a_{h+1,1}, a_{h+1,2}, \ldots, s_t, a_{t,1}, a_{t,2})$ is the trajectory starting from $s_{h+1} = s'$ whose regression-error step $t$ has action marginal $(a_{t,1}, a_{t,2}) \sim \pi_{t,i}^* \times \hat{\pi}_{t,-i}$ (a Player-$i$-Nash distribution at step $t$, lying in single-step $\Pi_{\text{uni}}$). The iterative max-to-sum substitution in (55) would in principle produce cross-branch intermediate prefixes (Player 1 at Nash at some intermediate steps, Player 2 at Nash at others), which are *not* single multi-step $\Pi_{\text{uni}}$ trajectories. However, by Lemma B.4 part (iv) (Step D.7 of Appendix B.3) applied under the side condition (5), the state marginal at step $t$ of any such cross-branch prefix is bounded above by a constant factor times the state marginal under a single multi-step Player-$i$-at-Nash policy $\pi^{(i)} \in \Pi_{\text{uni}}$ (with $\pi^{(1)} \in \{\pi_1^*\} \times \Pi_2$ or $\pi^{(2)} \in \Pi_1 \times \{\pi_2^*\}$). Specifically, since a cross-branch prefix may differ from the target $\pi^{(i)}$ in BOTH players' policies on some segments, part (iv) gives the cumulative bound $C_{\text{trans}}^2 = e^2 \alpha^{-8H^2}$ (rather than the single-player $C_{\text{trans}}$ from parts (i)–(iii)). We absorb this $\alpha$-dependent constant into the prefactor of (62) below (which is in turn absorbed into the $(1/\alpha)^{\mathcal{O}(H^2)}$ multiplier of the headline bound), allowing us to take $\tau_t^{(i)}$ to be generated by a single multi-step $\Pi_{\text{uni}}$ policy $\pi^{(i)}$ in (60).

Define the *single-player squared Bellman regression error*

$$\mathcal{Z}_{t,s}^2 := \max_{i \in \{1,2\}} \sup_{\nu_i \in \Pi_i} \mathbb{E}_{(a_1,a_2) \sim \nu_i \times \pi_{t,-i}^*}[(\hat{Q}_t(s, a_1, a_2) - \mathcal{T}_t \hat{V}_{t+1}(s, a_1, a_2))^2]. \tag{61}$$

Each per-step regression-error term in (60) is of the form $\mathbb{E}_{\nu_i \times \pi_{-i}^*}[\mathcal{E}_t^2(s_t, \cdot, \cdot)]$ for some Player-$i$ stage policy $\nu_i$, and is therefore dominated by $\mathcal{Z}_{t,s_t}^2$ by definition. Combining (59) and (60) (the joint-action sup in (59) corresponds to choosing the worst Player-$i$ deterministic deviation $\nu_h$ at step $h$ with the opponent at Nash) and applying Cauchy–Schwarz on the $H$ unrolled regression-error terms,

$$\Delta Q_h(s)^2 \leq (4H) \sup_{\pi' \in \Pi_{\text{uni}}} \mathbb{E}_{\tau \sim \pi'(\cdot|s)}\left[\sum_{t=h}^{H} \mathcal{Z}_{t,s_t}^2\right], \tag{62}$$

where the prefactor $4H$ absorbs (i) the factors of 2 from (57) and from $\max(\cdot, \cdot) \leq (\cdot) + (\cdot)$, (ii) the union bound over the two unilateral branches $i \in \{1, 2\}$, and (iii) the Cauchy–Schwarz factor $H$ on the unrolled regression-error terms; the cumulative density-ratio transfer cost (bounded by $C_{\text{trans}}^2 = e^2 \alpha^{-8H^2}$ under (5), per the discussion preceding (60)), which converts cross-branch intermediate prefixes into a single multi-step $\Pi_{\text{uni}}$ trajectory, is absorbed separately into the headline $(1/\alpha)^{\mathcal{O}(H^2)}$ multiplier of Theorem 6.3. The trajectory $\tau$ in (62) is therefore induced by a single multi-step policy $\pi^{(i)} \in \Pi_{\text{uni}}$, *not* by a step-varying joint deterministic policy in $\Pi_1 \times \Pi_2$, so the sup-restriction to $\Pi_{\text{uni}}$ is genuinely tight up to the absorbed absolute constants.

Taking expectation of (62) under $s_h \sim d_h^{\mathrm{uni}}$ and applying the summation swap $\sum_{h=1}^{H} \sum_{t=h}^{H} = \sum_{t=1}^{H} \sum_{h=1}^{t}$,

$$\sum_{h=1}^{H} \mathbb{E}_{s_h \sim d_h^{\mathrm{uni}}}\big[\Delta Q_h(s_h)^2\big] \;\leq\; (4H^2) \sum_{t=1}^{H} \mathbb{E}_{s_t \sim d_t^{\mathrm{uni}}}\big[\mathcal{Z}_{t,s_t}^2\big].$$

Applying the same Cauchy–Schwarz unrolling to the single-player Nash-averaged recursion (58) in place of (59) yields

$$\sum_{h=1}^{H} \mathbb{E}_{s_h \sim d_h^{\mathrm{uni}}}\big[\xi_h^2(s_h)\big] \;\leq\; (4H^2) \sum_{t=1}^{H} \mathbb{E}_{s_t \sim d_t^{\mathrm{uni}}}\big[\mathcal{Z}_{t,s_t}^2\big], \tag{63}$$

where every per-step regression-error term has its action marginal at step $t$ in the single-player Nash-averaged form $\mathbb{E}_{\nu_i \times \pi_{-i}^*}[\mathcal{E}_t^2] \leq \mathcal{Z}_{t,s_t}^2$, and the absorbed density-ratio transfer (above) converts intermediate cross-branch state marginals into a single multi-step $\Pi_{\mathrm{uni}}$ trajectory at constant cost.

Substituting (63) into the combined bound (54), we obtain

$$\mathrm{Gap}(\hat{\pi}) \;\leq\; \mathcal{O}\big(\eta H^2 + \eta^3 H^6 (1 + \eta^{-1}\log(1/\alpha))^2\big) \sum_{t=1}^{H} \mathbb{E}_{s \sim d_t^{\mathrm{uni}}}\big[\mathcal{Z}_{t,s}^2\big], \tag{64}$$

where the $\eta H^2$ term arises from the linear-$\eta$ Gap 2 contribution combined with the Cauchy–Schwarz factor $4H^2$ from (63), and the $\eta^3 H^6 \lambda^2$ term arises from the cubic-$\eta$ Gap 1 prefactor combined with the same $4H^2$.

### B.7. Final Statistical Bound via Unilateral Concentrability.

Now, we provide the following two lemmas to bound the generalization error of the Q-function estimator $\hat{Q}_h$.

**Lemma B.6.** *For any step $h \in [H]$ and state-action samples $\{(s_{i,h}, a_{i,h,1}, a_{i,h,2})\}_{i=1}^{n}$ drawn i.i.d. from the marginal dataset distribution $\mu_h$, with probability at least $1 - \delta$, for any $Q_1$ and $Q_2 \in \mathcal{Q}$ we have*

$$\mathbb{E}_{\mu_h}\big[\big(Q_1(s, a_1, a_2) - Q_2(s, a_1, a_2)\big)^2\big] \leq \frac{2}{n} \sum_{i=1}^{n} \big(Q_1(s_{i,h}, a_{i,h,1}, a_{i,h,2}) - Q_2(s_{i,h}, a_{i,h,1}, a_{i,h,2})\big)^2 + \frac{80 C_{\mathrm{val}}^2}{3n} \log(2|\mathcal{Q}|/\delta),$$

*where $C_{\mathrm{val}} = H(1 + \eta^{-1}\log(1/\alpha))$.*

*Proof.* Let $z_{i,h} = (s_{i,h}, a_{i,h,1}, a_{i,h,2})$ be the samples drawn i.i.d. from the marginal dataset distribution $\mu_h$. For any fixed pair of functions $Q_1, Q_2 \in \mathcal{Q}$, define the random variable $X(z) = (Q_1(z) - Q_2(z))^2$. By Assumption B.1, every $f \in \mathcal{Q}$ satisfies $\|f\|_\infty \leq C_{\mathrm{val}} = H(1 + \eta^{-1}\log(1/\alpha))$, so $X(z) \in [0, (2C_{\mathrm{val}})^2]$. For simplicity in notation, let $B = (2C_{\mathrm{val}})^2$.

We apply Bernstein's inequality. For a fixed pair $Q_1, Q_2$, with probability at least $1 - \delta'$, we have:

$$\mathbb{E}[X] - \frac{1}{n} \sum_{i=1}^{n} X(z_{i,h}) \leq \sqrt{\frac{2 \mathrm{Var}(X) \log(1/\delta')}{n}} + \frac{2B \log(1/\delta')}{3n}.$$

A key property for the squared loss is relating the variance to the expectation. Since $X(z) \geq 0$ and $X(z) \leq B$, we have:

$$\mathrm{Var}(X) \leq \mathbb{E}[X^2] \leq B\mathbb{E}[X].$$

Substituting this bound into Bernstein's inequality:

$$\mathbb{E}[X] - \frac{1}{n} \sum_{i=1}^{n} X(z_{i,h}) \leq \sqrt{\frac{2B\mathbb{E}[X] \log(1/\delta')}{n}} + \frac{2B \log(1/\delta')}{3n}.$$

Using the inequality $\sqrt{xy} \leq \frac{x}{2} + \frac{y}{2}$ (AM-GM inequality) with $x = \mathbb{E}[X]$ and $y = \frac{2B \log(1/\delta')}{n}$, we bound the square root term:

$$\sqrt{\frac{2B\mathbb{E}[X] \log(1/\delta')}{n}} \leq \frac{1}{2}\mathbb{E}[X] + \frac{B \log(1/\delta')}{n}.$$

Plugging this back in:

$$\mathbb{E}[X] - \frac{1}{n}\sum_{i=1}^{n} X(z_{i,h}) \le \frac{1}{2}\mathbb{E}[X] + \frac{B\log(1/\delta')}{n} + \frac{2B\log(1/\delta')}{3n}$$

$$\implies \frac{1}{2}\mathbb{E}[X] \le \frac{1}{n}\sum_{i=1}^{n} X(z_{i,h}) + \frac{5B\log(1/\delta')}{3n}.$$

Multiplying by 2:

$$\mathbb{E}[X] \le \frac{2}{n}\sum_{i=1}^{n} X(z_{i,h}) + \frac{10B\log(1/\delta')}{3n}.$$

Finally, we apply a union bound over all possible pairs $(Q_1, Q_2) \in \mathcal{Q} \times \mathcal{Q}$. The total number of pairs is $|\mathcal{Q}|^2$. We set $\delta' = \delta/|\mathcal{Q}|^2$. Then:

$$\log(1/\delta') = \log(|\mathcal{Q}|^2/\delta) = 2\log|\mathcal{Q}| + \log(1/\delta) \le 2\log(2|\mathcal{Q}|/\delta).$$

Substituting this into the bound, we obtain:

$$\mathbb{E}_{\mu_h}\big[(Q_1 - Q_2)^2\big] \le \frac{2}{n}\sum_{i=1}^{n}\big(Q_1(z_{i,h}) - Q_2(z_{i,h})\big)^2 + \frac{20B}{3n}\log(2|\mathcal{Q}|/\delta).$$

This holds for all $Q_1, Q_2 \in \mathcal{Q}$ simultaneously with probability at least $1 - \delta$. $\qquad\square$

**Lemma B.7** (Least Squares Error Bound for Finite Q-Function Class). *Let $\mathcal{Q}$ be a finite function class with cardinality $|\mathcal{Q}|$. Assume completeness, i.e., for any $V_{h+1}$ induced by Algorithm 1, the Bellman update $\mathcal{T}_h V_{h+1} \in \mathcal{Q}$. Define the algorithm-induced next-step value class*

$$\mathcal{V}_{h+1}^{\mathrm{alg}} := \big\{V_{h+1}^{\tilde{Q}_{h+1:H}} : \tilde{Q}_{h+1:H} \in \mathcal{Q}^{H-h}\big\},$$

*where $V_{h+1}^{\tilde{Q}_{h+1:H}}$ is the regularized value obtained by running the backward-induction step of Algorithm 1 from $\tilde{Q}_H, \tilde{Q}_{H-1}, \ldots, \tilde{Q}_{h+1}$ with terminal value $V_{H+1} \equiv 0$. By construction, $\hat{V}_{h+1} \in \mathcal{V}_{h+1}^{\mathrm{alg}}$ and $|\mathcal{V}_{h+1}^{\mathrm{alg}}| \le |\mathcal{Q}|^{H-h}$.*

*Let $\mathcal{D}_h = \{(z_{i,h}, y_{i,h})\}_{i=1}^{n}$ be a dataset for step $h$, where the regression target is*

$$y_{i,h} = r_{i,h} + \hat{V}_{h+1}(s'_{i,h}) = (\mathcal{T}_h \hat{V}_{h+1})(z_{i,h}) + \epsilon_{i,h}, \qquad \epsilon_{i,h} = \xi_{i,h} + \big(\hat{V}_{h+1}(s'_{i,h}) - \mathbb{E}_{s' \sim P_h(\cdot|z_{i,h})}[\hat{V}_{h+1}(s')]\big).$$

*For any fixed $V \in \mathcal{V}_{h+1}^{\mathrm{alg}}$ chosen independently of the stage-$h$ samples, the corresponding noise $\epsilon_i(V) := \xi_{i,h} + (V(s'_{i,h}) - \mathbb{E}_{s' \sim P_h(\cdot|z_{i,h})}[V(s')])$ is conditionally zero-mean and $C_{\mathrm{sub}}$-sub-Gaussian given $z_{i,h}$, with $C_{\mathrm{sub}} := 1 + C_{\mathrm{val}}$: the reward noise $\xi_{i,h}$ is 1-sub-Gaussian by assumption, and the centered transition-induced term is $C_{\mathrm{val}}$-sub-Gaussian by Hoeffding's lemma (Assumption B.1 gives $|V(s')| \le C_{\mathrm{val}}$ uniformly in $V \in \mathcal{V}_{h+1}^{\mathrm{alg}}$, so the centered version has half-range $C_{\mathrm{val}}$).*

*Let $\hat{Q}_h$ be the least squares estimator*

$$\hat{Q}_h \in \arg\min_{f \in \mathcal{Q}} \sum_{i=1}^{n}\big(f(z_{i,h}) - y_{i,h}\big)^2.$$

*Then for any $\delta \in (0, 1)$, with probability at least $1 - \delta$,*

$$\sum_{i=1}^{n}\big(\hat{Q}_h(z_{i,h}) - (\mathcal{T}_h\hat{V}_{h+1})(z_{i,h})\big)^2 \le 8C_{\mathrm{sub}}^2 \log\left(\frac{|\mathcal{Q}| \cdot |\mathcal{V}_{h+1}^{\mathrm{alg}}|}{\delta}\right) \le 8C_{\mathrm{sub}}^2(H - h + 1)\log\left(\frac{|\mathcal{Q}|}{\delta}\right).$$

*Proof.* By the definition of the least squares estimator and the completeness assumption $(\mathcal{T}_h\hat{V}_{h+1}) \in \mathcal{Q}$,

$$\sum_{i=1}^{n}\big(\hat{Q}_h(z_{i,h}) - y_{i,h}\big)^2 \le \sum_{i=1}^{n}\big((\mathcal{T}_h\hat{V}_{h+1})(z_{i,h}) - y_{i,h}\big)^2.$$

Substituting $y_{i,h} = (\mathcal{T}_h \hat{V}_{h+1})(z_{i,h}) + \epsilon_{i,h}$ and expanding both sides yields

$$\sum_{i=1}^{n} \big(\hat{Q}_h(z_{i,h}) - (\mathcal{T}_h \hat{V}_{h+1})(z_{i,h})\big)^2 \leq 2 \sum_{i=1}^{n} \epsilon_{i,h}\big(\hat{Q}_h(z_{i,h}) - (\mathcal{T}_h \hat{V}_{h+1})(z_{i,h})\big). \tag{65}$$

Because $\hat{V}_{h+1}$ is constructed from the same trajectories that produce $\{(z_{i,h}, s'_{i,h})\}_{i=1}^n$ via backward induction (recall $s'_{i,h} = s_{i,h+1}$), the noise $\epsilon_{i,h}$ is *not* automatically conditionally zero-mean given $z_{i,h}$. We bypass this dependence by chaining over the algorithm-induced value class $\mathcal{V}_{h+1}^{\mathrm{alg}}$ defined above. Fix *any* pair $(f, V) \in \mathcal{Q} \times \mathcal{V}_{h+1}^{\mathrm{alg}}$, and let

$$\epsilon_i(V) := r_{i,h} + V(s'_{i,h}) - (\mathcal{T}_h V)(z_{i,h}), \qquad \Delta_{f,V,i} := f(z_{i,h}) - (\mathcal{T}_h V)(z_{i,h}).$$

For this fixed $V$, the variables $\{\epsilon_i(V)\}_{i=1}^n$ are independent zero-mean $C_{\mathrm{sub}}$-sub-Gaussian conditional on $\{z_{i,h}\}_{i=1}^n$ (Hoeffding's lemma applies because $V$ is a fixed function bounded by $C_{\mathrm{val}}$).

Consider

$$M_{f,V} = \exp\left(\frac{1}{2C_{\mathrm{sub}}^2} \sum_{i=1}^{n} \epsilon_i(V) \, \Delta_{f,V,i} - \frac{1}{8C_{\mathrm{sub}}^2} \sum_{i=1}^{n} \Delta_{f,V,i}^2 \right).$$

Conditioning on $z_{1:n,h}$ and applying the standard $C_{\mathrm{sub}}$-sub-Gaussian moment generating function bound,

$$\mathbb{E}\left[\exp\left(\frac{1}{2C_{\mathrm{sub}}^2}\epsilon_i(V)\,\Delta_{f,V,i} - \frac{1}{8C_{\mathrm{sub}}^2}\Delta_{f,V,i}^2\right) \,\Big|\, z_{i,h}\right] \leq 1.$$

By conditional independence, $\mathbb{E}[M_{f,V}] \leq 1$.

Markov's inequality gives, for any $t > 0$,

$$\mathbb{P}\left(2\sum_{i=1}^{n} \epsilon_i(V)\,\Delta_{f,V,i} > \frac{1}{2}\sum_{i=1}^{n}\Delta_{f,V,i}^2 + 4C_{\mathrm{sub}}^2 t\right) \leq e^{-t}.$$

Taking a union bound over all $(f, V) \in \mathcal{Q} \times \mathcal{V}_{h+1}^{\mathrm{alg}}$ (a total of $|\mathcal{Q}| \cdot |\mathcal{V}_{h+1}^{\mathrm{alg}}|$ events) and setting $t = \log(|\mathcal{Q}| \cdot |\mathcal{V}_{h+1}^{\mathrm{alg}}|/\delta)$, with probability at least $1 - \delta$ we have simultaneously for all $f \in \mathcal{Q}$ and $V \in \mathcal{V}_{h+1}^{\mathrm{alg}}$,

$$2\sum_{i=1}^{n} \epsilon_i(V)\big(f(z_{i,h}) - (\mathcal{T}_h V)(z_{i,h})\big) \leq \frac{1}{2}\sum_{i=1}^{n}\big(f(z_{i,h}) - (\mathcal{T}_h V)(z_{i,h})\big)^2 + 4C_{\mathrm{sub}}^2 \log\left(\frac{|\mathcal{Q}| \cdot |\mathcal{V}_{h+1}^{\mathrm{alg}}|}{\delta}\right).$$

Since $\hat{Q}_h \in \mathcal{Q}$ and $\hat{V}_{h+1} \in \mathcal{V}_{h+1}^{\mathrm{alg}}$, we may instantiate this uniform bound at $(f, V) = (\hat{Q}_h, \hat{V}_{h+1})$, in which case $\epsilon_i(\hat{V}_{h+1}) = \epsilon_{i,h}$ and $\Delta_{\hat{Q}_h, \hat{V}_{h+1}, i} = \hat{Q}_h(z_{i,h}) - (\mathcal{T}_h \hat{V}_{h+1})(z_{i,h})$. Combined with (65),

$$\sum_{i=1}^{n}\big(\hat{Q}_h(z_{i,h}) - (\mathcal{T}_h \hat{V}_{h+1})(z_{i,h})\big)^2 \leq \frac{1}{2}\sum_{i=1}^{n}\big(\hat{Q}_h(z_{i,h}) - (\mathcal{T}_h \hat{V}_{h+1})(z_{i,h})\big)^2 + 4C_{\mathrm{sub}}^2 \log\left(\frac{|\mathcal{Q}| \cdot |\mathcal{V}_{h+1}^{\mathrm{alg}}|}{\delta}\right).$$

Rearranging and using $|\mathcal{V}_{h+1}^{\mathrm{alg}}| \leq |\mathcal{Q}|^{H-h}$ to bound $\log(|\mathcal{Q}| \cdot |\mathcal{V}_{h+1}^{\mathrm{alg}}|/\delta) \leq (H - h + 1)\log(|\mathcal{Q}|/\delta) + \log(1/\delta)$ completes the proof. $\qquad\square$

**Final Bound Integration via $D^2$-divergence and density-ratio stability.** Combining Lemmas B.6 and B.7, the expected squared Bellman regression error under the data distribution $\mu_h$ satisfies

$$\mathbb{E}_{z \sim \mu_h}\left[(\hat{Q}_h(z) - (\mathcal{T}_h \hat{V}_{h+1})(z))^2\right] \leq \frac{16C_{\mathrm{sub}}^2(H - h + 1)\log(|\mathcal{Q}|/\delta) + \frac{80C_{\mathrm{val}}^2}{3}\log(2|\mathcal{Q}|/\delta)}{n} =: \epsilon_{\mathrm{stat},h}^2,$$

We now relate the trajectory error $\mathbb{E}_{d_h^{\mathrm{uni}}}[\mathcal{Z}_{h,s}^2]$ to this data-distribution error via a two-step argument that combines the $D^2$-divergence concentrability (Assumption 6.2, instantiated through (53)) with a Player-2 prefix transfer from the density-ratio stability lemma (Lemma B.4 (ii)).

*Step (a): unilateral marginal under Nash–Nash prefix.* For each $t \in [H]$, the worst-case squared regression error is

$$\mathcal{Z}_{t,s_t}^2 = \sup_{\nu_t} \mathbb{E}_{(a_1,a_2) \sim \nu_t \times \pi_{t,-i}^*} \big[(\hat{Q}_t - \mathcal{T}_t \hat{V}_{t+1})^2(s_t, a_1, a_2)\big],$$

which is the worst-case (over Player-$i$ deviations $\nu_t \in \Pi_i$) action-marginal squared error at fixed state $s_t$. Taking expectation under the unilateral state-marginal $s_t \sim d_t^{\pi_1^*, \hat{\pi}_2}$ (corresponding to the prefix $(\pi_1^*, \hat{\pi}_2) \in \Pi_{\text{uni}}$ that arises in Lemma 6.5) yields, for any deterministic $\nu_t \in \Pi_i$,

$$\mathbb{E}_{s_t \sim d_t^{\pi_1^*, \hat{\pi}_2}}\big[\mathcal{Z}_{t,s_t}^2\big] = \sup_{\nu_t} \mathbb{E}_{(s_t,a_1,a_2) \sim d_t^{\pi_1^*, \hat{\pi}_2} \otimes \nu_t \times \pi_{t,-i}^*}\big[(\hat{Q}_t - \mathcal{T}_t \hat{V}_{t+1})^2\big].$$

By Lemma B.4 (ii), under the condition (5) ($\eta \leq 1/(4H^2)$), the trajectory prefix can be transferred from $d_t^{\pi_1^*, \hat{\pi}_2}$ to $d_t^{\pi_1^*, \pi_2^*}$ at constant cost $C_{\text{trans}} = e \cdot \alpha^{-4H^2}$:

$$\mathbb{E}_{s_t \sim d_t^{\pi_1^*, \hat{\pi}_2}}\big[\mathcal{Z}_{t,s_t}^2\big] \leq C_{\text{trans}} \cdot \sup_{\nu_t} \mathbb{E}_{(s_t,a_1,a_2) \sim d_t^{\pi_1^*, \pi_2^*} \otimes \nu_t \times \pi_{t,-i}^*}\big[(\hat{Q}_t - \mathcal{T}_t \hat{V}_{t+1})^2\big].$$

*Step (b): $D^2$-divergence absorbs the deterministic deviation.* For each fixed $\nu_t \in \Pi_i$, the joint distribution $d_t^{\pi_1^*, \pi_2^*} \otimes \nu_t \times \pi_{t,-i}^*$ corresponds to the visitation under the time-varying joint policy with Nash play prior to step $t$ and the deterministic deviation $\nu_t$ for Player $i$ at step $t$ (with Player $-i$ at Nash throughout); concretely, this distribution equals $d_t^{\pi_i' \times \pi_{-i}^*}$ for a single multi-step Player-$i$ policy $\pi_i' \in \Pi_i$ that agrees with $\pi_i^*$ on steps $1, \ldots, t-1$ and equals $\nu_t$ at step $t$. Therefore $d_t^{\pi_i' \times \pi_{-i}^*}$ lies within the family covered by Assumption 6.2, and (53) gives

$$\mathbb{E}_{(s_t,a_1,a_2) \sim d_t^{\pi_i' \times \pi_{-i}^*}}\big[(\hat{Q}_t - \mathcal{T}_t \hat{V}_{t+1})^2\big] \leq C_{\text{uni}} \cdot \mathbb{E}_{\mu_t}\big[(\hat{Q}_t - \mathcal{T}_t \hat{V}_{t+1})^2\big].$$

Taking the sup over $\nu_t$ (equivalently, over $\pi_i' \in \Pi_i$) preserves the $C_{\text{uni}}$ bound by definition.

*Combining steps (a) and (b)* and summing over $t \in [H]$ yields

$$\sum_{t=1}^{H} \mathbb{E}_{s \sim d_t^{\pi_1^*, \hat{\pi}_2}}\big[\mathcal{Z}_{t,s}^2\big] \leq C_{\text{trans}} \cdot C_{\text{uni}} \cdot \sum_{t=1}^{H} \mathbb{E}_{\mu_t}\big[(\hat{Q}_t - \mathcal{T}_t \hat{V}_{t+1})^2\big] \leq C_{\text{trans}} \cdot C_{\text{uni}} \cdot \sum_{t=1}^{H} \epsilon_{\text{stat},t}^2, \tag{66}$$

where the first inequality uses Steps (a)–(b) at each $t$ and the second invokes the per-stage regression-error concentration from Lemma B.7. Plugging in $\epsilon_{\text{stat},t}^2 \leq 16 C_{\text{sub}}^2 (H - t + 1) \log(|\mathcal{Q}|/\delta)/n + \frac{80 C_{\text{val}}^2}{3} \log(2|\mathcal{Q}|/\delta)/n$ and summing the arithmetic series $\sum_{t=1}^{H}(H - t + 1) = H(H+1)/2$ gives $\sum_{t=1}^{H} \epsilon_{\text{stat},t}^2 = O\big(C_{\text{val}}^2 H^2 \log(|\mathcal{Q}|/\delta)/n\big)$, a factor of $H$ larger than the naive $H \cdot \epsilon_{\text{stat}}^2$ one would obtain with a fixed $\log(|\mathcal{Q}|/\delta)$ cost (this $H$ is the price of the algorithm-induced-value-class union bound). (The same chain holds verbatim for the symmetric unilateral marginal $d_t^{\hat{\pi}_1, \pi_2^*}$ that arises on the Term B side of the duality gap, with the Player-1 analog established as part (iii) of Step D.7 (Appendix B.3) replacing part (ii) in Step (a) since Player 1 rather than Player 2 is the deviating player in that case.)

Substituting (66) into (64) and combining with $C_{\text{val}}^2 = H^2(1 + \eta^{-1} \log(1/\alpha))^2$ yields the final duality gap:

$$\text{Gap}(\hat{\pi}) \leq \widetilde{\mathcal{O}}\bigg(\frac{(\eta H^6 + \eta^3 H^{10}) C_{\text{uni}} \log(|\mathcal{Q}|/\delta)}{n} \cdot \Big(\frac{1}{\alpha}\Big)^{\mathcal{O}(H^2)}\bigg),$$

where the two $\eta$-scalings correspond to the Gap 1 and Gap 2 chains and the $(1/\alpha)^{\mathcal{O}(H^2)}$ factor collects all $\alpha$-dependent multipliers (the $C_{\text{trans}}$ transfer cost from Lemma B.4 and the polylog factors $(1 + \eta^{-1} \log(1/\alpha))^k$ in $C_{\text{val}}^k$).

- **Gap 2 contribution** ($\eta H^6$)**:** From Lemma 6.6, $\text{Gap}_2 \leq \mathcal{O}(\eta(C_{\text{trans}} + \lambda^2)) \sum_{h=1}^{H} \mathbb{E}_{s \sim d_h^{\pi_1^*, \hat{\pi}_2}}[\xi_h^2(s)]$ on the Term A side (the symmetric marginal $d_h^{\hat{\pi}_1, \pi_2^*}$ arises on the Term B side; both are in $\Pi_{\text{uni}}$). The $H^6$ exponent decomposes as (i) $H^2$ from $C_{\text{val}}^2 = H^2 \lambda^2$ in $\epsilon_{\text{stat}}^2$, (ii) $H^2$ from the Bellman unrolling of $\xi^2$ via (63), and (iii) $H^2$ from horizon summation at the concentrability step combined with the value-class union-bound factor $(H - h + 1)$ in $\epsilon_{\text{stat},h}^2$.

- **Gap 1 contribution** ($\eta^3 H^{10}$): Lemma 6.5 contributes an additional $H^4$ multiplier on top of the shared $H^6$ from $\sum_h \mathbb{E}[\xi_h^2]$ (computed with the value-class union-bound). This $H^4$ decomposes as (iv) $H$ from the V-stability outer factor (the $6H$ in (27), equivalently the $2eH$ in the pointwise version (30)); (v) $H$ from the $\sum_{h=1}^H \sum_{t=h+1}^H \to \sum_{t=1}^H \sum_{h=1}^{t-1}$ swap in (36); and (vi) $H^2$ from $L_{t+1}^2 \le H^2 \lambda^2$ via Lemma B.2. The $\eta^3$ scaling comes from $\eta \cdot \eta^2$, where the $\eta^2$ originates from the squared stability (45).

Under the condition (5) we have $\eta^2 H^4 \le 1/16$, so $\eta^3 H^{10} \le \eta H^6/16$ and the $\eta H^6$ term dominates. This concludes the proof of Theorem 6.3.

## C. Proof of Optimization Convergence (Theorem 6.8)

### C.1. Sequential Duality Gap and Decomposition

We evaluate the convergence of the algorithm using the Duality Gap defined on the learned policy pair $\pi^{(T)} = (\pi_1^{(T)}, \pi_2^{(T)})$ (generated by Algorithm 2). To handle general zero-sum games without assuming symmetric transitions, we define the duality gap as the sum of the exploitability of each player with respect to the initial distribution $\rho$:

$$\mathrm{Gap}(\pi^{(T)}) := \mathbb{E}_{s_1 \sim \rho}\left[\underbrace{\left(V_1^{\pi^\dagger, \pi_2^{(T)}}(s_1) - V_1^{\pi_1^{(T)}, \pi_2^{(T)}}(s_1)\right)}_{\text{Player 1 Exploitability}} + \underbrace{\left(V_1^{\pi_1^{(T)}, \pi_2^{(T)}}(s_1) - V_1^{\pi_1^{(T)}, \pi^\ddagger}(s_1)\right)}_{\text{Player 2 Exploitability}}\right],$$

where $\pi^\dagger$ is the best response to $\pi_2^{(T)}$ (maximizing $V_1$) and $\pi^\ddagger$ is the best response to $\pi_1^{(T)}$ (minimizing $V_1$).

Since the Nash equilibrium value $V_1^{\pi_1^*, \pi_2^*}$ is unique, we can decompose this gap into two separate regret terms relative to the Nash equilibrium:

$$\mathrm{Gap}(\pi^{(T)}) = \underbrace{\mathbb{E}_{s_1 \sim \rho}\left[V_1^{\pi^\dagger, \pi_2^{(T)}}(s_1) - V_1^{\pi_1^*, \pi_2^*}(s_1)\right]}_{\text{Term A (Player 1 Regret)}} + \underbrace{\mathbb{E}_{s_1 \sim \rho}\left[V_1^{\pi_1^*, \pi_2^*}(s_1) - V_1^{\pi_1^{(T)}, \pi^\ddagger}(s_1)\right]}_{\text{Term B (Player 2 Regret)}}.$$

Our proof strategy is to rigorously bound **Term A**, and then show that **Term B** satisfies an identical bound due to the symmetry of the zero-sum optimization landscape.

### C.2. Proof Strategy

Our strategy decouples the optimization analysis from the statistical analysis via a Lipschitz transfer argument. Concretely,

$$\mathrm{Gap}(\pi^{(T)}) \le \mathrm{Gap}(\hat{\pi}) + \left|\mathrm{Gap}(\pi^{(T)}) - \mathrm{Gap}(\hat{\pi})\right|. \tag{67}$$

The first term is the statistical error already controlled by Theorem 6.3. The second term is the additional optimization error introduced by running self-play instead of computing the empirical equilibrium exactly. We control the second term via Lipschitzness of the duality gap (Lemma C.4), reducing it to a bound on $\sup_s \|\pi_h^{(T)}(\cdot|s) - \hat{\pi}_h(\cdot|s)\|_1$, which is in turn controlled by mirror-descent KL convergence (Lemma 6.7) and Pinsker's inequality.

The Lipschitzness argument requires both $\pi^{(T)}$ and $\hat{\pi}$ to lie in a uniformly bounded log-density-ratio class relative to the reference policy $\pi^{\mathrm{ref}}$. We establish this regularity in Lemma C.1, which we prove first.

### C.3. Proof of Lemma C.1 (Log-linear Boundedness)

**Lemma C.1** (Log-linear Boundedness of Self-play Iterates). *Let $\{\pi^{(t)}\}_{t=0}^T$ be the iterates generated by Algorithm 2 with stepsize $\gamma_t = \frac{2\eta}{t+2}$. Then for every $t \ge 0$, every player $i \in \{1, 2\}$, every step $h \in [H]$, every state $s \in \mathcal{S}$, and every $a \in \mathrm{supp}(\pi_{h,i}^{\mathrm{ref}}(\cdot|s))$,*

$$\left|\log \frac{\pi_{h,i}^{(t)}(a|s)}{\pi_{h,i}^{\mathrm{ref}}(a|s)}\right| \le 2\eta H\left(1 + \eta^{-1}\log(1/\alpha)\right).$$

*The empirical Nash equilibrium $\hat{\pi}_h$ also satisfies this bound.*

The proof proceeds by induction on $t$ and uses the boundedness of $\hat{Q}_h$ from Assumption B.1.

Fix an arbitrary context $(h, s)$ and let $\mathcal{A}_x := \text{supp}(\pi_{h,i}^{\text{ref}}(\cdot|s))$ denote the set of actions supported by the reference policy. For any vector $v \in \mathbb{R}^{\mathcal{A}_x}$, write $\text{osc}(v) := \max_{a \in \mathcal{A}_x} v(a) - \min_{a \in \mathcal{A}_x} v(a)$ for its oscillation.

*Proof of Lemma C.1.* For each player $i \in \{1, 2\}$, define the log-density ratio

$$u_i^{(t)}(a) := \log \frac{\pi_{h,i}^{(t)}(a|s)}{\pi_{h,i}^{\text{ref}}(a|s)}, \qquad a \in \mathcal{A}_x.$$

We show by induction on $t$ that $\text{osc}(u_i^{(t)}) \le C_{\log} := 2\eta H(1 + \eta^{-1}\log(1/\alpha))$ for both players. At initialization, $\pi_{h,i}^{(0)}(\cdot|s) = \pi_{h,i}^{\text{ref}}(\cdot|s)$, so $u_i^{(0)} \equiv 0$ and the claim holds.

Consider Player 1; the argument for Player 2 is identical. The update (3) can be written equivalently as $u_1^{(t+1)}(a) = w_1^{(t)}(a) - \log Z_1^{(t)}$, where

$$w_1^{(t)}(a) := \big(1 - \gamma_t\eta^{-1}\big)u_1^{(t)}(a) + \gamma_t\hat{q}_{h,1}^{(t)}(s, a), \qquad Z_1^{(t)} := \sum_{b \in \mathcal{A}_x} \pi_{h,1}^{\text{ref}}(b|s)\exp(w_1^{(t)}(b)).$$

Since $\hat{q}_{h,1}^{(t)}(s, a_1) = \mathbb{E}_{a_2 \sim \pi_{h,2}^{(t)}}[\hat{Q}_h(s, a_1, a_2)]$ and $\|\hat{Q}_h\|_\infty \le C_{\text{val}} = H(1 + \eta^{-1}\log(1/\alpha))$ by Assumption B.1, we have $\text{osc}(\hat{q}_{h,1}^{(t)}(s, \cdot)) \le 2H(1 + \eta^{-1}\log(1/\alpha))$. With the stepsize $\gamma_t = 2\eta/(t + 2) \le \eta$ for all $t \ge 0$, so $\gamma_t\eta^{-1} \in [0, 1]$, the inductive hypothesis $\text{osc}(u_1^{(t)}) \le C_{\log}$ yields

$$\text{osc}(w_1^{(t)}) \le (1 - \gamma_t\eta^{-1})C_{\log} + \gamma_t \cdot 2H(1 + \eta^{-1}\log(1/\alpha)) = (1 - \gamma_t\eta^{-1})C_{\log} + (\gamma_t\eta^{-1})C_{\log} = C_{\log},$$

where the second equality uses $\gamma_t \cdot 2H(1 + \eta^{-1}\log(1/\alpha)) = (\gamma_t\eta^{-1}) \cdot 2\eta H(1 + \eta^{-1}\log(1/\alpha)) = (\gamma_t\eta^{-1})C_{\log}$. Since subtracting a constant does not change oscillation, $\text{osc}(u_1^{(t+1)}) = \text{osc}(w_1^{(t)}) \le C_{\log}$. Moreover, because $\pi_{h,1}^{\text{ref}}(\cdot|s)$ is a probability distribution on $\mathcal{A}_x$, $\min_{a \in \mathcal{A}_x} w_1^{(t)}(a) \le \log Z_1^{(t)} \le \max_{a \in \mathcal{A}_x} w_1^{(t)}(a)$, so $|u_1^{(t+1)}(a)| \le \text{osc}(w_1^{(t)}) \le C_{\log}$ for all $a \in \mathcal{A}_x$. The argument for Player 2 with $-\hat{q}_{h,2}^{(t)}(s, \cdot)$ replacing $\hat{q}_{h,1}^{(t)}(s, \cdot)$ is identical.

It remains to show that the empirical Nash equilibrium $\hat{\pi}_h$ also satisfies the bound. By the saddle-point first-order conditions for the regularized stage game with payoff $\hat{Q}_h$, $\hat{\pi}_{h,1}$ and $\hat{\pi}_{h,2}$ take the Gibbs (softmax) form

$$\hat{\pi}_{h,1}(a|s) \propto \pi_{h,1}^{\text{ref}}(a|s)\exp\big(\eta\,\mathbb{E}_{a_2 \sim \hat{\pi}_{h,2}}[\hat{Q}_h(s, a, a_2)]\big), \quad \hat{\pi}_{h,2}(a|s) \propto \pi_{h,2}^{\text{ref}}(a|s)\exp\big(-\eta\,\mathbb{E}_{a_1 \sim \hat{\pi}_{h,1}}[\hat{Q}_h(s, a_1, a)]\big).$$

The unnormalized log-density ratio in both cases has oscillation at most $2\eta H(1 + \eta^{-1}\log(1/\alpha)) = C_{\log}$, so $|\log(\hat{\pi}_{h,i}(a|s)/\pi_{h,i}^{\text{ref}}(a|s))| \le C_{\log}$ by the same normalization argument. $\qquad\square$

The state-wise OMD lemma below is the workhorse of the optimization analysis. The version we use sharpens the standard $2\|\delta\|_\infty^2$ bound to a Hoeffding-type oscillation bound.

## C.4. State-wise Online Mirror Descent Lemma

We adapt the proof from Theorem 4 of Zhang et al. (2025b) into the following lemma, sharpened with the Hoeffding-type oscillation bound of Munos et al. (2024).

**Lemma C.2** (State-wise OMD Lemma). *Fix a state $s$. Let $\phi(\pi) = \sum_a \pi(a)\log\pi(a)$ be the negative entropy regularizer. Note that the Bregman divergence generated by $\phi$ corresponds to the KL-divergence: $D_\phi(\pi, \pi') = \text{KL}(\pi\|\pi')$.*

*Given a payoff vector $\delta(s, \cdot) \in \mathbb{R}^{|\mathcal{A}|}$ and a current policy $\pi^-(\cdot|s)$, let the updated policy $\pi^+(\cdot|s)$ be defined by the Mirror Descent step:*

$$\pi^+(\cdot|s) = \arg\max_{\pi \in \Delta(\mathcal{A})}\left[\sum_a \pi(a)\delta(s, a) - D_\phi(\pi, \pi^-(\cdot|s))\right].$$

*Then for any $\pi \in \Delta(\mathcal{A})$, the following Hoeffding-type descent inequality holds:*

$$\mathrm{KL}(\pi\|\pi^+(\cdot|s)) \leq \mathrm{KL}(\pi\|\pi^-(\cdot|s)) + \sum_a (\pi^-(a|s) - \pi(a))\delta(s, a) + \tfrac{1}{8}\mathrm{osc}(\delta(s, \cdot))^2,$$

*where $\mathrm{osc}(v) := \max_a v(a) - \min_a v(a)$ is the oscillation of the payoff vector.*

*Proof.* Write $\delta(a) := \delta(s, a)$ and $\pi^-(a) := \pi^-(a|s)$ for brevity. The closed-form solution of the Mirror Descent step is the exponential-weights update

$$\pi^+(a) = \frac{\pi^-(a)\exp(\delta(a))}{\sum_b \pi^-(b)\exp(\delta(b))}.$$

Therefore, for any $\pi \in \Delta(\mathcal{A})$,

$$\mathrm{KL}(\pi\|\pi^+) - \mathrm{KL}(\pi\|\pi^-) = \sum_a \pi(a)\log\frac{\pi^-(a)}{\pi^+(a)} = -\sum_a \pi(a)\delta(a) + \log\sum_a \pi^-(a)\exp(\delta(a)).$$

By Hoeffding's lemma applied to the bounded random variable $\delta(a)$ with $a \sim \pi^-$,

$$\log\sum_a \pi^-(a)\exp(\delta(a)) \leq \sum_a \pi^-(a)\delta(a) + \tfrac{1}{8}\mathrm{osc}(\delta)^2.$$

Substituting yields

$$\mathrm{KL}(\pi\|\pi^+) - \mathrm{KL}(\pi\|\pi^-) \leq \sum_a (\pi^-(a) - \pi(a))\delta(a) + \tfrac{1}{8}\mathrm{osc}(\delta)^2,$$

which proves the claim. $\qquad\square$

*Remark* C.3. The Hoeffding bound $\tfrac{1}{8}\mathrm{osc}(\delta)^2$ improves on the looser bound $2\|\delta\|_\infty^2$ that follows from 1-strong convexity of $\phi$ with respect to the $\ell_1$-norm. The improvement is by at least a factor of 16 since $\mathrm{osc}(\delta) \leq 2\|\delta\|_\infty$.

## C.5. Proof of Lemma 6.7 (Last-Iterate KL Convergence)

*Proof.* The proof proceeds by analyzing the iterates pointwise for each state $s$, then taking the expectation.

**Pointwise Analysis.** Fix an arbitrary context $(h, s)$ and let $\mathcal{A}_x := \mathrm{supp}(\pi_{h,i}^{\mathrm{ref}}(\cdot|s))$. We verify that the SOS-MD update (3)–(4) corresponds to the OMD step in Lemma C.2 with the direction

$$\delta_t(s, a) := \gamma_t\, \hat{q}_{h,i}^{(t)}(s, a) - \gamma_t\eta^{-1}\log\frac{\pi_{h,i}^{(t)}(a|s)}{\pi_{h,i}^{\mathrm{ref}}(a|s)}, \tag{68}$$

where the player-1 update uses $+\hat{q}_{h,1}^{(t)}$ and the player-2 update uses $-\hat{q}_{h,2}^{(t)}$. Apply Lemma C.2 with $\pi^- = \pi_h^{(t)}(\cdot|s)$, $\pi^+ = \pi_h^{(t+1)}(\cdot|s)$, target policy $\pi = \hat{\pi}_h(\cdot|s)$, and direction $\delta_t$ as above; sum the resulting inequalities for both players to obtain

$$V_{t+1}(s) \leq V_t(s) + \underbrace{\sum_{i=1}^2 \sum_{a \in \mathcal{A}_x} \left(\pi_{h,i}^{(t)}(a|s) - \hat{\pi}_{h,i}(a|s)\right)\delta_{t,i}(s, a)}_{\text{(I)}} + \tfrac{1}{8}\left(\mathrm{osc}(\delta_{t,1}(s, \cdot))^2 + \mathrm{osc}(\delta_{t,2}(s, \cdot))^2\right), \tag{69}$$

where $V_t(s) := \mathrm{KL}(\hat{\pi}_{h,1}(\cdot|s)\|\pi_{h,1}^{(t)}(\cdot|s)) + \mathrm{KL}(\hat{\pi}_{h,2}(\cdot|s)\|\pi_{h,2}^{(t)}(\cdot|s))$.

**Bounding the oscillation term.** We have $\mathrm{osc}(\hat{q}_{h,i}^{(t)}(s, \cdot)) \leq 2\|\hat{Q}_h\|_\infty \leq 2H(1 + \eta^{-1}\log(1/\alpha))$ by Assumption B.1, and by Lemma C.1, $\mathrm{osc}\left(\log(\pi_{h,i}^{(t)}(\cdot|s)/\pi_{h,i}^{\mathrm{ref}}(\cdot|s))\right) \leq 2C_{\log} = 4\eta H(1 + \eta^{-1}\log(1/\alpha))$. Therefore,

$$\mathrm{osc}(\delta_{t,i}(s, \cdot)) \leq \gamma_t \cdot 2H(1 + \eta^{-1}\log(1/\alpha)) + \gamma_t\eta^{-1} \cdot 4\eta H(1 + \eta^{-1}\log(1/\alpha)) = 6\gamma_t H(1 + \eta^{-1}\log(1/\alpha)).$$

Hence the oscillation term in (69) is bounded by

$$\tfrac{1}{8}\sum_{i=1}^2 \mathrm{osc}(\delta_{t,i}(s, \cdot))^2 \leq 9\,\gamma_t^2\, H^2(1 + \eta^{-1}\log(1/\alpha))^2.$$

**Bounding term (I) via the joint saddle-point property of $\hat{\pi}_h$.** Substituting (68) into term (I) and using $\sum_a (\pi_{h,i}^{(t)}(a|s) - \hat{\pi}_{h,i}(a|s)) = 0$,

$$(I) = \gamma_t \big[ \langle \pi_{h,1}^{(t)} - \hat{\pi}_{h,1}, \hat{q}_{h,1}^{(t)} \rangle - \langle \pi_{h,2}^{(t)} - \hat{\pi}_{h,2}, \hat{q}_{h,2}^{(t)} \rangle \big]$$
$$- \gamma_t \eta^{-1} \sum_{i=1}^2 \Big\langle \pi_{h,i}^{(t)} - \hat{\pi}_{h,i}, \log \frac{\pi_{h,i}^{(t)}}{\pi_{h,i}^{\mathrm{ref}}} \Big\rangle.$$

Using the identity $-\langle \pi^{(t)} - \hat{\pi}, \log(\pi^{(t)}/\pi^{\mathrm{ref}}) \rangle = -\mathrm{KL}(\pi^{(t)}\|\pi^{\mathrm{ref}}) + \mathrm{KL}(\hat{\pi}\|\pi^{\mathrm{ref}}) - \mathrm{KL}(\hat{\pi}\|\pi^{(t)})$ on both players, term (I) becomes

$$(I) = \gamma_t \big[ \langle \pi_{h,1}^{(t)} - \hat{\pi}_{h,1}, \hat{q}_{h,1}^{(t)} \rangle - \langle \pi_{h,2}^{(t)} - \hat{\pi}_{h,2}, \hat{q}_{h,2}^{(t)} \rangle \big]$$
$$+ \gamma_t \eta^{-1} \big[ -\mathrm{KL}(\pi_{h,1}^{(t)}\|\pi_{h,1}^{\mathrm{ref}}) + \mathrm{KL}(\hat{\pi}_{h,1}\|\pi_{h,1}^{\mathrm{ref}}) - \mathrm{KL}(\pi_{h,2}^{(t)}\|\pi_{h,2}^{\mathrm{ref}}) + \mathrm{KL}(\hat{\pi}_{h,2}\|\pi_{h,2}^{\mathrm{ref}}) \big]$$
$$- \gamma_t \eta^{-1} V_t(s).$$

Now substitute $\hat{q}_{h,1}^{(t)}(s,\cdot) = \mathbb{E}_{a_2 \sim \pi_{h,2}^{(t)}}[\hat{Q}_h(s,\cdot,a_2)]$ and $\hat{q}_{h,2}^{(t)}(s,\cdot) = \mathbb{E}_{a_1 \sim \pi_{h,1}^{(t)}}[\hat{Q}_h(s,a_1,\cdot)]$ into the first line: the $\mathbb{E}_{a \sim \pi_{h,1}^{(t)} \times \pi_{h,2}^{(t)}}[\hat{Q}_h(s,a)]$ cross-terms cancel, leaving $\mathbb{E}_{\pi_{h,1}^{(t)} \times \hat{\pi}_{h,2}}[\hat{Q}_h] - \mathbb{E}_{\hat{\pi}_{h,1} \times \pi_{h,2}^{(t)}}[\hat{Q}_h]$. Combining with the KL terms on the second line yields

$$(I) = \gamma_t \cdot \underbrace{\big[ J_s(\pi_{h,1}^{(t)}, \hat{\pi}_{h,2}; \hat{Q}_h) - J_s(\hat{\pi}_{h,1}, \pi_{h,2}^{(t)}; \hat{Q}_h) \big]}_{\leq 0 \text{ by joint saddle-point of } \hat{\pi}_h} - \gamma_t \eta^{-1} V_t(s), \tag{70}$$

where $J_s(\pi_1, \pi_2; Q) := \mathbb{E}_{a \sim \pi_1 \times \pi_2}[Q(s,a)] - \eta^{-1}\mathrm{KL}(\pi_1\|\pi_{h,1}^{\mathrm{ref}}(\cdot|s)) + \eta^{-1}\mathrm{KL}(\pi_2\|\pi_{h,2}^{\mathrm{ref}}(\cdot|s))$ is the stage-game regularized objective at state $s$ with payoff $Q$. Since $\hat{\pi}_h$ is the saddle point of $J_s(\cdot,\cdot; \hat{Q}_h)$, for all $\pi_1, \pi_2$,

$$J_s(\pi_1, \hat{\pi}_{h,2}; \hat{Q}_h) \leq J_s(\hat{\pi}_{h,1}, \hat{\pi}_{h,2}; \hat{Q}_h) \leq J_s(\hat{\pi}_{h,1}, \pi_2; \hat{Q}_h),$$

which (setting $\pi_1 = \pi_{h,1}^{(t)}$, $\pi_2 = \pi_{h,2}^{(t)}$) makes the bracketed term in (70) $\leq 0$. Hence

$$(I) \leq -\gamma_t \eta^{-1} V_t(s).$$

**Combining and solving the recursion.** Substituting both bounds into (69):

$$V_{t+1}(s) \leq (1 - \gamma_t \eta^{-1}) V_t(s) + 9\gamma_t^2 H^2 (1 + \eta^{-1} \log(1/\alpha))^2. \tag{71}$$

With $\gamma_t = 2\eta/(t+2)$, we have $1 - \gamma_t \eta^{-1} = t/(t+2)$ and $\gamma_t^2 = 4\eta^2/(t+2)^2$. Let $D := 36\eta^2 H^2 (1 + \eta^{-1} \log(1/\alpha))^2$. Then (71) becomes

$$V_{t+1}(s) \leq \frac{t}{t+2} V_t(s) + \frac{D}{(t+2)^2}.$$

We prove $V_t(s) \leq D/(t+1)$ for all $t \geq 1$ by induction. Base case ($t=1$): $V_1(s) \leq 0 \cdot V_0(s) + D/4 \leq D/2$. Inductive step: assume $V_t(s) \leq D/(t+1)$. Then

$$V_{t+1}(s) \leq \frac{t}{t+2} \cdot \frac{D}{t+1} + \frac{D}{(t+2)^2} = D \cdot \frac{t(t+2) + (t+1)}{(t+1)(t+2)^2} = D \cdot \frac{t^2 + 3t + 1}{(t+1)(t+2)^2} \leq \frac{D}{t+2},$$

where the final inequality uses $t^2 + 3t + 1 \leq (t+1)(t+2) = t^2 + 3t + 2$. This proves $V_T(s) \leq D/(T+1)$ pointwise.

**Aggregation.** Since the induction above is pointwise in $s$, the bound $V_T(s) \leq D/(T+1)$ holds uniformly over $s \in \mathcal{S}$. Taking the supremum,

$$\sup_{s \in \mathcal{S}} V_T(s) \leq \frac{36\eta^2 H^2 (1 + \eta^{-1} \log(1/\alpha))^2}{T+1}.$$

Since $V_T(s) \geq \mathrm{KL}(\hat{\pi}_{h,i}(\cdot|s)\|\pi_{h,i}^{(T)}(\cdot|s))$ for each player $i$, we obtain in particular

$$\sup_{s \in \mathcal{S}} \mathrm{KL}(\hat{\pi}_{h,i}(\cdot|s)\|\pi_{h,i}^{(T)}(\cdot|s)) \leq \frac{36\eta^2 H^2 (1 + \eta^{-1} \log(1/\alpha))^2}{T+1}.$$

$\square$

## C.6. Proof of Lemma C.4 (Lipschitzness of the Duality Gap)

**Lemma C.4** (Lipschitzness of the Duality Gap). *Let $\pi = (\pi_1, \pi_2)$ and $\pi' = (\pi'_1, \pi'_2)$ be two policy pairs that both lie in the log-linear-bounded class of Lemma C.1. Then*

$$\left|\mathrm{Gap}(\pi) - \mathrm{Gap}(\pi')\right| \leq \mathcal{O}\big(H^2(1 + \eta^{-1}\log(1/\alpha))\big) \cdot \sum_{h=1}^{H} \sup_{s \in \mathcal{S}}\big(\|\pi_{h,1}(\cdot|s) - \pi'_{h,1}(\cdot|s)\|_1 + \|\pi_{h,2}(\cdot|s) - \pi'_{h,2}(\cdot|s)\|_1\big).$$

The proof expands $\mathrm{Gap}$ via Bellman recursion using the bounded log-density ratios from Lemma C.1 to control the KL gradient.

*Proof.* Define the best-response value functionals

$$\mathcal{V}_1(\pi_2) := \mathbb{E}_{s_1 \sim \rho}\big[V_1^{\dagger,\pi_2}(s_1)\big], \qquad \mathcal{V}_2(\pi_1) := \mathbb{E}_{s_1 \sim \rho}\big[V_1^{\pi_1,\dagger}(s_1)\big],$$

so that $\mathrm{Gap}(\pi) = \mathcal{V}_1(\pi_2) - \mathcal{V}_2(\pi_1)$. By the triangle inequality, it suffices to show that $\mathcal{V}_1$ and $\mathcal{V}_2$ are each Lipschitz in their argument.

We focus on $\mathcal{V}_1$; the argument for $\mathcal{V}_2$ is symmetric. For any two Player-2 policies $\pi_2, \pi'_2$ in the log-linear-bounded class,

$$\left|\mathcal{V}_1(\pi_2) - \mathcal{V}_1(\pi'_2)\right| \leq \sup_{\tilde\pi_1} \mathbb{E}_{s_1 \sim \rho}\big|V_1^{\tilde\pi_1,\pi_2}(s_1) - V_1^{\tilde\pi_1,\pi'_2}(s_1)\big|.$$

Fix any $\tilde\pi_1$ (which is itself the regularized best response to either $\pi_2$ or $\pi'_2$ in the worst case, and hence by the same Gibbs-form argument as in Lemma C.1's second part lies in the log-linear class with constant $C_{\log}$). For each $h \in [H]$ and state $s$, define $\Delta_h(s) := V_h^{\tilde\pi_1,\pi_2}(s) - V_h^{\tilde\pi_1,\pi'_2}(s)$. By the regularized Bellman recursion,

$$\Delta_h(s) = \mathbb{E}_{a \sim \tilde\pi_{h,1} \times \pi_{h,2}}\big[Q_h^{\tilde\pi_1,\pi_2}(s,a)\big] - \mathbb{E}_{a \sim \tilde\pi_{h,1} \times \pi'_{h,2}}\big[Q_h^{\tilde\pi_1,\pi'_2}(s,a)\big]$$
$$+ \eta^{-1}\big[\mathrm{KL}(\pi_{h,2}\|\pi_{h,2}^{\mathrm{ref}}) - \mathrm{KL}(\pi'_{h,2}\|\pi_{h,2}^{\mathrm{ref}})\big](s).$$

The first line decomposes via add-and-subtract into a stage-$h$ Player-2 policy difference (bounded by $\|\pi_{h,2}(\cdot|s) - \pi'_{h,2}(\cdot|s)\|_1 \cdot \|Q_h^{\tilde\pi_1,\pi_2}\|_\infty$) plus a continuation difference $\mathbb{E}_{s' \sim P_h}[\Delta_{h+1}(s')]$. Using $\|Q_h^{\tilde\pi_1,\nu}\|_\infty \leq H(1 + \eta^{-1}\log(1/\alpha))$ for any policy $\nu$ in the log-linear class (which follows from Lemma B.2 together with the fact that the KL-regularization terms are bounded by $\eta^{-1}C_{\log}$), the stage-$h$ contribution is at most $H(1 + \eta^{-1}\log(1/\alpha)) \cdot \|\pi_{h,2}(\cdot|s) - \pi'_{h,2}(\cdot|s)\|_1$.

For the KL-regularization term, by the same mean-value argument as in the $B_{\mathrm{reg}}$ derivation in the proof of Lemma B.3 (Appendix B.2), and using that $\pi_{h,2}, \pi'_{h,2}$ both lie in the log-linear class with $|\log(\pi_{h,2}/\pi_{h,2}^{\mathrm{ref}})| \leq C_{\log}$,

$$\eta^{-1}\big|\mathrm{KL}(\pi_{h,2}(\cdot|s)\|\pi_{h,2}^{\mathrm{ref}}(\cdot|s)) - \mathrm{KL}(\pi'_{h,2}(\cdot|s)\|\pi_{h,2}^{\mathrm{ref}}(\cdot|s))\big| \leq \eta^{-1}(1 + C_{\log}) \cdot \|\pi_{h,2}(\cdot|s) - \pi'_{h,2}(\cdot|s)\|_1.$$

Combining (and using $\eta^{-1}(1 + 2\eta H(1 + \eta^{-1}\log(1/\alpha))) = \mathcal{O}(H(1 + \eta^{-1}\log(1/\alpha))))$,

$$|\Delta_h(s)| \leq \mathcal{O}(H(1 + \eta^{-1}\log(1/\alpha))) \cdot \|\pi_{h,2}(\cdot|s) - \pi'_{h,2}(\cdot|s)\|_1 + \mathbb{E}_{s' \sim P_h}\big[|\Delta_{h+1}(s')|\big].$$

Unrolling from $h = 1$ to $H$ along trajectories,

$$|\Delta_1(s_1)| \leq \mathcal{O}(H(1 + \eta^{-1}\log(1/\alpha))) \sum_{h=1}^{H} \mathbb{E}_{s_h \sim d_h^{\tilde\pi_1,\pi_2}}\big[\|\pi_{h,2}(\cdot|s_h) - \pi'_{h,2}(\cdot|s_h)\|_1\big].$$

Taking the supremum over the visitation distribution (the resulting bound is dominated by the state-uniform $L_1$ norm $\sup_s \|\pi_{h,2}(\cdot|s) - \pi'_{h,2}(\cdot|s)\|_1$), and aggregating over $\rho$,

$$|\mathcal{V}_1(\pi_2) - \mathcal{V}_1(\pi'_2)| \leq \mathcal{O}(H \cdot H(1 + \eta^{-1}\log(1/\alpha))) \cdot \sum_{h=1}^{H} \sup_s \|\pi_{h,2}(\cdot|s) - \pi'_{h,2}(\cdot|s)\|_1.$$

The symmetric bound for $\mathcal{V}_2$ in $\pi_1$ holds. Combining,

$$|\mathrm{Gap}(\pi) - \mathrm{Gap}(\pi')| \leq \mathcal{O}(H^2(1 + \eta^{-1}\log(1/\alpha))) \sum_{h=1}^{H} \big(\sup_s \|\pi_{h,1}(\cdot|s) - \pi'_{h,1}(\cdot|s)\|_1 + \sup_s \|\pi_{h,2}(\cdot|s) - \pi'_{h,2}(\cdot|s)\|_1\big).$$

$$\square$$

### C.7. Proof of Theorem 6.8

*Proof.* By Lemma 6.7, $\sup_s \mathrm{KL}(\hat{\pi}_h(\cdot|s)\|\pi_h^{(T)}(\cdot|s)) \le 36\eta^2 H^2(1 + \eta^{-1}\log(1/\alpha))^2/(T+1)$. Pinsker's inequality gives, pointwise in $s$,

$$\|\hat{\pi}_h(\cdot|s) - \pi_h^{(T)}(\cdot|s)\|_1 \le \sqrt{2\,\mathrm{KL}(\hat{\pi}_h(\cdot|s)\|\pi_h^{(T)}(\cdot|s))}.$$

Taking the supremum over $s$ on both sides and monotonicity of $\sqrt{\cdot}$,

$$\sup_{s\in\mathcal{S}} \|\hat{\pi}_h(\cdot|s) - \pi_h^{(T)}(\cdot|s)\|_1 \le \sqrt{2\cdot\sup_s \mathrm{KL}(\hat{\pi}_h(\cdot|s)\|\pi_h^{(T)}(\cdot|s))} \le 6\sqrt{2}\,\eta H(1+\eta^{-1}\log(1/\alpha))/\sqrt{T+1} = \widetilde{\mathcal{O}}(\eta H\lambda/\sqrt{T}),$$

where $\lambda = 1 + \eta^{-1}\log(1/\alpha)$. $\qquad\square$

### C.8. Proof of Total Error Bound (Corollary 6.9)

*Proof.* By the transfer inequality (67),

$$\mathrm{Gap}(\pi^{(T)}) \le \mathrm{Gap}(\hat{\pi}) + \left|\mathrm{Gap}(\pi^{(T)}) - \mathrm{Gap}(\hat{\pi})\right|.$$

By Lemma C.1, both $\pi^{(T)}$ and $\hat{\pi}$ lie in the log-linear class. Lemma C.4 therefore applies:

$$\left|\mathrm{Gap}(\pi^{(T)}) - \mathrm{Gap}(\hat{\pi})\right| \le \mathcal{O}(H^2(1 + \eta^{-1}\log(1/\alpha))) \sum_{h=1}^{H} \sup_s \left(\|\pi_{h,1}^{(T)}(\cdot|s) - \hat{\pi}_{h,1}(\cdot|s)\|_1 + \|\pi_{h,2}^{(T)}(\cdot|s) - \hat{\pi}_{h,2}(\cdot|s)\|_1\right).$$

By Theorem 6.8, each summand $\sup_s \|\hat{\pi}_{i,h}(\cdot|s) - \pi_{i,h}^{(T)}(\cdot|s)\|_1$ is $\widetilde{\mathcal{O}}(\eta H\lambda/\sqrt{T})$, so the sum over $h \in [H]$ is $\widetilde{\mathcal{O}}(\eta H^2\lambda/\sqrt{T})$. Multiplied by the Lipschitz constant $\mathcal{O}(H^2\lambda) = \mathcal{O}(H^2(1 + \eta^{-1}\log(1/\alpha)))$, the optimization error is

$$\left|\mathrm{Gap}(\pi^{(T)}) - \mathrm{Gap}(\hat{\pi})\right| \le \widetilde{\mathcal{O}}\left(\frac{\eta H^4\lambda^2}{\sqrt{T}}\right).$$

By Theorem 6.3, $\mathrm{Gap}(\hat{\pi}) \le \widetilde{\mathcal{O}}\left(\eta H^6 C_{\mathrm{uni}}\log(|\mathcal{Q}|/\delta)/n \cdot C_\alpha\right)$ with probability at least $1 - \delta$. Combining,

$$\mathrm{Gap}(\pi^{(T)}) \le \widetilde{\mathcal{O}}\left(\frac{\eta H^4\lambda^2}{\sqrt{T}}\right) + \widetilde{\mathcal{O}}\left(\frac{\eta H^6 C_{\mathrm{uni}}\log(|\mathcal{Q}|/\delta)}{n}\cdot C_\alpha\right).$$

$\qquad\square$

