# OpenReview forum: "Offline Two-Player Zero-Sum Markov Games with KL Regularization"
_ICML.cc/2026/Conference — ICML 2026 regular_

### Official Review · Reviewer_P8MJ · 2026-03-08

**Soundness:** 3
**Presentation:** 3
**Significance:** 3
**Originality:** 2
**Overall Recommendation:** 4
**Confidence:** 3

**Summary:**

This paper investigates learning Nash Equilibrium in offline two-player zero-sum game using KL-regularization as implicit pessimism. Compared to previous algorithms with $\tilde{O}(1/\sqrt{n})$ convergence rate, the authors introduce the Regularized Offline Sequential Equilibrium (ROSE) framework and a more practical Sequential Offline Self-play Mirror Descent (SOS-MD), both with $\tilde{O}(1/n)$ convergence rate.

**Compliance With Llm Reviewing Policy:**

Affirmed.

**Final Justification:**

The authors listed the complete regret bound comparison with previous works in the rebuttal. While the regret bounds improve its dependence on $n$, other dependencies (e.g. $C$, $H$) make the regrets worse. Therefore, I raised my score to be 4.

**Key Questions For Authors:**

1. Compared to Ye et al., can you explain the algorithmic difference of your algorithm and intuitively how the error rate is improved.

2. Can you do some experiments to show your algorithm behaves better than the state of the art, e.g. in LLM alignment?

**Limitations:**

yes

**Strengths And Weaknesses:**

Strengths:

1. The paper provides solid analysis on the duality gap of the proposed algorithms. It is nice to see that the algorithms improve the convergence rate from $1/\sqrt{n}$ to $1/n$.

2. The authors provide a clear presentation of the algorithms, the analysis and the literature review.

Weaknesses:

1. Lack of novelty. The Mirror Descent framework with KL divergence has been widely studied in the literature of multi-agent alignment(e.g. Munos et al.). The paper seems more of its counterpart analysis in offline Markov games.

2. The comparison in Tabel 1 is not complete. The table only shows the convergence rate dependece on $n$, while other dependences are missing.

---

> ### Author Rebuttal · Authors · 2026-03-31
>
> We appreciate your thoughtful comments. As you suggested, our paper improves the convergence rate from $\frac{1}{n}$ to $\frac{1}{\sqrt{n}}$. Below, we provide clarifications to address the remaining concerns.
>
> ## For Weaknesses:
> 1. >The Mirror...has been widely studied…
>
> Thank you for the question and the opportunity to clarify the novelty of our work. We will update our comparison to highlight these fundamental distinctions:
> - **Planning vs. Learning**: The most critical difference is that Munos et al. focus on iteration complexity, establishing the convergence of the Nash-MD optimization algorithm assuming access to the game’s payoffs, which corresponds to a **planning or full-information** setting where the game is known. In contrast, our work addresses the statistical sample complexity of **learning from a fixed finite offline dataset**, where both rewards and transitions, and thus the underlying game, must be inferred from data. This makes our setting more challenging, since the learner must first estimate the game from a fixed offline dataset and then compute a near-equilibrium policy.
> - **Bandit vs. Multi-step**: Munos et al. primarily address the one-step bandit setting ($H=1$). Our work generalizes this to the multi-step Markov game ($H>1$) setting. This is a non-trivial extension that must account for the recursive propagation of estimation errors through state transitions.
> - **Implicit Pessimism**: While Munos et al. use KL-regularization for optimization stability, our work demonstrates that it serves as a mechanism for implicit pessimism. This property is essential for stabilizing offline learning under distribution shift, a challenge that does not arise in the iterative optimization framework of NLHF in Munos et al..
>
> In summary, we establish the first fast statistical rate of $\tilde{\mathcal{O}}(1/n)$ for multi-step offline games, providing a theoretical bridge between practical alignment methods and game-theoretic sample complexity.
>
> 2. >The comparison in Tabel 1 is not complete …
>
> Thanks for this constructive suggestion. In our comparison, we prioritized the statistical rate with respect to the sample size $n$, as the improvement from the "slow" $\tilde{\mathcal{O}}(1/\sqrt{n})$ rate to the "fast" $\tilde{\mathcal{O}}(1/n)$ rate is the most important metric for evaluating sample efficiency and represents our primary theoretical contribution.
> While we focused on $n$ to highlight this fundamental shift, we agree that a complete comparison is valuable. In the revision, we will update Table 1 to include the full dependencies on all relevant parameters ($H, S, A$, and the concentrability coefficient $C$) to provide a more comprehensive overview of the landscape.
>
> ## For Questions:
> 1. >Compared to Ye et al., can you explain the algorithmic difference…
>
> Thank you for raising this question. Our work fundamentally differs from Ye et al. (2024) in its pessimism-free mechanism and statistical efficiency:
> - **Implicit vs. Explicit Pessimism**: Ye et al. rely on explicit pessimism through manually-tuned uncertainty bonuses. In contrast, our SOS-MD is pessimism-free, naturally inducing implicit pessimism via KL-regularization. This handles distribution shifts automatically without the sensitivity and complexity of manual tuning.
> - **Fast $\tilde{\mathcal{O}}(1/n)$ Rate**: By leveraging the strong convexity of the KL-regularizer, we establish a fast statistical rate, surpassing the standard $\tilde{\mathcal{O}}(1/\sqrt{n})$ rate achieved in Ye et al.
> - **Multi-step Setting**: We generalize the analysis from contextual bandits ($H=1$) to multi-step Markov games ($H>1$), addressing the complex recursive error propagation absent in the bandit setting.
>
> 2. >Can you do some experiments to show your algorithm behaves better than the state of the art, e.g. in LLM alignment?
>
> We appreciate the suggestion. Focusing on theoretical foundations is an **established standard** for papers introducing new sample complexity frameworks in offline Markov games (e.g., [Cui’22, Zhong’22]). Our primary goal is establishing the first fast $\tilde{\mathcal{O}}(1/n)$ rate and demonstrating how KL-regularization provides implicit pessimism. Applying this framework to multi-turn LLM alignment presents non-trivial challenges, such as reward credit assignment for sequential feedback and defining evaluation dynamics for multi-step interactions. As resolving these empirical design questions constitutes a substantial research endeavor in its own right, we defer the practical application of our framework to future work.
>
> We hope these clarifications fully address your concerns and demonstrate the technical contributions of our work. We are also happy for further discussion if any points require additional detail.
>
> ## Reference:
> (for [Cui'22] and [Zhong'22] see response to Reviewer A4ge)
>
> [Ye'24] Ye et al. Optimistic nonlinear mirror descent for offline games. ICML.
>
> [Munos'23] Munos et al. Nash learning from human feedback. ICML.

---

> > ### Author Rebuttal · Reviewer_P8MJ · 2026-04-02
> >
> > I appreciate the authors for addressing my questions and highlighting the novelty of the paper. I am willing to raise the score. While I understand that the authors claim the main contribution to establish a first $1/n$ for multi-step offline games, I am curious how the regret bound is compared to previous works, in terms of other relevant parameters. Can the authors list them in the comments?

---

> > > ### Author Response · Authors · 2026-04-07
> > >
> > > Thank you for your kind words and for considering raising the score. We are happy to provide the complete parameter dependencies beyond the $n$-rate.
> > >
> > > **Cui & Du (2022)** achieves a duality gap of $\widetilde{\mathcal{O}}\left(\sqrt{CSABH^3/n}\right)$, where $S$ is the size of the state space, $A$ and $B$ are the action space sizes of the two players, $C$ is the concentrability coefficient, and $H$ is the horizon. Since they study the tabular setting, the bound has explicit dependence on $S$, $A$, and $B$.
> > >
> > > **Zhong et al. (2022)** focuses on the linear function approximation setting and achieves a duality gap of $\widetilde{\mathcal{O}}\left(d^{3/2}H^2\sqrt{C/n}\right)$, where $d$ is the feature dimension.
> > >
> > > **Zhang et al. (2023)** studies the discounted setting with general function approximation and achieves a duality gap of $\widetilde{\mathcal{O}}\left(\frac{\sqrt{C\log|\mathcal{F}|}}{(1-\gamma)^2\sqrt{n}}\right)$, where $\frac{1}{1-\gamma}$ plays the role of the horizon $H$ in the discounted setting, and $\log|\mathcal{F}|$ reflects the complexity of the function class.
> > >
> > > **Ye et al. (2024)** studies the single-step (contextual bandit) setting and achieves a duality gap of $\widetilde{\mathcal{O}}\left(\sqrt{C\log|\mathcal{P}|/n}\right)$, where $\log|\mathcal{P}|$ is the complexity of the policy class.
> > >
> > > In comparison, our bound is $\widetilde{\mathcal{O}}\left(\eta^3 H^7 C\log|\mathcal{Q}|/n\right)$. While our $H$ dependence is higher than prior works, this is the structural cost of achieving the fast $1/n$ rate via KL strong convexity, consistent with Nayak et al. (2025) who incur the same $H^7$ factor in the online setting. We will incorporate this expanded comparison into the revision. We sincerely appreciate your thoughtful question and your willingness to raise the score.
> > >
> > > **References**:
> > >
> > > Cui & Du (2022) When are offline two-player zero-sum markov games solvable?
> > >
> > > Zhong et al. (2022) Pessimistic minimax value iteration: Provably efficient equilibrium learning from offline datasets.
> > >
> > > Zhang et al. (2023) Offline learning in markov games with general function approximation.
> > >
> > > Ye et al. (2024) Online iterative reinforcement learning from human feedback with general preference model.
> > >
> > > Nayak et al. (2025) Achieving logarithmic regret in kl-regularized zero-sum markov games.

---

### Official Review · Reviewer_A4ge · 2026-03-10

**Soundness:** 3
**Presentation:** 3
**Significance:** 3
**Originality:** 4
**Overall Recommendation:** 4
**Confidence:** 3

**Summary:**

Authors examine the problem of finding optimal strategies for two player stochastic games from offline information.
They use various techniques, including kl regularization to find optimal policies with respect to the reference policies.
The convergence is faster than the results from the literature.

**Compliance With Llm Reviewing Policy:**

Affirmed.

**Key Questions For Authors:**

You talk about Nash equilibria, why? Based on the results of Shapley 1953, stochastic games have an optimal strategy. Is it true or some small choice of the model invalidates the result?

How would the results change if the objectives were a discounted sum with no time horizon H.

Why do you add noise (line 180), does it make the problem more interesting? (model should give deterministic reward)

**Limitations:**

There should be a paragraph about limitation, about the choice in the model and something about relationship of \eta, n, and the optimality of policies.

**Strengths And Weaknesses:**

Soundness is good provided that the authors clarify some of my questions.
(My questions are mainly about some choices of the model and techniques, not about the soundness).

Presentation is good.

Significance is ok, some choices of the model seems restrictive (like time horizon).

Originality is very good.

---

> ### Author Rebuttal · Authors · 2026-03-31
>
> We appreciate your thoughtful suggestions. We are happy that you found our work highly original, and we address your questions in details below.
> ## For Questions:
> 1. >You talk about Nash equilibria, why?
>
> Thanks for your question. We use "Nash Equilibrium" (NE) to align with standard modern Multi-Agent RL (MARL) literature. In two-player zero-sum games, a NE is mathematically equivalent to the "optimal strategies" proven to exist by Shapley (1953). While Shapley guarantees their existence, our work addresses the **statistical challenge of learning** these strategies from a fixed, **finite dataset** where transition dynamics and rewards are unknown.
>
> 2. >How would the results change if the objectives were a discounted sum with no time horizon H.
>
> Thank you for this insightful question. We chose the finite-horizon episodic setting as it is the **standard framework** for establishing foundational sample complexity in offline Markov games (e.g., [Cui'22, Zhong'22]). While extending our results to the infinite-horizon discounted setting is a promising direction, it is technically non-trivial.
>
> Although the horizon $ H $ would naturally translate to the effective horizon $1/(1-\gamma)$, the stationary Bellman operator introduces recurrent error propagation challenges that are absent in stage-wise backward induction. In the episodic case, errors are decoupled step-by-step; however, in the discounted case, the target values shift alongside the policy updates, making it significantly more difficult to telescope estimation errors and formally establish the global $\tilde{\mathcal{O}}(1/n)$ rate. We believe the implicit pessimism mechanism of the KL-regularizer would remain effective in this regime, but we leave the formal analysis of this extension for future work.
>
> 3. >Why do you add noise (line 180), does it make the problem more interesting? (model should give deterministic reward)
>
> We incorporate sub-Gaussian noise into the reward signal for two reasons. First, it provides **greater generality**, as deterministic rewards are a special case where noise variance is zero. This ensures our guarantees hold in practical scenarios involving sensor noise or labeling variability. Second, stochastic reward models are **standard in RL theory** (e.g., [Cui'22, Zhong'22]), ensuring our $\tilde{\mathcal{O}}(1/n)$ rates remain valid under statistical fluctuations and are directly comparable to prior state-of-the-art benchmarks. We will clarify in Section 3 that the deterministic case is a subset of our analyzed model.
>
> ## For Limitations:
> 1. >There should be a paragraph about limitation, about the choice in the model and something about relationship of \eta, n, and the optimality of policies.
>
> Thanks for the suggestions. In the revision, we will add a Limitations section addressing our modeling choice. Regarding the relationship between $\eta, n$, and optimality: the parameter $\eta$ is calibrated to manage the "implicit pessimism." While a larger $\eta$ suppresses statistical noise more effectively (via the quadratic scaling in Lemma 6.6), it also increases regularization bias. By optimally scaling $\eta$ with $n$, we balance this trade-off to ensure the total suboptimality gap decays at the fast $\tilde{\mathcal{O}}(1/n)$ rate.
>
> We hope our clarifications resolve the questions you raised, and we are delighted to engage in any further discussion should that be helpful.
>
> ## Reference:
> [Cui'22] Cui & Du. When are offline two-player zero-sum markov games solvable? NeurIPS.
>
> [Zhong'22] Zhong et al. Pessimistic minimax value iteration. ICML.

---

> > ### Author Rebuttal · Reviewer_A4ge · 2026-04-02
> >
> > Thank you for your answer. This answers my questions.
> > If you have space, consider adding a small explanation about the technical difficulties of getting from H to the discounted sum.
> > I think that adding "limitations" paragraph improves the work.

---

### Official Review · Reviewer_BaNj · 2026-03-12

**Soundness:** 4
**Presentation:** 3
**Significance:** 2
**Originality:** 2
**Overall Recommendation:** 4
**Confidence:** 4

**Summary:**

This paper studies how to learn a NE in offline two-player zero-sum Markov games. The authors show that when the equilibrium includes a KL regularization term (i.e., a regularized NE), it is possible to achieve a faster $O(1/n)$ convergence rate using the ROSE algorithm. They further propose a more practical algorithm, SOS-MD, by approximating the equilibrium computation oracle in ROSE with mirror descent. The theoretical analysis shows that SOS-MD achieves the same $O(1/n)$ convergence rate.

**Compliance With Llm Reviewing Policy:**

Affirmed.

**Final Justification:**

I maintain my score.

**Key Questions For Authors:**

1. In my opinion, offline zero-sum Markov Game settings usually do not require many novel techniques to solve, given the established techniques in single-agent setting. Could the author clarify what the main challenge is for extending the single-agent setting to the zero-sum two-player Markov Game?

2. Is it possible to apply similar techniques to solve the general-sum Markov Game?

3. Could the authors further explain why KL regularization induces implicit pessimism? In prior work on online KL-regularized RL like [1], optimism is still typically required for exploration. I am therefore curious about the key reason why optimism is no longer necessary in this setting.

[1]. Zhao et al. Logarithmic Regret for Online KL-Regularized Reinforcement Learning, 2026.

**Limitations:**

Yes.

**Strengths And Weaknesses:**

### Strength

1. The writing of the paper is good. The technical introduction and the proofs are easy to follow.

2. It is appealing that the authors use mirror descent to approximate the equilibrium computation in SOS-MD and derive the corresponding convergence guarantee, rather than fully relying on an equilibrium computation oracle which is generally difficult to guarantee.

### Weakness

1. Although the main contribution of the paper is theoretical, the paper does not provide any experiments to demonstrate the effectiveness of the algorithm. In particular, the authors claim that one advantage of their approach is the ability to handle multi-step dynamics, but this benefit remains unclear without practical scenarios or empirical results that illustrate the advantage of modeling multi-step dynamics. Also, since the author proposes a more practical SOS-MD algorithm, it would be helpful if they could provide some preliminary empirical implementation approaches or guidance.

---

> ### Author Rebuttal · Authors · 2026-03-31
>
> Thank you for the constructive feedback. We are glad that you found the writing of our paper to be good and easy to follow, and that our approach to using mirror descent was appealing. We address each of your concerns below:
>
> ## For Weaknesses:
> 1. >Although the main contribution…
>
> Thanks for your question. Theoretical focus is **standard** for papers introducing new sample complexity frameworks in offline games (e.g., [Cui'22, Zhong'22]). Our primary goal is establishing the **first** $\tilde{\mathcal{O}}(1/n)$ statistical rate for multi-step settings and demonstrating how KL regularization provides **implicit pessimism**. To assist practitioners, we will add the following implementation guidance:
> - Function Approximation: Q-function estimation (Step 5) can utilize deep architectures (e.g., MLPs or Transformers) optimized via standard gradient-based methods like Adam.
> - Mirror Descent Loop: The self-play update (Steps 8–15) uses efficient closed-form exponentiated updates, shifting the policy toward the estimated advantage while anchored by $\pi^{ref}$.
> - Multi-step Dynamics: Implementation follows backward induction, recursively constructing regression targets from the final turn ($h=H$) to the first ($h=1$).
>
> ## For Questions:
> 1. >In my opinion, offline zero-sum Markov Game settings…
>
> The transition to the two-agent regime introduces unique technical and statistical complexities:
> - **Pessimism-Free**: Standard offline RL—even in the single-agent setting—typically requires **explicit pessimism** (e.g., LCB). Our work provides a pessimism-free approach, showing that KL regularization induces sufficient implicit pessimism to handle distribution shifts. This is a major advancement over both single-agent and multi-agent benchmarks.
> - **Optimal "Fast" Rates**: We achieve a $\tilde{\mathcal{O}}(1/n)$ rate for multi-step settings, surpassing the $\tilde{\mathcal{O}}(1/\sqrt{n})$ rates in [Cui'22, Zhong'22, Zhang'23]. Reaching this in games is challenging because policy coupling complicates the algebraic error cancellation used in single-agent settings.
> - **Stability**: Unlike single-agent maximization, minimax optimization introduces instabilities like oscillations. Our framework uses KL regularization as a stabilizer for these dynamics, ensuring convergence where standard single-agent extensions often fail.
>
> 2. >Is it possible to apply similar techniques to solve the general-sum Markov Game?
>
> Thanks for this forward-looking question. Extending our framework presents two non-trivial challenges:
> - Solution Concept and Uniqueness: General-sum games lack a unique Nash Equilibrium (NE) payoff. Our current duality gap analysis relies on saddle-point properties that do not directly translate to the general-sum setting. Furthermore, computing an exact NE in general-sum games is **intractable** (PPAD-complete, see Daskalakis et al., 2009, and Chen et al., 2009), even in the two-player case.
> - Complexity and Statistical Rates: While targeting more tractable concepts like Coarse Correlated Equilibria (CCE) is possible, the inherent correlation in CCE complicates the statistical error cancellation we leverage to achieve fast $\tilde{\mathcal{O}}(1/n)$ rates.
> We provide a foundational step for stabilizing offline learning in broader multi-agent general-sum settings and would like to leave the extension to general-sum games for future work.
>
>
> 3. >Could the authors further explain why KL regularization …
>
> Thanks for the question. We clarify the role of KL-regularization through two lenses:
> - **Online Optimism vs. Offline Pessimism**: In online RL (e.g., [1]), the primary challenge is exploration, which requires optimism to encourage taking risks in unknown regions to find optimal paths. Conversely, in the offline setting, the central challenge is robustness against distribution shift. Here, the agent should be pessimistic—staying within the "safe" support of the observed data to avoid exploiting erroneous value estimates in out-of-distribution regions.
> - **Implicit Pessimism via KL**: Our formulation induces this pessimism through the multiplicative re-weighting in Eq. (2). By anchoring the learned policy to the data-driven reference $\pi^{ref}$ and assigning no probability to actions unsupported by the reference policy, the algorithm effectively restricts the search to the safe support. This provides implicit pessimism, removing the need for the manually tuned explicit penalty terms (e.g., Lower Confidence Bounds) common in standard offline methods [Cui', Zhong' 22] .
>
> ## Reference:
> [Cui'22] Cui & Du. When are offline two-player zero-sum markov games solvable? NeurIPS.
>
> [Zhong'22] Zhong et al. Pessimistic minimax value iteration. ICML.
>
> [Zhang'23] Zhang et al. Offline learning in markov games with general function approximation. ICML.
>
> [Daskalakis'09] Daskalakis et al. The complexity of computing a Nash equilibrium. Comm. ACM.
>
> [Chen'09] Chen et al. Settling the complexity of computing two-player Nash equilibria. J. ACM.

---

> > ### Author Rebuttal · Reviewer_BaNj · 2026-04-02
> >
> > Thanks for the authors' response. My concerns have been addressed. I will keep my score.

---

### Official Review · Reviewer_1Cq3 · 2026-03-14

**Soundness:** 4
**Presentation:** 3
**Significance:** 3
**Originality:** 3
**Overall Recommendation:** 5
**Confidence:** 3

**Summary:**

This paper studies the problem of learning Nash equilibria in offline two-player zero-sum Markov games. The authors show that KL regularization can stabilize learning and ensure convergence to the equilibrium, avoiding the need to use pessimism.

The paper develops an algorithm called ROSE (Regularized Offline Sequential Equilibrium) which converges under an assumption of unilateral concentrability, improving known rates of convergence.
ROSE Can be broken into two steps, estimation of the Q-function (using least squares) and computation of the regularized equilibrium, a value update follows.
Unilateral concentrability corresponds to a distributional shifts of a policy pair where at least one of the policies must adhere to a Nash equilibrium policy.  The authors prove that the ROSE framework has $1/n$ statistical guarantees

**Compliance With Llm Reviewing Policy:**

Affirmed.

**Final Justification:**

I will maintain my score of accepting this paper, my opinion was largely reinforced by the authors reply.

**Key Questions For Authors:**

None

**Strengths And Weaknesses:**

Strengths:
I found the paper by and large well written with in-depth discussion on many of the choices made in the algorithm and their consequences. I think the overall results are solid and the paper makes a technical contribution to the field.

Weaknesses: The paper suggests that it can be used when the number of states go to infinity. The computation of $\hat{Q}$ on page 4 (possibly in several other places) shouldn't work when the number of states is infinite.

This paper seems to in line with other works that perform NPG updates such as "Faster Last-iterate Convergence of Policy Optimization in Zero-Sum Markov Games" by Cen et al. The results in this paper seem to be of the same order as the current work. I did not see a comparison with this work even though it appears highly relevant to me.

In section 4, the paragraph on implicit pessimism under equation (2) is very vaguely written. I think it can be written more concretely after an exposition on unilateral concetrability. The assumption prevents the reference policy from being "too small" in actions that are close to the equilibrium actions, which implies that if they are too small, they must be so for actions that aren't going to appear in the equilibrium policy.

---

> ### Author Rebuttal · Authors · 2026-03-31
>
> Many thanks for the encouraging and thoughtful feedback! We are grateful that you find our paper well written with in-depth discussion on many of the choices we make. We also appreciate the recognition that our results are solid and make a technical contribution to the field. We answer each remaining question in detail below.
>
> ## For Weaknesses:
> 1. >The paper suggests that it can be used when the number of states go to infinity.
>
> We appreciate you pointing this out and allowing us to clarify. Our theoretical framework does **support infinite state spaces**, and we will clarify the presentation in the revision to prevent any confusion.
>
> Because our approach relies on **general function approximation** (the class $\mathcal{Q}$) rather than tabular representations, we do not need to compute or store the policy for all infinitely many states simultaneously. In our Algorithm 2, the Self-Play Optimization loop (Lines 7-15) is only utilized to compute the value function. Crucially, to construct the empirical regression targets (Line 4), this value function only needs to be evaluated point-wise at the finite empirical next-states $s'_{i,h}$ present in the offline dataset. Therefore, solving the matrix game via mirror descent is a finite-dimensional operation per empirical state, making the algorithm highly scalable and **applicable to continuous or infinite state spaces**. Our statistical bounds reflect this: as shown in Theorem 6.3, the sample complexity scales with the function class capacity $\log(|\mathcal{Q}|/\delta)$ rather than the state space size $|\mathcal{S}|$.
>
> 2. >This paper seems to in line with other works that perform NPG updates such as "Faster Last-iterate Convergence of Policy Optimization in Zero-Sum Markov Games" by Cen et al. The results in this paper seem to be of the same order as the current work. I did not see a comparison with this work even though it appears highly relevant to me.
>
> Thank you for highlighting this relevant work. Cen et al. (2022) establish fast last-iterate convergence for zero-sum Markov games. We will include a comparison in our revision to clarify the following fundamental distinctions:
> - **Planning vs. Learning**: The most critical difference is that Cen et al. operate in a **full-information tabular setting**, where the transition probability kernel $P$ and reward function $R$ are fully known. Their work focuses on computation-efficient planning rather than learning from finite samples. In contrast, our work addresses the **offline learning setting**, where the transition model is unknown and must be estimated from a fixed, finite dataset.
> - **Statistical Uncertainty**: Because Cen et al. assume a **known model**, they do not encounter statistical estimation error or the challenges of distribution shift. Our analysis explicitly accounts for *both* the optimization error $\tilde{\mathcal{O}}(1/T)$ and the statistical estimation error $\tilde{\mathcal{O}}(1/n)$ inherent in learning from finite offline datasets.
> - **Implicit Pessimism vs. Explicit Optimism**: Cen et al. utilize optimistic updates to achieve stability in a planning context. Our SOS-MD algorithm instead employs anchored KL-regularization to provide implicit pessimism. This acts as the offline counterpart to their optimism, naturally preventing distribution shift into low-coverage regions without requiring the **manually tuned** explicit pessimism common in other offline RL methods [Cui'22, Zhong'22].
>
> 3. >In section 4, the paragraph on implicit pessimism under equation (2) is very vaguely written. I think it can be written more concretely after an exposition on unilateral concetrability.
>
> We appreciate the constructive feedback. In the revision, we will update Section 4.2 to concretely link KL-regularization to Unilateral Concentrability (Assumption 6.1).
>
> Equation (2) provides implicit pessimism by anchoring the learned policy to $\pi^{ref}$ via multiplicative re-weighting. This mechanism ensures that actions lacking support in the reference policy receive no probability, preventing the agent from exploiting overoptimistic $Q$-value estimates in low-coverage regions. Combined with Assumption 6.1—which guarantees the dataset covers the Nash Equilibrium and its unilateral deviations—this approach restricts the search to "safe" support and eliminates the need for manually tuned pessimistic bonuses.
>
> We hope our responses address all remaining concerns, and we would be very happy to engage in further discussion if that would be helpful.
>
> ## Reference:
>
> [Cui'22] Cui & Du. When are offline two-player zero-sum markov games solvable? NeurIPS.
>
> [Zhong'22] Zhong et al. Pessimistic minimax value iteration. ICML.

---

> > ### Author Rebuttal · Reviewer_1Cq3 · 2026-04-01
> >
> > All my points were addressed adequately.

---

### Decision · Program_Chairs · 2026-04-30

**Decision:**

Accept (regular)

**Comment:**

This paper studies the problem of learning Nash equilibria in offline two-player zero-sum Markov games. The authors show that KL regularization can stabilize learning and ensure convergence to the equilibrium, avoiding the need to use pessimism. The paper develops an algorithm called ROSE (Regularized Offline Sequential Equilibrium) which converges under an assumption of unilateral concentrability, improving known rates of convergence. The authors prove that the ROSE framework has statistical guarantees.

Overall, the Reviewers were slightly positive on this paper. I personally believe that the paper provides a nice contribution to the literature on learning in games, even though the significance of the obtained results is somehow limited by the assumption needed (unilateral deviation policies), which may be hard to be met in practice. I also think that some synthetic experiments would have helped to strengthen the paper (as also suggested by some Reviewers). Thus, given the above observations, I suggest (weakly) acceptance of the paper.